# Adaptive stretching of representations across brain regions and deep learning model layers

Xin-Ya Zhang [1] ✉, Sebastian Bobadilla-Suarez [2], Xiaoliang Luo[2], Marilena Lemonari [3], Scott L. Brincat [4,5], Markus Siegel [6,7], Earl K. Miller [4,5] & Bradley C. Love [8] ✉

Prefrontal cortex (PFC) is known to modulate the visual system to favor goal-relevant information by accentuating task-relevant stimulus dimensions. Does the brain broadly re-configures itself to optimize performance by stretching visual representations along task-relevant dimensions? We considered a task that required monkeys to selectively attend on a trial-by-trial basis to one of two dimensions (color or motion direction) to make a decision. Although effects were most prominent in frontal areas, representations stretched along task-relevant dimensions in all sites considered: V4, MT, lateral PFC, frontal eye fields (FEF), lateral intraparietal cortex (LIP), and inferotemporal cortex (IT). Spike timing was crucial to this code. A deep learning model was trained on the same visual input and rewards as the monkeys. Despite lacking an explicit selective attention or other control mechanism, by minimizing error during learning, the model's representations stretched along task-relevant dimensions, indicating that stretching is an adaptive strategy.

Adaptive agents, whether biological or artificial, configure themselves for the learning and decision task at hand. For example, when searching for one's car keys, one might focus on features consistent with the shape and metallic sheen of a key. Prefrontal cortex (PFC) modulates the visual system to favour goal-relevant information[1,2]. Goal-directed attention can reconfigure the visual system to highlight task-relevant features and suppress irrelevant features[3]. Over longer time horizons, learning processes build internal representations that reflect these task pressures[4,5] with vmPFC critical for determining which aspects of the current context are relevant[6]. Attention can be viewed as stretching representations along relevant dimensions[7–9], which is reflected both in behavior and brain response[5,10,11]. While numerous attentional influences on neural activity have been observed[12–15], we focus on the specific case of adaptive stretching along

a goal-relevant stimulus dimension. Stretching accentuates differences along goal-relevant dimensions while minimizing differences along goal-irrelevant dimensions. For example, when searching for one's car keys, objects varying in metallic sheen should become more dissimilar.

One possibility is that the brain uses stretching to radically re-configures itself across regions to optimize for the current task. Consistent with this possibility, effects of endogenous attention have been observed across the visual cortical hierarchy[16–18], including as early as V1[19,20]. Alternatively, some areas devoted to modality specific processing, such as the middle temporal cortex (MT) for movement direction and visual area V4 for object properties like color, may be invariant across task contexts.

The ideal study to evaluate whether radical reconfiguration occurs would record from multiple brain sites while cueing the

[1]Center for Interdisciplinary Studies and Department of Physics, School of Science, Westlake University, Hangzhou, PR China. [2]Department of Experimental Psychology, University College London, London, UK. [3]Computer Science, University of Cyprus, Nicosia, Cyprus. [4]The Picower Institute for Learning and Memory, Massachusetts Institute of Technology, Cambridge, MA, USA. [5]Department of Brain and Cognitive Sciences, Massachusetts Institute of Technology, Cambridge, MA, USA. [6]Department of Neural Dynamics and Magnetoencephalography, Hertie Institute for Clinical Brain Research, University of Tübingen, Tübingen, Germany. [7]Centre for Integrative Neuroscience, University of Tübingen, Tübingen, Germany. [8]Los Alamos National Laboratory, Los Alamos, NM, USA. ✉e-mail: zhangxinya@westlake.edu.cn; love@lanl.gov

relevant dimension for a categorization decision on a trial-by-trial basis. We analyzed spiking data from a study[21] that met these criteria. On each trial, the rhesus monkeys viewed a cue that indicated whether color or direction of movement was the relevant dimension for the decision (Fig. 1a). Once cued, colored dots moved with 100% coherency in a given direction. The monkeys responded by moving their eyes left or right, depending on the value of relevant (either color or motion) stimulus dimension (Fig. 1c). Recording sites included lateral PFC, frontal eye fields (FEF), lateral intraparietal cortex (LIP), inferotemporal cortex (IT), V4, and MT. Unlike previous fMRI studies[5,11] that considered stretching along relevant dimensions during categorization decisions, monkey multi-unit spiking data affords the possibility of evaluating whether spike timing, over and above spiking rate, is a critical component of the neural code.

Considering goal-relevant stretching along a relevant dimension in non-human animals offers opportunities to examine how control systems themselves can be learned and configured absent language instruction. Non-human animals, like artificial neural networks (ANNs), are naive to these laboratory tasks and, unlike verbally instructed humans, have to learn how to allocate their attention through trial-and-error. Nevertheless, non-human animals, including rats, can learn to selectively attend to relevant dimensions in related paradigms[22].

While models of goal-directed attention contain dedicated attention control systems that selectively weight relevant dimensions[7–9,23,24], one possibility is that these control systems themselves can be learned by non-human animals and ANNs, enabling them to reconfigure themselves in response to task cues. According to this hypothesis, brains and ANNs are powerful statistical learning machines that build control structures that adaptively stretch learnt representations along relevant dimensions to facilitate task performance. In effect, these systems may build the cognitive machinery that is presupposed in previous work.

To evaluate this possibility, we constructed an ANN consisting of a deep convolutional neural network (CNN) and a stacked long short-term memory (LSTM). The CNN was pretrained to perform object recognition on naturalistic images[25]. The CNN part of the ANN, akin to the monkeys' visual system[26–31], was assumed to be developed and stable prior to the study. Thus, its weights were fixed. The sequences of images forming a trial (e.g., the task cue and the moving dots; Fig. 1a) were fed into the CNN, whose outputs served as inputs to the LSTM[32]. The LSTM is a recurrent network that can process time series to make the left/right decision the monkeys did (see Supplementary Information for details). Like the monkeys, the LSTM part of the ANN learned through trial-and-error from sequences of images, absent verbal instruction.

Of course, the brain embodies constraints that ANNs do not. Brain regions may be limited to processing certain stimulus dimensions and may lack the flexibility to completely reconfigure. For example, in the present investigation, V4 and MT may remain bounded to processing color and motion, respectively. However, to

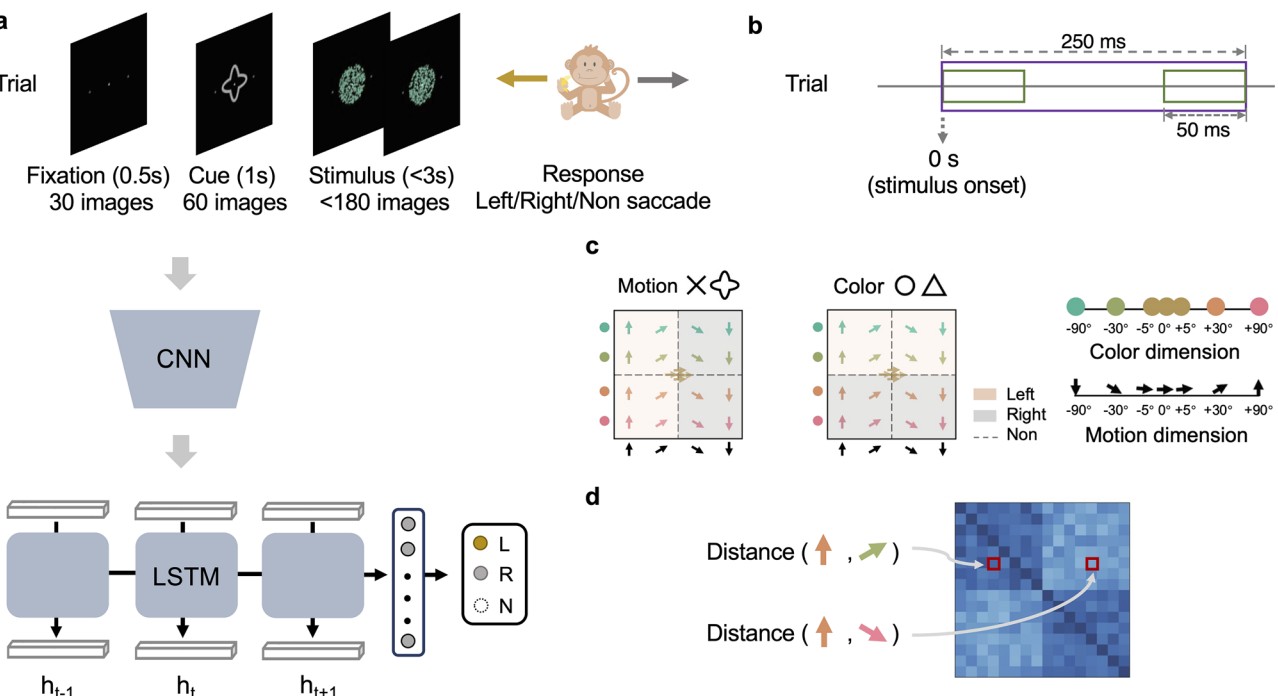

**Fig. 1 | Overview of the behavioral task and CNN-LSTM modeling. a** Monkeys were shown images depicting the fixation point (0.5 s), cue (1 s), and stimulus (up to 3 s). During each trial, monkeys held central fixation until they responded by making either a left or right saccade. They were not to respond on ambiguous trials. These same images were shown to the ANN model consisting of a pretrained CNN for visual processing and a stacked (i.e., multilayer) LSTM that learned how to appropriately respond given the task cue and stimulus. Like the monkeys, the CNN-LSTM learned through trial-and-error to respond left (L), right (R), or to withhold (N) a response on ambiguous trials. **b** Spiking data were analyzed over a 250 ms period (shown in purple) that began at stimulus onset. Analyses either considered this time period as a whole or in 50 ms overlapping segments using a sliding window (shown in green) that was 50 ms wide and moved in 10 ms steps. **c** There were 21 color-motion stimuli constructed from the color and motion space. Five stimuli were ambiguous at the origin of the color-motion space. The remaining 16 stimuli were evenly spread across the four quadrants. A cross or quatrefoil cue indicated that motion was relevant to the decision. A circle or triangle cue indicated the color was relevant. In both task contexts, the monkeys responded with a leftward or rightward saccade. **d** From the 16 non-ambiguous trials, we constructed 16 × 16 dissimilarity matrices (i.e., RDMs) for data analyses. To determine the value of an entry in the RDM, the spiking measures for two items would be compared using a particular distance function (e.g., ISI, Euclidean distance, etc.) and the value was stored in the row and column corresponding to the two items. Likewise, RDMs were made for the ANN using the activity patterns from LSTM's hidden unit activations.

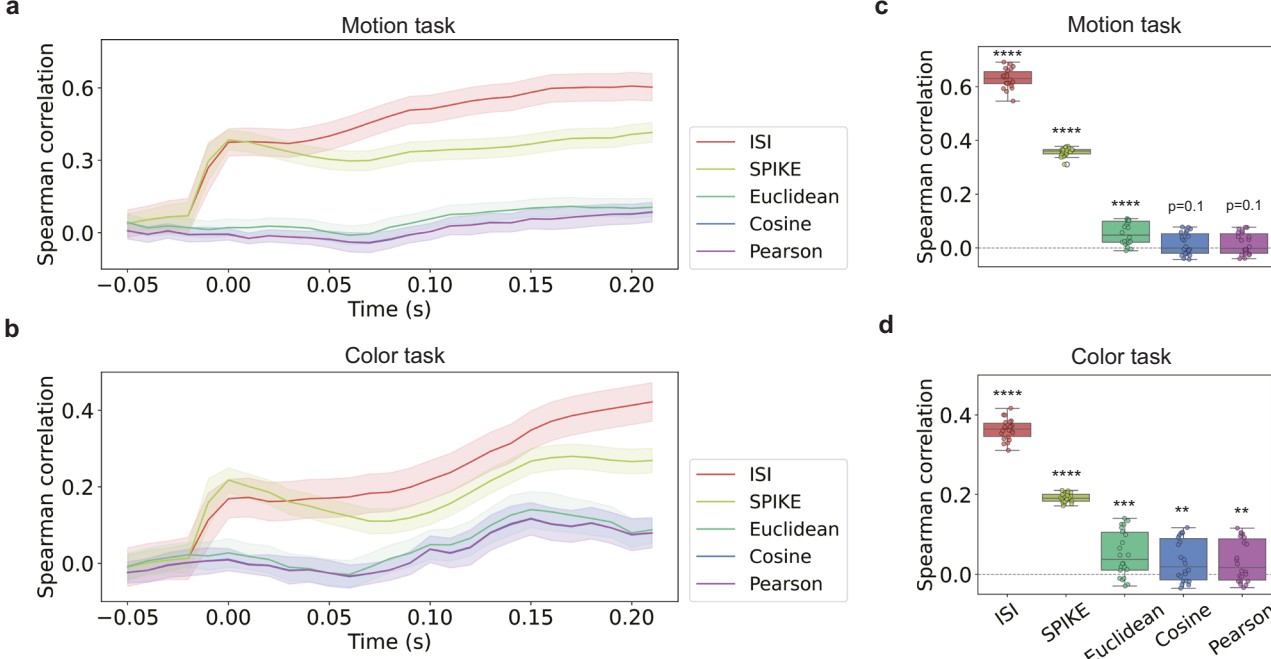

**Fig. 2 | Spike timing measures best capture the experimenter intended coordinates.** We compared the Spearman rank correlation between dissimilarity matrices (i.e., RDMs) constructed from the monkey data for the 16 items (see Fig. 1d) with an RDM derived from the experimenter-intended stimulus coordinates (see Fig. 1c). Higher correlation indicates better agreement. **a** When motion was relevant, the timing measures ISI and SPIKE surpass rate coding measures and best capture the experimenter-intended coordinates. Time bins are 0.05 s wide and start at the indicated time. For instance, time bin that begins at 0 s covers 0–0.05 s. Time bins beginning at −0.05–0.21 s (27 time points) are shown. The mean correlation (line) is computed across 53 independent recording sessions, and the shaded bands indicate the 95% confidence interval. **b** The same pattern was found when color was relevant to the monkey's decision with ISI once again proving best. Time on the horizontal axis is measured from stimulus onset (see Fig. 1b). **c, d** The same pattern of results holds when the entire 250 ms time period is analyzed as a whole. Box plots indicate the distribution across 250 ms time bins ($n = 22$). The box spans from the first to the third quartile, with the horizontal line inside the box indicating the median. Whiskers extend to the most extreme values within 1.5 times the inter-quartile range. A two-tailed test was conducted to determine whether the correlation is different from 0, with statistical significance noted ($^*p < 0.05$, $^{**}p < 0.01$, $^{***}p < 0.001$, $^{****}p < 0.0001$). Exact $p$ values are reported for each metric: motion task: ISI $p = 7.2e^{-28}$, SPIKE $p = 2.1e^{-30}$, Euclidean $p = 5.0e^{-6}$, Cosine and Pearson $p = 0.1$; color task: ISI $p = 7.6e^{-26}$, SPIKE $p = 6.6e^{-28}$, Euclidean $p = 4.1e^{-4}$, Cosine $p = 6.2e^{-3}$ and Pearson $p = 6.7e^{-3}$.

the extent the brain can be viewed as an overparameterized statistical learning engine[33], the ANN and brain should converge. We predict this convergence will be realized by representations stretching along the goal-relevant stimulus dimension. While each brain region considered is associated with a different cognitive function[21], we predict that stretching will broadly be observed. Akin to how object category signatures are observed in early visual areas as a result of activity in higher-level areas[34], task-relevant signals should propagate and representations should stretch in a system optimizing performance. Observing stretching in the ANN would demonstrate how control and top-down attentional mechanisms can arise from simply maximizing task performance, as opposed to relying on preordained mechanisms. Rather than serving as a model of the brain, the ANN is intended to illustrate that a discriminative model will adaptively stretch its representations to minimize error[35], which invites general comparisons to the brain's operation.

To foreshadow the results, we observed stretching across all layers of the ANN's LSTM and in all brain regions, though V4 (color) and MT (motion) were largely modality-bound. We also used a cognitive model with a dedicated attention mechanism to assess stretching in the brain data, which corroborated these conclusions. Finally, we analyzed the similarity and fidelity of neural representations[36] and discovered that spike timing was critical to how the brain coded representations of the stimuli and task.

## Results

### Spike timing and neural representation

One initial question is which measure of neural similarity is most aligned with the neural recordings[36]. To address this question, we applied representational similarity analysis (RSA)[37], a method that computes and compares representational dissimilarity matrices (RDMs) to characterize the information encoded in brain data across various modalities, including electrophysiology recording[38–40], as well as in computational models. Here, we used RSA to determine how spiking neural activity relates to behavioral data (250 ms period data was used as our default setting, as indicated in Fig. 1b). Specifically, by assessing various neural similarity measures, we determined which one maximized representational similarity with the experimenter-defined stimulus coordinates of color and motion.

For some time interval, rate coding measures sum the spikes at each recording site to calculate distances. For rate coding measures, spike timing within the interval is inconsequential. For example, the Euclidean distance between two items involving five recording sites is simply the Euclidean distance between the two five-dimensional vectors, where each vector entry is the sum of the spikes at a site. In contrast, rather than simply sum across the temporal interval, spike timing measures take into account temporal information within the interval, such as the timing between spikes for two items at a site, pooling these distances across recording sites for an overall measure

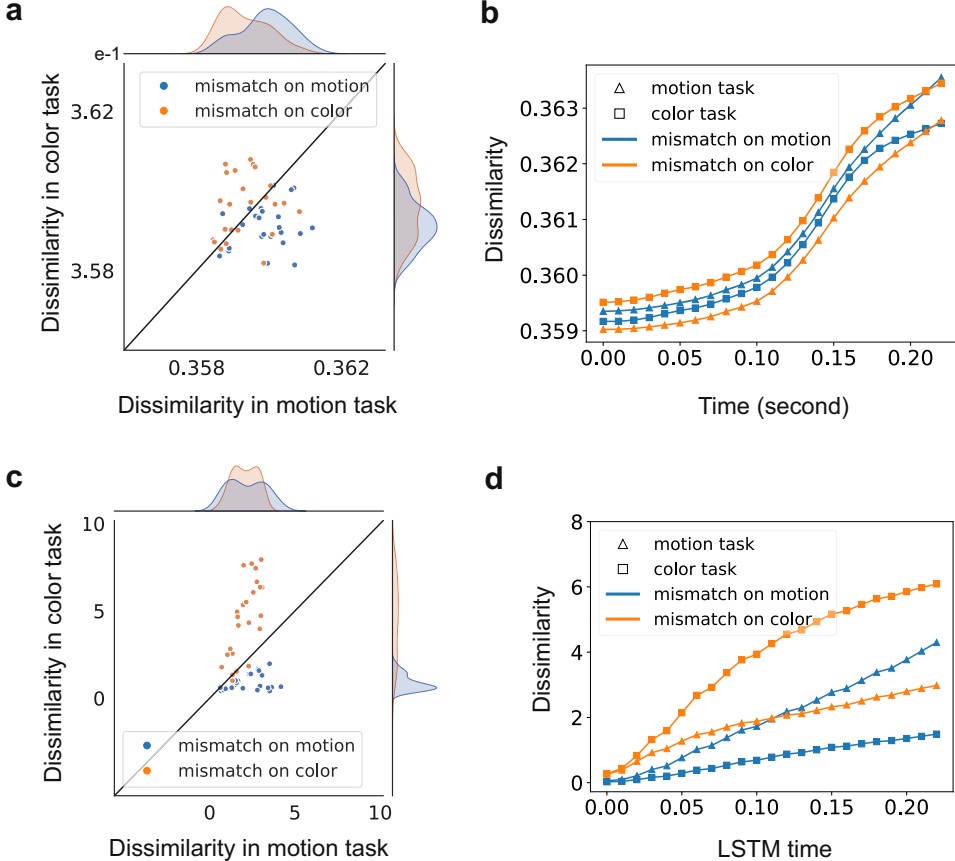

**Fig. 3 | Dimensional stretching occurs in both neural data and model representations. a** Dissimilarity between item pairs mismatching on motion or color is increased when that dimension is task relevant. The density distributions of these dissimilarities also indicate this task-modulation. The time period considered is from stimulus onset until 0.25 s post onset. **b** Changes in dissimilarity over trial time (0.05 s time bins beginning at the indicated time). Item pairs that mismatch on motion are more different in the motion-relevant context than in the color-relevant context. Mirroring, item pairs that mismatch on color are more different in the color-relevant context than in the motion-relevant context. Time bins that begins at 0–0.22 s (23 time points) are shown. **c**, **d** The CNN-LSTM model (layer 4, more layers see Supplementary Fig. S11) shows the same qualitative pattern of performance–stimulus pairs that mismatch on the task relevant dimension were most dissimilar. LSTM time is estimated based on the stimulus images input within the long (**c**) and sliding (**d**) time window.

of distance. From stimulus onset, measures of similarity did differ in how well they aligned neural similarity with the experimenter-defined stimulus coordinates of color and motion, $F(4, 105) = 148.38$, $p < 0.0001$. The timing measures, ISI and SPIKE, did surpass the non-timing measures (Fig. 2). In particular, ISI performed best and will be used throughout the remaining analyses in this contribution (see Bonferroni corrected t-tests in Supplementary Tables S1 and S2). ISI emphasizes the relative intervals between spikes, whereas SPIKE also incorporates the absolute timing of spikes, which can be useful for evaluating synchrony between spike trains (see Methods and Supplementary Information). Thus, ISI may have an advantage over SPIKE given the potential absolute timing differences across trials in temporal codes. The advantage of spike timing measures over rate-based coding held across all recording sites (Supplementary Tables S3–S8 and Fig. S6).

**Dimensional stretching found in both brain and model activity**
One key question is whether the brain radically reconfigures itself across regions to optimize for the current task, which in the present study would manifest as stretching along the relevant dimension (color or motion) on each trial. Likewise, we consider whether the LSTM, simply by maximizing performance absent an explicit control or attentional mechanism, will adaptively stretch its representations.

Stretching was assessed by considering item pairs that mismatched on one dimension and matched on the other. For example, we predict items mismatching on color and matching on motion should be more dissimilar when color is relevant than when motion is relevant. This prediction was confirmed both in the brain and LSTM activity (Fig. 3). Mismatches along a stimulus dimension were more consequential when that dimension was task relevant.

We assessed this stretching effect at the whole brain an regional level by two-way repeated measures ANOVA with fixed factors of task (color or motion relevant) and mismatching dimension (color or motion mismatch) with a random effect for the monkey subjects. Stretching is consistent with an interaction such that dissimilarity is greatest for pairs mismatching on the dimension (color or motion) that is task relevant. This interaction was significant in the whole brain analysis, as well for analyses of PFC, FEF, LIP (see Supplementary Fig. S7, 250 ms period data in Tables S9). Following previous studies[34,41–43], our primary analyses were over a 250 ms period. To evaluate the potential impact of analysis duration, we extended the time window to 500 ms and found that extending this period did not qualitatively alter our key results (extended 500 ms period data in Table S10). For comparison, rate coding results are reported in Supplementary Figs. S8–10 and Tables S11, S12.

We analyzed stretching in the LSTM using the same analysis strategy used for the brain, but simulation served as the random effect

instead of monkey and a third continuous factor was included for LSTM layer (1–6). Our main prediction was that the degree of stretching would increase across layers (i.e., stretching would be most pronounced at advanced layers), which would lead a three-way interaction, which we observed ($F(1, 52981) = 1.59e4$, $p < 0.0001$). See Supplementary Fig. S11 and Table S13 for full details.

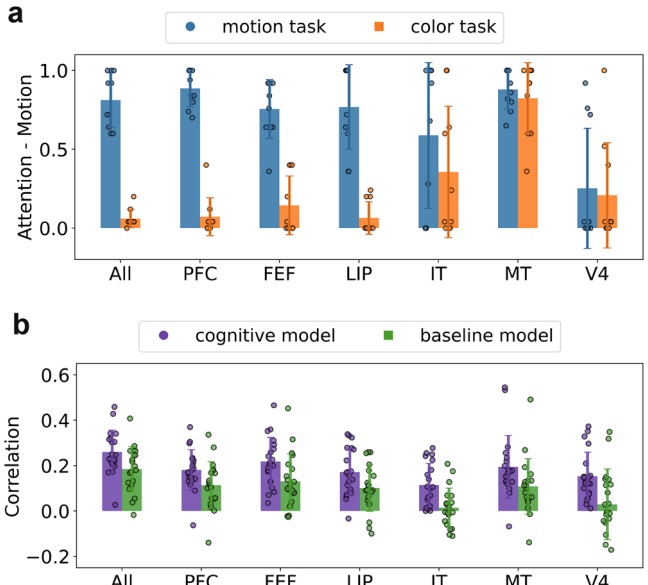

**Fig. 4 | Task-relevant attention allocation as estimated by the cognitive model.**
**a** The allocation of attention (estimated by fitting a cognitive model) in the whole brain and for the six brain regions is shown. The bars represent the averaged attention weights obtained by fitting the cognitive model using 5-fold cross-validation with weights trained on four folds within the same task context per subject. The error bars indicate the standard deviation, capturing the variability across 20 iterations, which were conducted across two task contexts and two subjects. Bars indicate the mean and error bars represent standard deviation across 20 held-out folds. **b** Representational similarity analyses were conducted that Spearman correlated a model's estimated similarity structure with that of the brain data for the 20 held-out validation folds. The baseline model was identical to the cognitive model, but was fitted to two folds from the color task and two folds from the motion task rather than four folds from the relevant task. The cognitive model is better aligned with the brain data in all cases, with higher correlation suggesting the presence of stretching in the neural representations.

## Model-based analyses of stretching in brain regions and model layers

To assess how the six brain regions and LSTM model layers are modulated across dimensions in a quantitative manner, we conducted a cognitive model-based analysis (adapted from the generalized context model[8]) using 5-fold cross-validation. Here, we hypothesize that attention weights are allocated to task-relevant and task-irrelevant dimensions and determine these weights by maximizing the correlation between RDMs from each fold of brain data and the experimenter-defined stimulus coordinates for color and motion (see "Methods" for formula details). Unlike the previous analysis, this model-based analysis uses all the data as opposed to select pairs of items and assumes a dimensional structure (color and motion) when estimating stretching along the relevant stimulus dimension. These aspects may make this analysis more sensitive and reveal stretching across all six brain regions, whereas the previous analysis only found significant effects in PFC, FEF, and LIP.

Specifically, we conducted a cognitive model that assigns $w_m$ (ranging from 0 to 1) to the motion dimension and $1 - w_m$ to the color dimension. By optimizing the cognitive model parameters to best match observed stimulus coordinates, we inferred the relative attention of motion and color information in the task context. Here we employed a 5-fold cross-validation procedure to train the attention weights $w_m$ where 4 folds were used for training, and the remaining fold was used for RSA-based correlation score within the same task context per subject. We observed that more attention parameters (i.e., $w_m$ on the motion dimension) are placed on the motion dimension when motion is relevant (Fig. 4a).

To assess the effectiveness of our cognitive model-based analysis, we compared it to a baseline model structurally identical to the cognitive model but fitted to two folds from the color task and two folds from the motion task. We evaluated the RSA-based correlation score using a one-way repeated ANOVA with a fixed factor of model (cognitive or baseline model) and a random effect for the monkey subjects. A stronger correlation was observed for the cognitive model and found to be statistically significant across all brain regions examined (whole brain: $F(1, 37) = 5.96$, $p = 0.0195$; PFC: $F(1, 37) = 4.84$, $p = 0.0342$; FEF: $F(1, 37) = 6.31$, $p = 0.0165$; LIP: $F(1, 37) = 4.73$, $p = 0.0362$; IT: $F(1, 37) = 11.8$, $p = 0.0015$; MT: $F(1, 37) = 4.31$, $p = 0.0449$; V4: $F(1, 37) = 10.2$, $p = 0.0028$; see Fig. 4b and Supplementary Table S14). While significant, MT and V4 exhibit weaker weight changes across tasks (Fig. 4a), which is consistent with known functional specializations of these regions—for example, the involvement of MT in motion processing and V4 in encoding shape and color information[44,45]. This

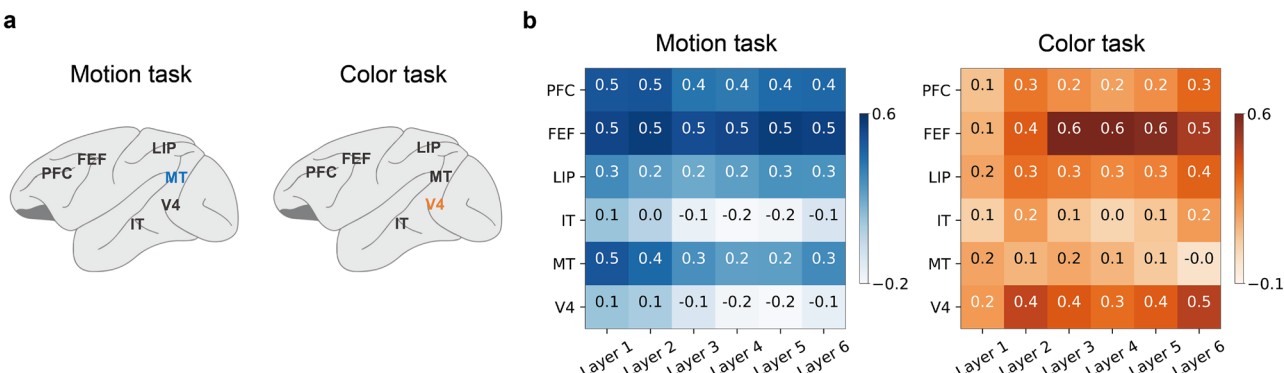

**Fig. 5 | Alignment between brain region and LSTM layers. a** One hypothesis, supported by our previous analyses, is that recording sites, with the exception of MT (motion fixated) and V4 (color fixated), should align equally well with LSTM activity in the motion and color tasks. **a** adapted from Macauley, S. B. (2020). Monkey Brain. Zenodo. https://doi.org/10.5281/zenodo.3910249. **b** Alignment, as measured by the Spearman correlation of the RDMs (Fig. 1d) for brain and model activity, is shown. Most brain regions generally aligned across model layers and tasks. IT was generally misaligned. In contrast, MT better aligned in the motion task, whereas V4 better aligned in the color task. Consistent with previous analyses, MT and V4 only moderately reconfigure as a function of task.

finding suggests that the allocation of attention to motion or color features is adaptively modulated based on task contexts, and that this modulation is particularly pronounced in all brain areas.

We conducted a cognitive model-based analysis on the LSTM that paralleled the previous analysis on the brain data. The LSTM afforded the opportunity to evaluate stretching across layers and training. The results were analyzed by a three-way repeated ANCOVA on the Spearman correlation scores, incorporating fixed factors of model (cognitive or baseline, a categorical variable), training step (1–10, a continuous variable) and LSTM layer number (1–6, a continuous variable), and with simulations (4 simulations) as a random effect. The key finding was that the cognitive model was superior to the baseline model ($F$(1, 4789) = 6.2$e$4, $p < 0.0001$, see Supplementary Table S15 for other statistics). The fitted attention parameter of the cognitive model was increasingly task driven as training progressed and at more advanced LSTM layers (Supplementary Fig. S13), whereas the baseline model's fitted parameter was flat across these two factors (Supplementary Fig. S14).

### LSTM more fully re-configures itself based on the task goal than MT and V4

Despite the LSTM and brain both adaptively stretching representations depending on the task, there is no straightforward layer-to-brain correspondence between LSTM layers and brain regions (Fig. 5). Previous analyses (e.g., Fig. 3) indicated that, relative to the brain, the LSTM reconfigures more responses to the task. Whereas LSTM layers freely reconfigure as a function of the task context, areas like MT and V4 appear dedicated to processing motion and color, respectively. While MT and V4 do exhibit stretching (e.g., Fig. 4), they appear less flexible and do not reconfigure to the same extent as the prefrontal cortex. Accordingly, V4 shows a decent correspondence with LSTM layers when color is relevant, whereas MT does when motion is relevant.

To quantify this relationship, we considered how the RSA-based correlation between brain region (V4 or MT) and model-layer activity varied with task context. We performed a three-way repeated ANCOVA with fixed factors of region (MT or V4), task (motion or color relevant) and layer (1–6, a continuous variable) and a random effect (4 simulations). The interaction between region (MT or V4) and task (color or motion) was $F_{1,85} = 670.5$, $p < 0.0001$ (Supplementary Table S16), indicating that MT aligned more with the LSTM when motion was relevant whereas V4 did when color was relevant.

## Discussion

We considered whether the brain reconfigures itself to optimize for the current task. For example, do neural representations of stimuli stretch along dimensions relevant to the task? We analyzed data from a task that, depending on a visual cue at the beginning of each trial, required monkeys to selectively attend to one of two stimulus dimensions to make a decision. We found support for the hypothesis that the brain radically reconfigures itself to optimize performance by accentuating differences along task-relevant dimensions. Despite differing functions ascribed to the brain regions considered, including cognitive control, object recognition, motion perception, and evidence accumulation, all regions showed adaptive stretching to varying degrees. An ANN that was trained on the same visual information as the monkeys also displayed task-relevant stretching as a consequence of trying to reduce error, indicating that stretching is an adaptive strategy that can arise in systems that aim to optimize performance.

Stretching was established in the monkey data by analyses that compared stimulus pairs matching or mismatching on a relevant dimension. Pairs that mismatched on a relevant dimension were more dissimilar than those that mismatched on an irrelevant dimension (Fig. 3). We also evaluated stretching by using a model-based approach in which a cognitive model with a selective attention parameter was fit to the neural data (Fig. 4). Using the more sensitive model-based

analyses, we found evidence of task-relevant stretching in all recording sites.

One way ANN activity diverged from the brain data is that the ANN's LSTM layers freely reconfigured themselves as a function of the task context. In other words, unlike the brain, the ANN was not modality-bound. Accordingly, there was no straightforward relationship between brain regions and LSTM layers. For example, when color was relevant, V4 and network activity correlated, whereas when motion was relevant, MT and network activity correlated (see Fig. 5). While V4 and MT, like other brain regions, did reconfigure according to the task, there appears to limits to this flexibility. However, feature specialization can be seen in the LSTM's dynamics. Specifically, we observed that earlier model layers reach their peak correlation more rapidly in the motion context, whereas later layers achieve their highest correlation earlier in the color context (Supplementary Figs. S16 and S17). In tasks with richer stimuli, perhaps related models would display feature hierarchies not just across layers[46,47] but also in time.

Finally, although not a primary hypothesis, the monkey spiking data afforded the possibility of evaluating whether spike timing or rate-based measures better characterize the neural code. Spike timing did appear relevant to how the brain represented stimulus information (see Fig. 2). These findings suggest a roll for spike timing that should be further explored, including in computational models. While our ANN was a recurrent network that processed the sequences of images forming a trial through time, the activity of each of its units was a scalar akin to rate-based coding. New models will need to developed to assess hypothesis concerning the functional role of spike-timing information in decision making.

One strength of the modeling approach is that the ANN was applied to the same image sequences forming a trial as those the monkeys experienced. Rather than attempt to incorporate biological constraints, we treated the model as a general-purpose statistical machine and observed how it tailored itself to the task. Under this approach, convergences between the ANN and the monkeys may reflect general information processing principles, whereas divergences may reflect unique properties of biological systems.

We found that the ANN performed as if it had a cognitive control mechanism that reconfigured network layers based on the task cue, stretching representations along task-relevant dimensions. In contrast, cognitive models, like the model we used in our analyses (Fig. 4), have dedicated attentional mechanisms. The ANN results suggest that general-purpose learning systems can learn to control themselves absent purpose-built control circuits. In terms of divergences, we found that, unlike V4 and MT, the ANN was not modality-bound by layer. This result suggests that there is a benefit or constraint in biological systems for localized processing of perceptual streams or dimensions.

Although this is just one study and model, we believe this modeling approach could complement other approaches in the literature, such as approaches that aim to test a priori hypotheses regarding correspondences between brain regions and model layers, as is commonly done in the object recognition literature[26–31,34]. Our approach may prove useful in other domains, including the meta-learning challenge of "learning to learn". In particular, one interesting question is whether the implicit control structures the ANN developed for the present task would provide a useful starting point for mastering new tasks that involve selective attention. There might be a great deal to learn about the brain by considering the computational challenges it faces.

## Methods
### Experiment and simulation description
In this work, we aim to model monkey brain learning on a flexible visuomotor decision-making task with our CNN-LSTM architecture. In the original experiment[21], two rhesus (one male and one female)

monkeys were shown a series of stimuli videos and learned to categorize the color or the orientation of stimuli dots based on cues they saw. Recordings were performed using three stereotactically positioned recording chambers, which covered the frontal cortex (FEF and PFC), parietal cortex (LIP), and occipitotemporal cortex (IT, MT, and V4). There were 21 color-motion stimuli (5 stimuli on or near the category boundary and 16 stimuli evenly spread across the four quadrants) from 7 possible colors and 7 possible motion directions (Fig. 1c). On each trial, monkeys saw one stimulus (dot pattern), that is, all dots with the same color moved 100% coherency in the same direction. Depending on the task cued at the beginning of each trial, the monkeys categorized either the color (red vs. green) or motion direction (up vs. down) of the stimulus and reported their percept with a left or right saccade. We did our best to replicate the training procedure as faithfully as possible by tasking our model with the same categorization problems, using identical fixation, cue and stimulus images (dot pattern) and training our model in a trial-by-trial fashion. We detail how experiments were performed while highlighting and justifying the necessary adjustments we have made in Supplementary Information.

To model the learning process and relate the neural data of trained monkeys, we employ training trials and test trials into our model and predict responses. Although these trials use the same color-motion stimulus (dot pattern), the dots' initial position in each trial is random, exactly as the original experiment. Matched with the original experiment, we include ambiguous trials of five boundary stimuli in our training and test, but excluding them from all data analyses, that is, we only consider 16 color-motion stimuli in our analyses. We use L (left) to represent the categorization of stimuli with upward motion in the motion task or stimuli made of greenish dots in the color task as the same class. Likewise, we use R (right) to represent the categorization of stimuli with downward motion in the motion task or stimuli made of reddish dots in the color task as the other class. We use N (no response) to represent ambiguous trials in training ANN like monkeys did.

## CNN-LSTM architecture
We tailor a CNN-LSTM framework containing a deep CNN and a stacked LSTM neural network (multilayer LSTM). The overall learning task of our model is to categorize the same set of stimuli (a left or right saccade based on either motion or color and no response if ambiguous trials), which is identical to the training procedure as monkeys did. The CNN front-end is used to simulate the monkey visual system and the stacked LSTM is to simulate multiple brain regions, such as MT, V4 which might be involved in the learning task.

Formally, we denote the CNN front-end as $\mathbf{f}$ which in our case is a pre-trained VGG-16 up to the pre-softmax layer. Given a trial of $n$ stimuli, $T_i$, $\mathbf{f}$ transforms raw images into representations $Q \in \mathbb{R}^{n \times d}$, where $d$ is the size of the pre-softmax layer,

$$\mathbf{f}(T_i) = Q. \tag{1}$$

We then denote the stacked LSTM as $\mathbf{g}$ which transforms the visual representations $Q$ into $H \in \mathbb{R}^{d \times h}$, following

$$\mathbf{g}(Q) = H \tag{2}$$

where $h$ represents the number of LSTM cells in each layer. Finally, to transform LSTM representations into the decision response, we apply two linear transformations $W_1 \in \mathbb{R}^{h \times ht}$ and $W_2 \in \mathbb{R}^{ht \times m}$, mapping $H$ to $P \in \mathbb{R}^{n \times m}$ where each row of $P$ being a probability distribution over the $m$ classes that the original experiment was trained to classify,

$$(H_i W_1) W_2 = P_i. \tag{3}$$

In our simulations, we employ 270 images in each trial ($n = 270$), extract 1000 high-level features from VGG-16 to represent each stimulus image ($d = 1000$), feed these features into 6-layer stacked LSTM with 1000 cells ($h = 1000$) and adopt linear transforms of 256 units (ht = 256) and 3 units (L, R, and N, $m = 3$). The number of layers stacked in LSTM is a hyperparameter. We adopt 6-layer stacked for the reason that we expect the number of 6 could give us a good chance to match brain regions and better examine whether each LSTM layer can learn representations at different levels of abstraction.

## Training and model evaluation
In training our deep neural network, a common cross-entropy loss function is used for optimization, estimated using the maximum likelihood method:

$$\mathcal{L} = -\sum_{i=1}^{N} y_i \log(p_i) \tag{4}$$

where label $y_0 = 1$ if monkey had a leftward saccade and $y_1 = 1$ if monkey had a rightward saccade else $y_2 = 1$. The five stimuli on the boundary corresponded to $y_2 = 1$ regardless of the cue shape. The Adam optimizer is used with the default learning rate of $1e-5$. We use a batch size of 1 (i.e., trial-by-trial learning) to match the training procedure of the monkeys. All the simulations are implemented using PyTorch[48].

To evaluate the categorization accuracy of our model, we use micro-averaged F1 score metric. The micro-averaged F1 score pools per-sample classifications across classes to compute the micro-averaged precision (P) and recall (R) by counting the total true positives, false negatives, and false positives, and then a harmonic class of P and R, as follows

$$F_1 = 2 \frac{P \times R}{P + R} \tag{5}$$

ranges between 0 and 1. The closer it is to 1, the better the model. Our model achieves the F1 score of 98% at least in multiple sampling frames (Supplementary Fig. S3).

## Representational similarity analysis
In order to relate our artificial deep neural network to the brain, we utilize RSA, an experimental and analytical framework, relating representations in the brain and our model by computing and comparing RDMs that characterize the information encoded in a given brain or model. Each 16 × 16 RDM in this study contains distances/dissimilarities for pairwise stimuli. The distance indicates how dissimilar those two stimuli are in the monkey brain or how dissimilar those two model representations are in the model.

## Candidate neural dissimilarity measures
There are many possible measures to calculate the distance/dissimilarity between two or more task-relevant and stimuli-relevant spike trains in neural data. In this study, we consider candidate measures, such as rate coding, inter-spike interval (ISI) distance and SPIKE distance. Rate coding considers spike count within a time bin and represents each stimulus as a $n_r$-dimensional neural firing rate vector where $n_r$ is the number of recording sites whereas ISI and SPIKE distance take spike timing into account. ISI emphasizes the relative intervals between spikes, while SPIKE also incorporates the absolute timing of spikes, which can be useful for evaluating synchrony between spike trains. Pairwise-stimuli ISI/SPIKE distance is calculated by temporal averaging over a period of trial time. Specifically, rate coding explicitly constructs a representation vector for individual stimulus before constructing the RDM using Euclidean distance, cosine distance or Pearson correlation, and the other two measures construct pairwise

statistics towards RDM directly. More detailed information of candidate neural dissimilarity measures, please see Supplementary Information.

When processing neural data, we extract the spike trains of isolated neurons detected under electrodes. Regardless of which candidate measure we use, each element in task-relevant RDMs is the averaged distance (dissimilarity) from spike trains (in the time bins) across task-relevant trials and monkeys using a Python toolbox, PySpike[49], implemented ISI-distance and SPIKE-distance for the numerical analysis of spike train dissimilarity.

## Measure evaluation
To identify the best neural dissimilarity measure to analyze neural data, we opt for an RSA-based approach where we correlate RDMs derived from candidate neural dissimilarity measures to a static reference matrix of stimuli. This reference matrix reflects experimenter's intended relationship between stimuli and we assume that the best measure should be the one that yields the highest Spearman's rank correlation to the reference matrix. To construct such a reference matrix $E$, we represent each stimulus using experimenter-indented coordinates and compute pairwise dissimilarities. 16 color-motion stimuli are distributed in four quadrants of a coordinate system, where each stimulus has a unique x-y coordinate. For example, bottom-left item in the coordinate system is denoted as $(1, 1)$ and the top-right item is denoted as $(4, 4)$ in Fig. 1c. For any two stimuli $a$ and $b$ with coordinates $(a_x, a_y)$ and $(b_x, b_y)$, we compute their distance/dissimilarity $E(a, b)$ as

$$E(a, b) = 1 - e^{-\sqrt{|a_x - b_x|^2 + |a_y - b_y|^2}}. \quad (6)$$

We derive such an experimenters-intended-based RDM, $E$, by calculating the exponential distance of the coordinates of pairwise color-motion stimuli. Note that being subtracted by 1 and the exponential function have no effect on the calculation of Spearman rank correlation. Although experimenter-intended-based RDM provides a fixed reference in both motion and color contexts, it inherently incorporated task-relevant distinctions in how it organized information based on stimuli coordinates.

## RDM based on neural data
In the original experimental setup[21], as the monkey data was recorded over a long duration, electrode replacements occurred, leading to variations in the number of electrodes across trials. Each session consisted of multiple trials (1000–1900 trials per session, 53 sessions in total), with electrodes repositioned and microdrive assemblies removed after each session. We excluded electrodes with missing recordings (e.g., empty units) and abnormal values (e.g., negative values) in the raw data. We apply RSA by constructing RDMs from random pairwise trials within a session, as only one stimulus was presented per trial. We compute pairwise trial correlations and average them within sessions, and report confidence intervals across sessions (e.g., Fig. 2). For rate coding, each stimulus was represented as a high-dimensional vector derived from neural activity recorded during that trial. To ensure consistency across trials and brain regions, we randomly selected 30 recording sites per region, maintaining uniformity across all stimuli.

## RDM based on model representations
The LSTM-based representational dissimilarity matrix (RDM) is constructed purely from the output of the LSTM cell. After the CNN-LSTM model processes a trial (i.e., a stimulus), it generates a stimulus representation as a vector with 1000 entries for each layer, as the LSTM cell size was set to $h = 1000$. During training trials, the layer parameters are updated, but they are fixed in test trials. Since each stimulus can be represented as a 1000-dimensional vector consisting

of model representations, we are able to compute the dissimilarity between these stimulus representations using Euclidean distance (Fig. 3) and Cosine distance (Supplementary Fig. S18). Specifically, we computed and averaged the Euclidean distance and Cosine distance for every pair of stimuli, forming a $16 \times 16$ dissimilarity matrix. In total, we derived 6 RDMs, one for each of the 6 individual LSTM layers. The entries in each RDM reflect the dissimilarity between the LSTM representations.

## LSTM time estimated
Given that the LSTM processes stimulus images sequentially, it encompasses hidden states $H$ that dynamically evolve within a trial (i.e., over the sequence of stimulus images in the trial). By capturing the hidden states from processing the first image (fixation) to the last image (moving colored stimulus), we can derive $n_m$ task-relevant RDMs within a trial, where $n_m$ is the number of stimulus images within a trial. To align with sliding windows, we estimate LSTM time based on the frames of images. For instance, if selecting 60 frames per second, each image is displayed for 1/60 of a second, allowing the input of each image to be considered as a timestamp. LSTM time at 0 is similar to the 0–0.05 s time window, encompassing the information of 0.05s/(1s/60images) = 3 input images. Note that with each time window (wide of 0.05 s) sliding by 0.01 s, we assume each sliding step of the window corresponds to one stimulus image. For the long time window (0–0.25 s), dissimilarity for LSTM (Fig. 3c) is estimated as the averaged information of 0.25 s/(1s/60images) = 15 input images.

## Dimensional stretching
In two categorization tasks, dimensional stretching refers to the expansion of the perceived psychological difference beyond the physical difference between stimuli. In our study, attentional effects refer to how the monkeys allocate attention to relevant dimension in response to their saccade response in task-relevant context. We evaluate dimensional stretching through model-free and model-fit approaches to gain insight into how the monkey brain reconfigured along motion-relevant and color-relevant dimensions in task-relevant context.

## Dimensional stretching: a model-free approach
The model-free approach evaluates dimensional stretching without assuming an underlying psychological space. Simply, we compare the dissimilarity between item pairs mismatch on one dimension and match on the other under two task contexts.

We denote the dissimilarity between item pairs $(i, j)$ as $D_m(i, j)$ and $D_c(i, j)$ that mismatch on motion and mismatch on color, respectively, for instance, in Fig. 1c, stimuli in the bottom-left corner (pink and upward) and stimuli in the top-left corner (green and upward) mismatch on color and match on motion. For $N(i, j) = 24$ pairs under motion-relevant context (i.e., $(i, j)^m$) and color-relevant context (i.e., $(i, j)^c$), we perform two-tailed paired t-test to show the difference of pairs that in different tasks, e.g., $D_c(i, j)^c - D_c(i, j)^m$ and $D_m(i, j)^m - D_m(i, j)^c$. To simplify the notation of the 24 mismatch pairs, we use a triangle to indicate the motion task and a square to indicate the color task. Blue and orange colors denote pairs mismatching on the motion dimension and color dimension, respectively, as shown in Fig. 3.

## Dimensional stretching: a cognitive model
The model-fit approach aims to explicitly model the psychological space itself, which assumes attentional effects exist and finds the best-fit attention for task-relevant context. We use a cognitive model that hypothesizes attention weights towards task-relevant and task-irrelevant dimension, then derive an attentional RDM, $AE$. Specifically, we assign the stimulus coordinates and for pairwise stimuli item $a$ and $b$, the distance/dissimilarity in the task content, for example, the

horizontal dimension is the task-relevant dimension, is calculated as

$$AE(a, b) = 1 - e^{-c*\sqrt{(w_r*|a_x - b_x|^2 + (1-w_r)*|a_y - b_y|^2)}} \quad (7)$$

where $c$ is constant, $w_r$ represents the attention weight in the dimension relevant to the task and $w_r \geq 0$. We find best-fit parameters $w_r$ to the task-relevant dimension by maximizing the Spearman rank correlation between attentional matrix $AE$ and representation-based RDM with the grid search,

$$w_r = \max \text{Spearman}(AE, RDM) \quad (8)$$

where all stimuli items are fit at once to provide one estimate of stretching along the dimension of color and motion. Based on the cognitive model, we can explore how attention weights change depending on the task cued.

### K-fold cross-validation procedure
To infer the relative attention of motion and color information in the task context, we optimize the cognitive model parameters to best match the observed stimulus coordinates. In this work, we employed the standard k-fold cross-validation approach. In this approach, the dataset is randomly divided into k equally sized partitions. The model is trained on $k-1$ folds and evaluated on the remaining fold. This process is repeated with k iterations, each iteration with a different fold used for evaluation. For our study, we use a 5-fold cross-validation procedure. To ensure that each subject's data is appropriately accounted for, this procedure is performed separately within each task context (motion and color) for each subject. Thus, this analysis results in a total of 20 iterations across the two task contexts and two subjects.

### Baseline model
The baseline model is identical to the cognitive model but is trained using an equal number of folds from each task context. Specifically, the baseline model is fitted using two folds from the color task and two folds from the motion task, as opposed to the four folds from the relevant task context in the cognitive model. This baseline model is also trained following the 5-fold cross-validation procedure, providing a direct comparison to determine how well the cognitive model captured neural activity patterns using the 20 held-out validation folds.

### Quantification and statistical analysis
In general, we assume that data were normally distributed, but this is not formally tested. We conduct a one-way ANOVA analysis to validate the significance of the type of measure, and employ a two-tailed paired t-test with Bonferroni correction for pairwise comparisons to demonstrate that ISI outperforms other measures. In order to evaluate adaptive stretching, we utilize a two-way repeated ANOVA with fixed factors of task (color or motion relevant) and mismatching dimension (color or motion mismatch) and a random effect for the monkey subjects. For the LSTM model, we utilize a three-way repeated ANOVA with fixed factors of task (color or motion relevant), mismatching dimension (color or motion mismatch), and layer number (1–6), with a random effect for simulations. To evaluate the stretching across dimensions in a quantitative manner, we conduct cognitive model-based analyses using 5-fold cross-validation per monkey and per task. We evaluate the RSA-based correlation score by a one-way repeated ANOVA with a fixed factor of model (cognitive or baseline model) and a random effect for the monkey subjects. Similarly, for the LSTM model, we perform five-fold cross-validation during the training process. The RSA-based correlation score was evaluated using a three-way repeated ANOVA with fixed factors of model (cognitive or baseline), training step (1–10), and layer number (1–6), with a random effect for simulations. All statistical tests are described in the main text. Levels of statistical significance are indicated as follows: $^*p < 0.05$, $^{**}p < 0.01$, $^{***}p < 0.001$, $^{****}p < 0.0001$.

### Reporting summary
Further information on research design is available in the Nature Portfolio Reporting Summary linked to this article.

## Data availability
Source data are provided with this paper.

## Code availability
The codes used in this work are available at https://github.com/xinyacheung/neural_similarity.

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

## Acknowledgements

This work was supported by ESRC (ES/W007347/1), Wellcome Trust Investigator Award WT106931MA, Royal Society Wolfson Fellowship 183029, and Foresight AI Safety grant to B.C.L., JPB Foundation, Picower Institute for Learning and Memory, and Office of Naval Research N00014-23-1-2768 to E.K.M., and China Scholarship Council (Grant No. 202206260103) to X.Y.Z.

## Author contributions

S.B.-S. and B.C.L. designed and proposed the research direction. X.-Y.Z., S.B.-S., and M.L. performed the experiments and simulations. X.-Y.Z., S.B.-S., and X.L. analyzed data. S.L.B., M.S., and E.K.M. contributed dataset. All authors analyzed results; X.-Y.Z., X.L., and B.C.L. wrote the paper.

## Competing interests

The authors declare no competing interests.
