## [Transparent Peer Review file · Nature Communications]

Adaptive stretching of representations across brain regions and deep learning model layers

Corresponding Author: Professor Bradley Love

Version 0:

Reviewer comments:

Reviewer #1

(Remarks to the Author)

This study reanalyzes an old awake animal neurophysiology data set to determine whether attention stretches perceptual space in several different brain areas. It then investigates the same attentional effect in a hybrid CNN/LSTM trained on the same task. The authors claim that several high-level brain areas “reconfigure” to stretch perceptual space, and that spike timing is important for this effect. Unfortunately the paper in its current form is very difficult to read. As it is written currently the paper reads as if it were a short conference paper written quickly at the last minute before a deadline. Nature Communications is an archival journal, and the paper should be revised accordingly.

1. The paper is, frankly, quite difficult to read. I could provide many examples of this problem from the text, but for simplicity just consider the first few sentences of the results: “One initial question is which measure of neural similarity is most aligned with the neural recordings [30]. To answer this question, we assessed various measures of neural similarity to determine which one maximized representational similarity [31] with the experimenter-defined stimulus coordinates of color and motion.” Why is this the first result? Is this some preliminary result that has to be established before the main results? Is it a result that is required to avoid a potential confound? And why is a representational similarity approach being used to assess attentional stretching at all? None of this information is provided. Instead, the reader is tossed into the maw of the paper without a rope. And it doesn't get any better in the rest of the results. The authors should not expect the reader to be willing to wade through the entrails of a paper to figure out which questions are being asked and why. They should not force readers to go look at other papers just to gain some vague idea about what the acronyms SPIKE and ISI actually measure. I could go on here but I think that I have made the point: the authors really need to go back and rewrite this paper with an eye toward conveying information clearly to the reader.

2. The paper really needs to be reorganized to make the logic of the data analysis procedures clear. It is conventional in neuroscience papers to present the neurophysiology results first, and then to present the computational model (if one is proposed). The current paper does not follow this format, instead blithely intermixing neurophysiology results with analysis of the CNN/LSTM, as if those are all the same thing and so directly comparable. They are not, and intermixing them just obscures the meaning of the results.

3. The rate versus timing issue should also be broken out into a separate and clear analysis. This is a classic question of enduring interest in neurophysiology. But the results presented here are so compressed and opaque that it is difficult to compare the two sorts of codes in the current paper.

4. Given the convoluted, unclear logic of the data analysis procedures, and the very sparse treatment of the methods and the results, I confess that I am not sure what I can really conclude from the results presented here. I think that the revision should expand the methods and the results so that things will be more comprehensible.

5 In the introduction and middle of the results the authors talk of attention stretching the perceptual space, which is fairly clear given prior theories and results in the field. But the latter part of the results focuses on the question of which brain regions (and the LSTM) reconfigures itself. This concept is much more woolly, at least in the neurophysiology literature. What precisely is meant by this? What kind of neurophysiology data would support this or not? How does this relate to stretching/warping? It is very unclear as presented in the current manuscript.

6. I can that the authors should cite their own prior work on attentional stretching (e.g., Mack et al 2016; Braunlich & Love 2019). But they ignore prior neurophysiology and fMRI work on precisely this issue that predates any of the experimental studies cited here. For example, work by Connor (1996), Gallant, (2008) and Treue (2010) showed these effects in V4 and MT. And Gallant (2013) used fMRI to show attentional warping across much of the brain. This prior literature should be described and cited here, to do otherwise would be poor scholarship.

7. Furthermore, the two neurophysiology studies showed stretching of the spatial or feature-based receptive fields of single V4 neurons, a result that seems at odds with the result reported here. In fact the Treue lab has had a whole series of papers that have shown this effect in MT. This difference seems likely either due to differences in the tasks used in the various studies, or to differences in the way that the data were analyzed and modeled. (If I have read the paper correctly, I suspect this is because attentional stretching/warping here was only assessed across dimensions, whereas prior studies on attentional stretching/warping also looked at changes in tuning within dimensions.) This needs to be addressed thoroughly and convincingly. If the cause of the difference in results cannot be determined definitively, the various options should at least be discussed and described.

8. I confess that I'm a bit confused about how the CNN/LSTM component of this study contributes anything. The CNN/LSTM architecture is quite different from the architecture of the brain, and the training regime is also quite different between the two systems.

9. The CNN/LSTM architecture used here is not standard. The authors write, "The CNN front-end is used to simulate the monkey visual system and the stacked LSTM is to simulate multiple brain regions such as MT, V4 which might be involved in the learning task." This is a potentially interesting but very unusual architecture. And it isn't very clear precisely why this was chosen, or why it is needed. The entire architecture is poorly motivated and poorly justified.

10. In most other studies using ANNs to model the visual system, the CNN is used to model the complete hierarchy. Given this it is a bit disappointing that the authors did not evaluate a pure CNN-based architecture, and that they did not try to evaluate precisely how the CNN and LSTM components of their network contribute to the claimed results. This really needs to be done, and given the ease of fitting ANNs these days there is really no reason not to give this a more systematic treatment.

11. The authors use RSA as a core method for their analysis. Justification for this choice of analysis should be provided. Why was this method chosen? What are the advantages and disadvantages relative to methods used in other papers (most commonly GLM-based methods)? Furthermore, some information should be provided about the statistical assumptions underlying RSA, and whether they are met here. No such information is provided in the current manuscript.

12. The procedures used to derive RDMs for the LSTM model are quite complicated, and it appears that many specific decisions were made that are not clearly justified based on first principles or prior art. How can we trust these results or their interpretation?

13 The model-free analysis of dimensional stretching is not, in fact, model free. Any statistical test has an underlying statistical model, and an RDM-based analysis certainly does. Rather than stating that this is a model-free approach, the authors should either state the underlying model and show that the data fit the assumptions of that model, or they should use a simpler approach (based on a GLM or some other straightforward method), as others have done.

14. The authors use ANOVA, t-tests, and permutation tests. Permutation tests should be used in all cases, just to provide some consistency in the statistical methods, and to maximize robustness in these classical significance-based statistical tests.

(Remarks on code availability)

Reviewer #2

(Remarks to the Author)

In the manuscript, the authors tested a hypothesis regarding the reconfiguration of visual representations along task-relevant dimensions using a previously published monkey dataset collected during color- and motion-related decision-making tasks. They also investigated whether the same reconfiguration was observed in an artificial neural network model (CNN_LSTM model) trained with the same inputs and rewards as the monkeys. The authors initially conducted a representational similarity analysis (RSA) to identify the best neural similarity measure from various versions of the rate coding measure and spike timing measures using the experimenter-intended coordinate, which assumes a fixed reference in both color and motion conditions. Subsequently, the authors employed inter-spike intervals to test their dimensional stretching hypothesis by examining whether neural distances between experimental conditions were modulated in task-dependent manners. The authors demonstrated modulations on neural distances between conditions relevant to the tasks in both the monkey brain and the deep learning model. They also applied a cognitive model that can take attentional effects on similarity matrices into consideration to illustrate task-dependent weight modulations for both the monkey brain and the model. Based on these results, the authors claimed that both the monkey brain and deep learning model exhibited response modulations depending on task relevance, indicating that stretching is an adaptive strategy for both biological and artificial agents.

The correspondence between the brain and artificial neural networks (ANN) beyond sensory cortical areas is a hot topic in the fields of neuroscience and computer science, attracting the interest of researchers with a broad background. While task-dependent attentional modulations on neural representations in broad brain areas were not entirely new (e.g., Çukur et al., Nat. Neurosci., 2013, for human data), demonstrations with color- and motion-relevant tasks with the monkey brain, along with the attempt to explore a similar mechanism in ANN, would provide additional insights into this topic. However, despite potential interest in the question and framework, the evidence supporting their claims is not compelling. Various concerns, particularly related to the transparency of results and methods, as well as the validity of statistical tests, need to be addressed. In particular, it was quite difficult to fully understand the details of the methods only from the manuscript to replicate the present study. Please see below for details on these issues.

1. The most crucial issue in this manuscript is the absence of statistical tests conducted to validate the reproducibility of the present findings with independent (or new) data samples. For instance, one of the main results illustrating dimensional stretching in the monkey brain and CNN_LSTM (Figure 3, lines 118-123) was performed to assess significance across variances of elements in RDMs (24 item pairs; combinations of stimulus conditions). With this setup, the authors can confirm that significant differences were observed in the current dataset and can claim that similar results might be observed in additional combinations of stimulus conditions. However, because variances across data samples were not tested, they cannot claim that this finding is generalizable to new data samples (e.g., trials and monkeys), which is crucial for supporting the reproducibility of the findings. Particularly, the observed effect size was quite small ($M = 0.0007$) compared to the baseline dissimilarity (ranging from 0.359 to 0.363) and the 95% confidence intervals (Fig. 2a and b, ~ 0.5). Therefore, it would be critical to examine if similar differences are observed robustly under variances across data samples. Likewise, the authors tested the significance of attentional modulations by comparing the results with null distributions produced by label shuffling (Figure 4), while such usage of statistical tests cannot support the reproducibility of the results. Overall, given the small effect size reported, the statistical tests performed in the present manuscript seem not sufficient to support the reproducibility of the findings. I recommend the authors conduct statistical testing to support the reproducibility of the findings, as the dataset contains rich data samples (e.g., 40 trials with different random dots' initial positions). The same analysis can be performed to test the CNN_LSTM model analysis.

2. Regarding the validity of the neural measure selection, the authors claimed that "Spike timing was crucial to this code" (Abstract) based on the results showing relatively high Spearman correlations by ISI and SPIKE in both the motion and color tasks than those based on rate coding (Figure 2). However, I was confused about the results in Fig. 2 at several points. For example, I wonder why the Spearman correlation was moderately high for ISI and SPIKE from the early timing after stimulus onset, while relatively late (at least ~ 50 -100 ms) rises of stimulus-related responses were reported in the original study of the dataset (Siegel et al., 2015). The 50 ms time window may explain this early rise of correlations, but the same trend was observed even with a shorter time window (20 ms, Extended Data Fig. 2). Is there any possibility that these high correlations do not actually reflect differences in neural responses to different experimental stimuli (modeled by experimenter-intended coordinates)? To make this point clearer, I recommend that the authors show the time course of the Spearman correlation for the whole trial period (from fixation to response) to confirm that the observed correlations truly reflected stimulus-related representations, which may provide positive control of the validity of the analysis. If high correlations were observed even before the stimulus onset, the rationale for selecting the spike timing measure for the main analysis might be violated, leading to the failure to support the authors' claim about the importance of spike timing as stated in the abstract.

3. Related to the results in Figure 2 and comment 2, I wonder why the results by the rate coding did not exhibit positive correlations at all. As the authors mentioned in the Discussion (lines 168-170), the activity of the LSTM is akin to rate-based coding. If the LSTM model showed positive correlations to the experimenter intended coordinates (actually it was not shown in the manuscript), the rate coding results should also exhibit positive correlations in this analysis. Therefore, I recommend the authors elaborate on this apparent contradiction and show the Spearman correlation between the experimenter intended coordinates and representational dissimilarity matrices (RDMs) derived from the LSTM model. Furthermore, because the manuscript lacks the details of how RDMs were computed from the LSTM model in the main analysis (e.g., Fig 3; although the authors explicitly described that they used cosine similarity in Extended Data Fig. 5), I suggest the authors provide additional details to allow readers understand and replicate the methods in the main analysis.

4. Related to comment 3, I suspect that the difference between the two spike measures and the rate coding measure might be due to the differences in the computation procedures for these measures. In the Supplemental Information, the authors described that "To construct rate-coding-based RDM, we first represent each stimulus as a 1D vector whose entries are the number of spikes across neurons/electrodes." This procedure was totally different from the procedure for constructing RDMs for spiking measures described in lines 277-281: "When processing neural data, we extract the spike trains of isolated neurons detected under electrodes. Regardless of which candidate measure we use, each element in task-relevant RDMs is the averaged distance (dissimilarity) from spike trains (in the time bins) across task-relevant trials and monkeys...". To conduct a fair comparison between the rate-coding measure and spike timing measures, it would be better to use a consistent method for constructing RDMs. Thus, I suggest the authors perform the same analysis in Fig. 2 using RDMs constructed by a similar procedure with the spike timing measure (e.g., first computing RDMs for individual trials/monkeys and then averaging them).

5. Related to the analysis in Fig. 2 and the validity of the experimenter intended coordinates, I actually was unable to understand the intention the authors have for the reference matrix. As mentioned by the authors, the matrix assumes a fixed reference in both color and motion conditions and then cannot suited for aligning neural/model RDMs, which can have representations biased to specific tasks (e.g., V4 or MT). An assumption of the matrix is that a single step difference along the color axis is equivalent to any difference along the color and motion axes (e.g., the difference between orange to pink is

equivalent to -90 to -30). As this assumption is not intuitive, it would be helpful to add more explanations about this point for readers to better understand the authors' intentions.

6. On lines 118-121 and in Fig. 3a and b, the manuscript lacks information on which brain area was used for the analysis. Please provide the details of this point. Furthermore, to support the claim that "the brain radically reconfigured itself to stretch representations along task-relevant dimensions in lateral PFC, frontal eye fields (FEF), lateral intraparietal cortex (LIP), and inferotemporal cortex (IT)" in the abstract, provide the results for individual brain areas with appropriate statistical tests as I suggest in comment 1.

7. In Fig. 3d, I was unable to understand why all dissimilarity values start from 0 at the 0.00 LSTM time. In my understanding, the dissimilarity does not necessarily go through 0 at this point. I recommend the authors show the time course during the entire trial for the LSTM model to enhance the clarity of the entire data. If the authors performed any normalization to set the dissimilarity at the onset to be 0, please also provide detailed information about the operation and elaborate on the rationale.

8. Regarding the statement in lines 64-65: "one possibility is that these control systems themselves can be learned by nonhuman animals and ANNs, enabling them to reconfigure themselves in response to task cues", it would be helpful if the authors could show the increasing effect of dimensional stretching through the training/learning of the LSTM model. This is a demonstration often reported in studies (Yamins et al., 2014; Kell et al., 2018).

9. In Fig. 4, the authors showed weights fitted to the data from individual brain areas and I agree that comparing weights estimated from the data under different tasks is a good demonstration of the task-specific reconfigurations. However, to make a stronger demonstration of the claim, I suggest the authors evaluate how this fitting enhances the Spearman correlations between RDMs derived from brain data and RDMs of the baseline reference (i.e., experimenter-intended coordinates). Furthermore, as I suggest in comment 1, the statistical test performed with null distributions generated by label shuffling does not support the reproducibility of the observed effect on new data samples. To support the reproducibility, the authors may want to fit weights using a subset of data and test using independent test data samples to examine if the weighted model show higher correlations than the original reference or the opposite model (e.g., using the color-task-fitted model for evaluation of motion task data fit) as the authors performed similarly in the CNN_LSTM model analysis.

10. In Fig. 5, the authors compared RDMs of V4/MT and the LSTM model for each of the color and motion tasks to claim that "LSTM layers freely reconfigure as a function of the task context". However, the observation that "For example, V4 shows a decent correspondence with LSTM layers when color is relevant, whereas MT does when motion is relevant." (lines 142-143) can also occur when the LSTM-RDM was fixed, and V4-/MT-RDMs changed depending on the tasks. For rigorously testing the claim, it would be better to additionally show higher Spearman correlations between motion-task-fitted brain-derived-RDMs (including MT and other areas) and the motion-data-fitted LSTM-RDM than between motion-task-fitted brain-derived-RDMs and the color-data-fitted LSTM-RDM. Similarly, Spearman correlations between color-task-fitted brain-derived-RDMs and the color-data-fitted LSTM-RDM can be compared with those between color-task-fitted brain-derived-RDMs and the motion-data-fitted LSTM-RDM. If higher correlations were observed from this analysis, it would strengthen the support for the authors' claim.

11. In Fig. 2c and d, asterisks are shown on ISI and SPIKE, but the type of statistical tests conducted was not clear. Please add details about this point.

12. On lines 118-123, in Fig. 3a and c, and Extended Data Tables 1 and 2, it was not clear which timing of data points was used for the analysis and for generating figures and tables (entire 250 ms stimulus period?). Please provide sufficient information to understand the procedures for the data analyses correctly.

13. The authors did not clearly describe how data samples from multiple neurons/electrodes are summarized for producing results of individual brain areas (e.g., Fig. 4a and Extended Data Fig. 3) and lobes (Extended Data Fig. 4). Please provide details on this point.

14. On lines 118-121 and in Fig. 3a and b, the information about which LSTM layer was used for the analysis is missing. Please provide this information.

15. In Fig. 4b, the authors only showed the combined modulation of attention, but to make the results more transparent, I recommend that the authors additionally show the attention weights for both motion and color tasks as in Fig. 4a.

16. Regarding the readability of the entire manuscript, I occasionally encountered difficulty in finding the correspondence between the descriptions in the main text and the figures (including main figures, Extended Data Figs, and Supplemental Figures). For instance, the authors did not refer to Figure 4b and Extended Data Fig. 2 elsewhere. Therefore, I recommend that the authors thoroughly review the manuscript and include appropriate references to all figures. This will help readers easily identify the connections between statements and results throughout the text.

1.Çukur, T., Nishimoto, S., Huth, A. G. & Gallant, J. L. Attention during natural vision warps semantic representation across the human brain. *Nat Neurosci* 16, 763–770 (2013).

2.Yamins, D. L. K. et al. Performance-optimized hierarchical models predict neural responses in higher visual cortex. *Proceedings of the National Academy of Sciences* 111, 8619–8624 (2014).

3.Kell, A. J. E., Yamins, D. L. K., Shook, E. N., Norman-Haignere, S. V. & McDermott, J. H. A Task-Optimized Neural

Network Replicates Human Auditory Behavior, Predicts Brain Responses, and Reveals a Cortical Processing Hierarchy. *Neuron* 98, 630-644.e16 (2018).

(Remarks on code availability)

The authors provided code for replicating their findings, primarily focusing on the visualization of the main results (demo code) with accompanying instructions on how to run the program. Additionally, they included code for the neural data preprocessing and LSTM model training but with fewer comments. The code for visualizing the main results only covers half of the results and does not include the analysis for presenting results from the CNN_LSTM model. In my view, considering the comparable importance of the CNN_LSTM section with the brain data section, it would be beneficial to include code for the CNN_LSTM section as well. While the requirements for running the code are provided, there are no instructions on how to install and set up the environment. Consequently, it may be challenging for novices to execute the entire code, except for the visualization part.

Reviewer #3

(Remarks to the Author)

Review for Adaptive stretching of representations across brain regions and deep learning model layers

I have now had time for an in-depth review of paper "Adaptive stretching of representations across brain regions and deep learning model layers". The paper presents a reanalysis of an existing dataset to answer a timely and interesting question: how does the brain dynamically adapt to the ever-changing demands as it goes from one task to another? To answer this question, the authors contrast brain responses in two different task settings based on spike timing information, as well as by training an LSTM model. They find that many brain areas can adapt their coding to the task - with the exception of V4 and MT, which both show attentional affects limited to their original coding preference (here colour and motion). The paper is well written and provides an important first datapoint that will spark further work in a direction that I am excited about. Yet, I must admit that the rather short submission made the impression as if the few analyses provided represent a starting point rather than a full-fledged paper. Below, I suggest a few plots/analyses that I think would complement the work and thereby allow the reader to better understand the underlying mechanisms/coding changes. I hope that the authors will find my comments useful and do take my suggestions as coming from a place of interest rather than unwarranted criticism.

Signed
Tim Kietzmann

Major:

1. Self-limitation/reliance on ISI:

- I find it surprising that, if representational stretching happens throughout the brain, it would not be evident in the rate codes. Would the claim be that the rate code is virtually untouched while all attentional differences are to be found predominantly in timing? This feels unaligned with common understanding of how attention affects rates and hence requires further analyses (see below).

- Second to the previous point, ISI was selected as it best mapped (via RSA) to the experimental design. On these grounds, only ISI is used to test for effects of dimensional stretching from thereon. Yet, changes across tasks could also have been tested for with the other measures to drive home the point that in fact, nothing else changes. In fact, I am unsure whether the experimenter-intended-based RDM (distances in design space) reliably map onto actual coding dimensions in the various brain regions. If this were not the case, then relying on a measurement that best aligns the data and model RDM will be misguided. The model-free approach can easily be applied to all distance measures, including spike timing and rate code based measures. For us to understand how the brain reconfigures itself, we cannot afford to not look at other measurements, including rate-based codes.

- Expanding on the above, ISI task stretching is estimated via univariate differences (a t-statistic). Why not look at patterns, too? Pattern differences are very powerful and are widely investigated in visual coding. Reversely, why not report univariate and multivariate differences for rate codes?

In summary, I think that all distance measures need to be tested univariately and multivariately to get a full picture as to what is happening. A reliance on univariate ISI, only based on RDM alignment is, in my eyes, not sufficient to describe the whole picture this exciting data has to offer.

2. Presentation of (ISI) results and elevated baselines:

- Figure 2 does not mention which brain area is shown (same for figure 3). I expect an average across all areas? The supplemental materials show effects for all areas and here, the effects development over time are often smaller than in the whole-brain average show in the main figure. When taking into account that areas MT and V4 are (i) uniquely positioned to code for the experimental dimensions and (ii) show quite different attentional effects to all other areas, then plotting an average across everything becomes questionable.

- More importantly, the results of ISI are, more or less across the board, significantly higher than chance already during stimulus onset. Why is this? How long into the past and future does the effect go? I think the authors need to show a much longer timeframe (from say -200ms to 400ms) and understand what is going on. Why would the experimenter-intended-based RDM already be significantly correlated before stimulus information arrives? If the stimulus sequence is random, this should not be happening and I am worried that this points to an uncontrolled confound.

3. Figure 3 - main result

- The main figure of the paper is Figure 3 (the data used for the graphic remains unknown again, but I am hoping that I did not miss it somehow). This plot focuses on the effect of stretching. In this case, an average across brain regions really misleading, because regions V4 and MT behave so very differently from the others. Think of it as an interaction effect that prohibits an easy analysis of the main effect. At the very least, the plots should be made for all regions separately. Same goes for the timecourse in Figure 3B.

- Please note furthermore that the effect size (e.g. condition differences) are more or less constant across the whole time period, and differences already appear at 0ms after stimulus onset (note that dissimilarity goes up over time, but the relevant difference between the curves are not changing). Is this due to a confound, or an aspect of the experimental design (as the cue was shown earlier)? If this is due to the cue, then the whole time course from the cue onwards should be presented.

4. Cognitive Modelling

- I do not understand the rationale behind using the cognitive model. What does it add other than another layer of complexity into the analysis? Why not plot the actual data across areas and conditions (as in, differences between activations, pattern or univariate, in the contrasted task conditions across time in the various regions)? This should reveal the same patterns and would much more informative.

Minor:

- The LSTM results are shown in "seconds" whereas model time does not adhere to real-world time.

(Remarks on code availability)

Reviewer #4

(Remarks to the Author)

In this paper the authors performed a novel analysis on a previously published data set (Siegel et al., 2015). They found evidence for "stretching" of task-relevant representations. They also simulated learning in the task using an ANN, which they then compared with the behavioral data results. The ANN also exhibited "stretching" as an emergent property.

I thought the reanalysis of the data set was interesting and well done. The authors tested the hypothesis that representations of task relevant features would be "stretched", which they operationalized as greater dissimilarity in activity for mismatched pairs when the mismatched dimension was task relevant. They found that the representations were stretched for both tasks in all the areas they examined except V4 and MT which were specific to color and motion, respectively. I had some concerns detailed below about the theoretical framing of the results and integration with other research in the field, but those should be easily addressable.

Major concerns:

I was not persuaded by the relevance of the model results. The model gives some predicted results that are qualitatively roughly similar to the empirical results, for example in Figure 3. However, the similarity was not very close, and I was left unpersuaded that the model provided significant insights into the mechanisms involved. It is interesting that a model will develop stretching of representations even without that being programmed into the model, and that does seem to parallel the neural results, but that nevertheless seems like a very general similarity to me. Is there any reason to think that such stretching wouldn't be a general feature of many computational models? In that case, what is the importance of the specific modeling approach taken?

I had difficulty understanding the rationale and importance of the results depicted in Figure 5 relating layers of the ANN to the neural regions. The text (lines 139-144) argues that there is no straightforward correspondence between the layers and brain regions, and that the model is more flexible than brain regions and can fully reconfigure. Both of these statements sound to me like reasons that this analysis should be uninterpretable – if the ANN is highly flexible and can reconfigure to adapt to many things, isn't that an argument that it is NOT a good model for what is happening in the neural regions? I don't understand why one would hypothesize alignment (that is correlation) between the layers specifically and the neural regions, and what that shows beyond the previous analyses in Figure 3.

The authors did not discuss the different roles of the areas they studied, with the exception of V4 and MT. I would like to see

a greater integration of these results with the results from the Siegel et al paper and other relevant research. What are the implications of "stretching" of representations? How does the presence of stretching relate to the roles of FEF, LIP, etc in decision making? In particular, area IT has a very different role in decision making than FEF, LIP, and PFC, and the results in the Siegel et al paper show different patterns of activity in IT. The authors note in Figure 5 that IT differed from the other regions in alignment between the model and the neural activity, but they did not further discuss this result or the role of IT.

Minor concerns:

The authors should define "stretching" of representations in a rigorous manner in the Introduction. What exactly is stretching? What does it indicate about representations?

The authors often refer to their results as indicating "radical" reconfiguration of representations. How exactly do they define radical? Why do they think that their results show radical reconfiguration, rather than a more limited degree of reconfiguration – what makes it radical?

The authors should add a precise but brief definition of ISI and SPIKE and how they differ to the main text so that a reader can understand the importance of the result without having to go to the methods or supplementary materials.

Figure 3: It is difficult to remember what the condition names in the legend for parts b and d mean – could more meaningful phrases or abbreviations be used?

Figure 5: The authors use a human anatomical model on which they superimposed the areas from the macaque study. This is problematic for two reasons. First, there were no human subjects in this study, so why use a human model? It is misleading. Second, and more important, areas MT, IT, and V4 are in different locations in the human brain than in the macaque so the figure is inaccurate. V4 for example in humans is much more ventral, as is IT.

(Remarks on code availability)

Version 1:

Reviewer comments:

Reviewer #1

(Remarks to the Author)

The current manuscript is much improved over the original submission. A lot of missing details have been filled in and additional statistical support provided.

I was very happy to see that the second reviewer provided many very specific and detailed suggestions for how the paper could be improved. Many of the concerns of the second reviewer were consistent with mine.

The paper still has some issues and there are some missing pieces and ad hoc methods, but the general conclusions are consistent with what is already known about attention in the brain from prior studies (including especially studies from the primate lab that supplied these data), so I expect that the main results will hold even after further revision.

My suggestions to improve the paper before publication are as follows:

The use of RSA to analyze spike timing is an interesting idea, and it might be a nice addition to the toolbox. However, the use of this novel method imposes a responsibility on the authors to verify the veracity of the method, and to provide some information about the underlying assumptions and the boundary conditions. Given the current statistical crisis in psychology and biology its not a good idea to use analyses that aren't fully vetted. The most appropriate thing to do here would be to prove the analysis, but that might be complicated and slow, and it might require working with a statistician. So in lieu of that a bare minimum would be to show that the method is valid in simulation. I suggest that the authors do these simulations and report them in the supplementary materials, as such an effort would increase confidence in the results.

It would be extremely helpful if the "cognitive model" were explained better in the main text. The term "cognitive model" is extremely general and its therefore a bit unclear on its face. I had to dig through the methods to figure out what was going on. The paper would be easier to read if some of the information given in the methods was moved to (or repeated in) the results.

I suggest that the authors try to improve the figure captions further. Some of them (e.g., Figure 4a) are difficult to understand as written.

Other more specific issues:

Abstract: "Despite lacking an explicit selective attention or other control mechanism, the model displayed task-relevant stretching as a consequence of error minimization, indicating that stretching is an adaptive strategy." Error minimization is always associated with some cost function. I think that in the context of ANN training, the cost function is analogous to some

specific state of a selective attention or control mechanism. Given that, this statement should be revised for clarity.

Lines 48-49: "The ideal study to evaluate whether radical reconfiguration occurs would record from multiple brain sites while cueing the relevant dimension for a categorization decision on a trial-by-trial basis." Many people (including this reviewer) might argue that the ideal study would parametrically vary both the stimulus and attention in order to examine tuning curves of neurons under different states of attention. This study doesn't do that, so in that sense it is not ideal. However, this is just a quibble: I understand that all studies are limited in some way, there is no perfect study.

Lines 65-67: "While models of goal-directed attention contain dedicated control systems that selectively weight relevant dimensions [7, 9, 8, 23, 24], one possibility is that these control systems themselves can be learned by non human animals and ANNs, enabling them to reconfigure themselves in response to task cues." It is unclear precisely what this means in terms of the conventional language of control systems theory. One possibility is that this is just a (mis)wording issue, but it should be revised for clarity.

Line 89-91: "Observing stretching in the ANN would provide a formal account of how control and top-down attentional mechanisms can arise from simply maximizing task performance, as opposed to relying on preordained mechanisms." This would not be a "formal account" in any way that an engineer or a statistician would use that word, especially if the LSTM is not interpreted but is taken at face value. This language should be revised or toned down.

Lines 127-129: "Stretching was assessed by considering item pairs that mismatched on one dimension and matched on the other. For example, we predict items mismatching on color and matching on motion should be more dissimilar when color is relevant than when motion is relevant." I thank the authors for clarifying the analyses, this brief summary lays out the main strategy of the analysis clearly.

Lines 153-155: "By utilizing representational similarity analysis (RSA), we were able to optimize the cognitive model parameters to best match observed patterns of neural activity (see Methods), thereby inferring the relative attention of motion and color information in the task context." Please clarify the steps taken to ensure that this analysis was not overfit. It seems that the description of the cognitive model is given in the methods, while the description of the cross-validation procedure is in the main results? It would be helpful to make sure that the cross-validation methods for this analysis (and all other analyses) are really clear in the methods.

Nature Portfolio Reporting Summary: The data analysis portion refers to EEG, which is not included in this paper. I suggest that the authors look over the reporting summary closely to see if there are other errors or omissions.

(Remarks on code availability)

I am not as concerned about the code as I am about the ad hoc RSA analysis of spike timing. RSA is a common procedure, but the underlying assumptions and boundary conditions of the method have been poorly validated. In the original submission I asked for references to support the method, but I know of none so I'm not surprised that those references weren't provided. In lieu of that, I suggest that the authors undertake some simulations just so that we can be assured that their new methods are likely to deliver the claimed results without undue bias. I don't think that this is a big deal, though it will require a bit of work.

Reviewer #2

(Remarks to the Author)

The authors have made efforts to address the issues raised in my previous review, and some of these revisions have successfully resolved the concerns. However, there remain several aspects of the revised manuscript that are still unclear, and new questions have emerged based on the changes in the results. These points require further clarification. Additionally, while I agree that some of Reviewer 1's criticisms were unnecessarily harsh and ungrounded, I share the overall concerns regarding the quality and readability of the manuscript. Specifically, the data used for the analysis (e.g., figures and statistical tests) is often unclear, with insufficient information provided for understanding the analytical conditions (e.g., time windows, brain regions, task contexts). Furthermore, although many results have been added to the supplementary materials in the revised manuscript, they appear to be presented in a rather disjointed manner, without sufficient explanation of how each result supports the main claims of the study. These issues seriously hinder readers from fully understanding the content and implications of the study, as well as from replicating it. Thus, the current version still presents significant issues in terms of clarity, which detracts from the overall quality of the work. As it stands, the current manuscript does not meet the standard expected for publication in Nature Communications.

I acknowledge the potential value of the scientific contributions in this work; however, further revisions are necessary to improve both the readability and presentation, ensuring the content is accessible to a broader readership.

1. The clarification of the results in Fig. 2 through the correction of the error in the rate coding analysis is commendable. Additionally, showing results over a longer time period in Fig. S4 is very helpful, as it provides a clearer understanding of the overall pattern. Given that this approach makes the data more comprehensible, it may be beneficial to present similar extended time period results for Fig. 2, Fig. 3, and Fig. S6-S11 as well.

However, these results raise a new question. Specifically, in Fig. S4, the peak performance for all neural measures occurs around the 250-500 ms period. Despite this, the authors have conducted many analyses (e.g., region-wise analysis in Tables 3-11) using data from only the first 50 ms post-stimulus onset or have shown results only up to around 250 ms (e.g.,

Figs. 2 and 3), without providing a clear explanation for this choice. Given that relatively high correlations are also observed in the rate coding results around the peak period, it is possible that stretching effects might be detected if analyses are conducted over this time frame. This could call into question the validity of the claim made in the abstract that "spike timing was crucial to this code" (indeed, Fig. S9 seems to show).

Therefore, it is recommended that the authors provide a rationale for the choice of time windows used in their analyses and discuss whether the claims made in the manuscript, including the assertion that spike timing is crucial, hold independently of the potentially arbitrary selection of time periods for analysis.

2. In Fig. 2, the rate coding results have been improved and appear reasonably smoothed compared to the spike timing measures. However, the region-wise and lobe-wise rate coding results in Figs. S6 and S7 still show considerable variability, which appears unreasonable compared to the results in Fig. 2. As these figures only display results for the 0-0.25s time window, similar to the earlier version of Fig. 2, it may be beneficial to extend the range, including time before 0s and a longer time period, as in the current Fig. 2. This could aid in debugging and help verify the validity of the results.

3. In the analysis using the cognitive model, while the authors modified the analytical procedure from the original manuscript, these changes were not reflected in the corresponding methods section. Specifically, there is no explanation of the baseline model, which was newly introduced in this revised version. To ensure that readers fully understand the rationale behind the methodology, it is recommended that the methods section be appropriately updated to include detailed explanations of the changes, including the newly added baseline model.

4. Regarding the results in Fig. 4, the authors claimed that adaptive task-relevant stretching was observed at all recording sites, including MT and V4, in the revised manuscript. However, in Fig. 4a, it appears that MT and V4, as well as IT, did not exhibit weight changes according to the task context. While the authors argued that the higher correlation of the cognitive model compared to the baseline model in Fig. 4b indicates adaptive modulation depending on the task context, this alone does not fully support their claim. In particular, the baseline model for V4 showed correlations close to chance level (near zero), raising questions about whether the fitting is working properly and suggesting potential issues with the validity of the method. It is recommended that the authors provide an appropriate discussion on these subtle differences in the results and revise their claims to better reflect the observed data.

5. On Page 5, lines 112-114, as well as in Fig. 2, Fig. S5, and Tables S3-S11, the authors presented the results of statistical tests. However, it is unclear how data from the two different task contexts (motion and color) were utilized in the analysis. Additionally, the exact stimulus timing used in the analysis is not clearly specified in the main text (lines 112-114; is it a 50 ms time bin beginning at 0 seconds?). Please provide sufficient information to clarify these points and ensure the procedures are fully understood.

6. In Fig. S4, the authors used the term "session," but its definition is not provided, making the meaning unclear. For example, does "session" refer to the same unit as a "trial," or is it a separate unit used in the monkey experiments that encompasses multiple trials? Please clarify the definition of "session."

If "session" differs from "trial," it may be more appropriate to redraw Fig. S4 based on variances across trials. This would better support the significance and reproducibility of the results.

Additionally, in the legend of Fig. 2, it is not clear how the 95% confidence interval was calculated. Please provide details regarding the basis for the confidence intervals and specify what was used to compute them (e.g., across trials).

7. In Fig. S15 and S16, the authors mentioned that the earlier and later model layers reach their highest correlation earlier in the motion and color contexts, respectively. However, they did not provide any discussion regarding the implications of this finding. As the significance of this distinctive trend remains unclear, it is recommended that the authors include a discussion on what these results suggest and how they contribute to the overall understanding of the analysis.

8. Regarding the readability, titles of some figures and tables lack sufficient information, making it difficult to understand what they represent and how they differ from other results. For instance, although Fig. S5 has the same format as Fig. 2a and b, with only different time bins, the title of Fig. S5 is identical to that of the entire Fig. 2. Furthermore, the titles and legends of the figures and tables often lack sufficient detail to help readers quickly grasp their content (e.g., analyzed time bins and brain regions). Additionally, there are some incorrect titles in the supplementary tables (e.g., Tables S1 and S2 are not ANOVA results, despite being listed under Section III.2 "ANOVA analysis to determine the best neural measure").

It is recommended that the authors carefully review all figure and table titles and revise them to be more specific and reflective of the content. This will help readers easily understand the data being presented and clearly distinguish between the figures and tables.

9. On page 3 of the Supplementary Information, in the section "II.1 Rate Coding," the authors mention that they used 30 recording sites per brain region but do not explain the criteria for selecting these sites. To ensure that the analysis is replicable, please provide a detailed explanation of the selection criteria used for choosing the recording sites.

10. Regarding the comment 10 of my review, it seems that the authors' response does not fully address the point I raised. While the authors mention that they provided inferential statistics, my comment was specifically about the interpretation of the results supporting reconfiguration. I kindly ask the authors to revisit my original comment and respond appropriately to this concern.

(Remarks on code availability)

The authors have improved the code provided for replicating their findings, addressing many of the concerns raised in my previous review. The updated code now includes more comprehensive coverage, particularly with additional sections for the CNN_LSTM model, which were previously missing. This enhancement ensures that both the brain data analysis and the CNN_LSTM section are adequately represented, reflecting the comparable importance of these parts of the study.

Moreover, the code is now better documented, with clearer instructions on how to run the programs and set up the necessary environment. This makes it much more accessible, even for users who may not have extensive technical expertise. Overall, the revisions have made the code richer and more user-friendly successfully addressing prior feedback.

Reviewer #3

(Remarks to the Author)

Thank you for engaging with my comments and for addressing many of them with this revision.

One comment that I 100% stand by, and that I think needs to still be addressed, is that the graphical display of the data of the individual ROIs should not be buried in the supplement (Figures 2 and 3). They may not be as pretty as the grand average, but especially in light of the authors' discussion on how the ROIs exhibit different effect sizes (see for example MT and V4), this becomes essential. In fact, the observation of different effect sizes alone could imply that a grand average does not precisely represent the data.

Furthermore, I have read the description of the "cognitive model" multiple times and still struggle understanding what exactly was computed/fitted. Could the authors significantly expand this section, and ideally provide a graphical depiction of the model and what was scaled where/how? The main claim of the paper rests on this analysis, as some ROIs only show the stretching effects in this, perhaps more uncommon, analysis. Hence, a clear and detailed description will be of importance. Could the authors also comment on how this analysis is different from deriving attentional model predictions using RSA and fitting the RDMs to jointly model the neural data? This would perhaps be a more common approach (or, say, one that I and perhaps others would be more familiar with).

(Remarks on code availability)

Reviewer #4

(Remarks to the Author)

The authors have thoroughly revised their paper, including taking a new analysis approach. They also added additional background information and definitions that make the manuscript much easier to read and better convey the important take-home message of the work.

The concerns that I raised in the first round of review were responded to appropriately overall. I have just one more remaining minor concern. In the first round of review I questioned the use of the word "radically" as a modifier for how brain areas were reconfigured. In the response, the authors made it clear that what they mean by radical is that stretching occurs across multiple brain areas, not just within limited perceptual regions, and is spread across regions to a larger degree than would be predicted by cognitive models. I found the way they expressed this in the response letter to be persuasive. However, it is still not clear in the manuscript what is meant. I recommend that the authors add a brief definition of what they mean by radical to the introduction and discussion. I think that pointing out how stretching reconfigures activity across many brain regions and going into a bit more depth about the implications of this finding would actually enhance the power of their message to the reader.

(Remarks on code availability)

Version 2:

Reviewer comments:

Reviewer #1

(Remarks to the Author)

This paper has improved substantially through the last two rounds of review.

I am still not fully satisfied with the RSA analysis, simply because as far as I know there are no foundational papers that formally prove the underlying statistical assumptions of this method. Without such proof it is impossible to know when a statistical test should be used or what biases it might produce in the results.

But as the authors note this has become a widely used tool nevertheless, and after all, we use many tools in neuroscience without fully understanding them. In any case the results reported here are broadly consistent with other reports of dynamic

changes in representation during different tasks.

So in my opinion this paper is now acceptable for publication.

(Remarks on code availability)

Reviewer #2

(Remarks to the Author)

The revised manuscript represents a substantial improvement over the original version. I appreciate the authors' efforts in addressing many of the concerns raised during the initial review. Below, I highlight a few points that, if further clarified or revised, could enhance the manuscript's informativeness and transparency for readers.

1. In response to my previous comment #1 regarding the analysis time window, the authors stated: "In our study, data from the 250 ms period was used as the primary analysis based on prior knowledge." However, the nature of this "prior knowledge" remains unclear. I recommend that the authors either cite appropriate references or elaborate on the specific empirical or theoretical basis for this choice. If the decision was informed by preliminary data or prior empirical findings (e.g., in this or related paradigms), this should be explicitly stated.

Clarifying this rationale is particularly important given that stretching effects were observed in later time periods (as reported in Tables S9–S12), suggesting that alternative temporal windows may yield different conclusions. Such information would be valuable for guiding future studies in selecting appropriate time frames for RSA-based analyses.

Furthermore, while the authors included new supplementary analyses that explore extended time windows (Tables S9–S12), these results are currently presented without sufficient context or interpretation in the main text. It would be helpful if the authors could clearly explain the motivation for conducting these additional analyses, describe how they relate to the central claims of the study, and clarify whether—and in what ways—these extended analyses qualitatively alter or reinforce the main conclusions. While some of this reasoning is outlined in the rebuttal, it should be included in the main text to ensure completeness and ease of understanding for readers.

2. Regarding the response to my comment #2, I appreciate the updated results shown in Figure S6, which appear more consistent with the trend observed in the main Figure 2. However, I noticed that the results of the rate coding analysis have changed substantially compared to the original manuscript, and this change is not explicitly acknowledged or discussed in the rebuttal.

I strongly encourage the authors to transparently report and briefly explain the reasons for these changes (e.g., bug fixes, analysis corrections, or different time windows). Additionally, they should comment on whether and how these changes affect the main conclusions of the manuscript.

3. In response to my previous concern (comment #3), the authors revised the manuscript to state that the results were "consistent with our understanding of the function of these areas" (line 178). However, the phrase "our understanding" remains vague and may be difficult for readers unfamiliar with the domain to interpret.

I recommend replacing this subjective phrasing with more objective language. For instance, referencing existing literature or known functional characteristics of the regions (e.g., MT, V4) would clarify the basis of the interpretation. The rebuttal provides a satisfactory explanation, and incorporating that into the manuscript would resolve the issue.

(Remarks on code availability)

I have no further concerns regarding the code.

Reviewer #3

(Remarks to the Author)

I appreciate the edits and clarifications provided in this new revision.

If in any way possible, I would like to ask the editor to allow showing the ROI-specific data in the main paper, instead of only the grand average data.

The grand average is misleading given the different effect sizes, as discussed with and appreciated by the authors earlier. Thus, if space is needed, an option to consider would be to move the grand average plots to the supplement, or dropping them altogether, and to be replace them with the ROI-specific plots.

(Remarks on code availability)

Version 3:

Reviewer comments:

Reviewer #1

(Remarks to the Author)

This most recent revision has further improved the manuscript. I think, for the most part, that we have reached the point of diminishing returns, and that the paper can be published with only a few minor tweaks that do not require further review. Specifically, I think that the authors should put ALL of the figures that they provided to the reviewers into the supplementary figures of the paper. There is no real limit on supplementary figures, so there is plenty of room to put them in there. If they information that they provide was valuable for the reviewers, it will be valuable for the readers.

(Remarks on code availability)

Reviewer #2

(Remarks to the Author)

The authors have made reasonable efforts to revise the manuscript, and I appreciate their responsiveness to the earlier comments. Key issues raised in previous rounds—such as the rationale for the analysis window and the preprocessing steps—have been addressed appropriately.

I consider the current version acceptable for publication. I thank the authors for their engagement throughout the review process.

(Remarks on code availability)

I have no further concerns regarding the code.

REVIEWER COMMENTS

Reviewer #1 (Remarks to the Author):

This study reanalyzes an old awake animal neurophysiology data set to determine whether attention stretches perceptual space in several different brain areas. It then investigates the same attentional effect in a hybrid CNN/LSTM trained on the same task. The authors claim that several high-level brain areas “reconfigure” to stretch perceptual space, and that spike timing is important for this effect. Unfortunately the paper in its current form is very difficult to read. As it is written currently the paper reads as if it were a short conference paper written quickly at the last minute before a deadline. Nature Communications is an archival journal, and the paper should be revised accordingly.

We thank Reviewer 1 for taking the time and effort to review our paper. The team has made its best effort here to strengthen our contribution in light of reviewer comments.

1. The paper is, frankly, quite difficult to read. I could provide many examples of this problem from the text, but for simplicity just consider the first few sentences of the results: “One initial question is which measure of neural similarity is most aligned with the neural recordings [30]. To answer this question, we assessed various measures of neural similarity to determine which one maximized representational similarity [31] with the experimenter-defined stimulus coordinates of color and motion.” Why is this the first result? Is this some preliminary result that has to be established before the main results? Is it a result that is required to avoid a potential confound? And why is a representational similarity approach being used to assess attentional stretching at all? None of this information is provided. Instead, the reader is tossed into the maw of the paper without a rope. And it doesn’t get any better in the rest of the results. The authors should not expect the reader to be willing to wade through the entrails of a paper to figure out which questions are being asked and why. They should not force readers to go look at other papers just to gain some vague idea about what the acronyms SPIKE and ISI actually measure. I could go on here but I think that I have made the point: the authors really need to go back and rewrite this paper with an eye toward conveying information clearly to the reader.

The initial analysis adds a layer of rigor. Rather than assume a similarity function, we first select one based on the data. We now also include some analyses for the rate measures in the Supplement. SPIKE and ISI are now discussed more in the main text. As in the initial submission, they are fully defined in the Supplement. The modifications are as follows:

Page 4, Line 106, “For some time interval, rate coding measures sum the spikes at each recording site to calculate distances. Spike timing within the interval is inconsequential. For example, the Euclidean distance between two items involving five recording sites is simply the Euclidean distance between the two five-dimensional vectors where each vector entry is the sum of the spikes at a site. In contrast, rather than simply sum across the temporal interval, spike timing measures take into account temporal information within the interval, such as the timing between spikes for two items at a site, pooling these distances across recording sites for an overall measure of distance.” has been added.

Supplement, Page 3, we have detailed the mathematical definition of neural coding measures and how we applied them in our cases, as “To construct rate-coding-based RDM, we first represent each stimulus as a n_r -dimensional vector, where n_r is the number of recording sites. Each element of the vector corresponds to the firing rate of a specific neuron/electrode in a time bin. To maintain uniform dimensionality for calculating dissimilarity between stimulus vectors, we used 30 recording sites per brain region, resulting in a total of $n_r=180$ recording sites (sites with no recordings during a trial were excluded from the analysis”, and Page 5, “Both ISI and SPIKE distances are calculated for each recording site and then averaged across all sites. Unlike rate coding, ISI and SPIKE distances are not affected by empty recording sites. Since empty sites return NaN values when no data is recorded, they are excluded from the calculations.”

Supplement, Page 8, Sec. III.2, new figures and revised t-test and ANOVA statistics to show why and how we select the best neural measure.

2. The paper really needs to be reorganized to make the logic of the data analysis procedures clear. It is conventional in neuroscience papers to present the neurophysiology results first, and then to present the computational model (if one is proposed). The current paper does not follow this format, instead blithely intermixing neurophysiology results with analysis of the CNN/LSTM, as if those are all the same thing and so directly comparable. They are not, and intermixing them just obscures the meaning of the results.

We considered this suggestion, but believe that an integrated presentation is consistent with our integrated theoretical approach.

3. The rate versus timing issue should also be broken out into a separate and clear analysis. This is a classic question of enduring interest in neurophysiology. But the

results presented here are so compressed and opaque that it is difficult to compare the two sorts of codes in the current paper.

The basic result is highlighted, including in the abstract. It's not the focus of the paper, but we are pleased Reviewer 1 thinks this finding alone merits publication of our work. We do better unpack the significance of these findings in the revision. The details are as follows:

Page 4, Line 109, "In contrast, rather than simply sum across the temporal interval, spike timing measures take into account temporal information within the interval, such as the timing between spikes for two items at a site, pooling these distances across recording sites for an overall measure of distance." has been added.

Page 5, Line 119, "Thus, ISI may have an advantage over SPIKE given the potential absolute timing differences across trials in temporal codes." has been added.

Page 10, Line 217, we discussed the significance of these findings as "Finally, although not a primary hypothesis, the monkey spiking data afforded the possibility of evaluating whether spike timing or rate-based measures better characterize the neural code. Spike timing did appear relevant to how the brain represented stimulus information (see Fig. 2). These findings suggest a roll for spike timing that should be further explored, including in computational models."

4. Given the convoluted, unclear logic of the data analysis procedures, and the very sparse treatment of the methods and the results, I confess that I am not sure what I can really conclude from the results presented here. I think that the revision should expand the methods and the results so that things will be more comprehensible.

Given the extensive revisions, we hope the reviewer now finds the manuscript easier to follow.

5. In the introduction and middle of the results the authors talk of attention stretching the perceptual space, which is fairly clear given prior theories and results in the field. But the latter part of the results focuses on the question of which brain regions (and the LSTM) reconfigures itself. This concept is much more woolly, at least in the neurophysiology literature. What precisely is meant by this? What kind of neurophysiology data would support this or not? How does this relate to stretching/warping? It is very unclear as presented in the current manuscript.

To clarify this point, in the revision, Page 1, Line 38, we have cited and added “While numerous attentional influences on neural activity have been observed \cite{Connor1996, Gallant2008, Treue2010, Gallant2013}, we focus on the specific case of adaptive stretching along a goal-relevant stimulus dimension.”

As we cite, there are many studies related to attention warping and even some related to stretching along a canonical stimulus dimension as we consider here. Our reported results provide a concrete answer to Reviewer 2’s question.

6. I can that the authors should cite their own prior work on attentional stretching (e.g., Mack et al 2016; Braunlich & Love 2019). But they ignore prior neurophysiology and fMRI work on precisely this issue that predates any of the experimental studies cited here. For example, work by Connor (1996), Gallant, (2008) and Treue (2010) showed these effects in V4 and MT. And Gallant (2013) used fMRI to show attentional warping across much of the brain. This prior literature should be described and cited here, to do otherwise would be poor scholarship.

As in the original submission, we cite other authors for this point. The Folstein et al. paper we cite was particularly influential.

The first author reviewed the papers Reviewer 1 suggested. They were related to the general topic of our submission, but do not directly bear on our research question. Our point is more specific than an attentional effect. There are many studies that show such effects. Instead, our focus on attentional stretching along a canonical stimulus dimension (i.e., feature) in response to varying task goals. We cite all the papers Reviewer 1 listed to make clearer how what we did differs from previous efforts, which hopefully clarifies this point.

From the first author’s notes supplied to me:

“Connor (1996) is top-down attention, where monkeys were trained to release the response lever when the attended ring disappeared. This reference focuses on the spatial interaction between stimulus ring position and response strength in V4 cells. While it indicates that V4 carries information about spatial attention towards stimuli, no representational stretching and no feature dimension to select, which is fundamentally different from our study.”

“Gallant (2008) is top-down attention, where the monkeys grab the capacitive touch bar until the array contains the target image (a random array of visually similar distractors, with no constraints on eye movements). This reference shows that many neurons can

actually change their spectral tuning in searching target. First, it focuses on neuronal tuning, while our study focuses on neural representation. Second, the target consists of natural images from a library of black and white digital photographs. While the specific features being attended to (such as texture, shape, or contrast) are not clear, they definitely do not include features like color or movement information. Hence it is different from our study, which focuses on representational effects in both the task-relevant and task-irrelevant dimensions, specifically examining color and motion dimensions.”

“Treue (2010) is top-down attention, where monkeys were trained to attend to spatial arrangement stimuli and hold the lever. This study differs significantly from ours. While Treue (2010) emphasizes the impact of spatial attention on the responses of neurons to preferred and anti-preferred stimuli, it does not involve feature dimensions.”

“Gallant (2013) focuses on top-down attention in a semantic category task where subjects searched for specific categories (humans or vehicles) in natural movies and pressed a response button whenever an exemplar of the attended category appeared. This reference shows that response tuning changes for a single voxel in different semantic categories (i.e., warping), which might shift their responses to favor certain signals over others and ultimately reshape the overall neural representation.

There are several key differences between Gallant (2013) and ours:

1. Behavioral Impact: Their behavior is not affected by trial-by-trial error, and the paper focuses on experiments involving passive viewing tasks.
2. Feature Selection: They involve attention stretching within dimension (semantic dimension i.e., selecting specific word from many words), while we focus on attention stretching across dimensions (color & motion).

We focus on the top-down attention influenced by trial-by-trial error and have identified selective attention between two feature dimensions.”

7. Furthermore, the two neurophysiology studies showed stretching of the spatial or feature-based receptive fields of single V4 neurons, a result that seems at odds with the result reported here. In fact the Treue lab has had a whole series of papers that have shown this effect in MT. This difference seems likely either due to differences in the tasks used in the various studies, or to differences in the way that the data were analyzed and modeled. (If I have read the paper correctly, I suspect this is because attentional stretching/warping here was only assessed across dimensions, whereas

prior studies on attentional stretching/warping also looked at changes in tuning within dimensions.) This needs to be addressed thoroughly and convincingly. If the cause of the difference in results cannot be determined definitively, the various options should at least be discussed and described.

I believe this has now been addressed.

8. I confess that I'm a bit confused about how the CNN/LSTM component of this study contributes anything. The CNN/LSTM architecture is quite different from the architecture of the brain, and the training regime is also quite different between the two systems.

Training closely parallels the stimulus sequences and feedback monkeys received. A formal model that is somewhat of a blank slate that also shows stretching effects by minimizing error bolsters our adaptive hypothesis.

9. The CNN/LSTM architecture used here is not standard. The authors write, "The CNN front-end is used to simulate the monkey visual system and the stacked LSTM is to simulate multiple brain regions such as MT, V4 which might be involved in the learning task." This is a potentially interesting but very unusual architecture. And it isn't very clear precisely why this was chosen, or why it is needed. The entire architecture is poorly motivated and poorly justified.

The model is fed the same image sequences that the animals experienced. Essentially, the input is a video, which is relatively innovative. The CNN is pretrained and provides the perceptual representations that the LSTM (time series model) learns from. The LSTM forms a state representation based on the task cue (processed by the CNN). The CNN could not learn this task alone because it is a model of object recognition with no persistent state through time (over a trial). The LSTM's representations stretch along the relevant stimulus dimension, like what we observe in the monkey data.

10. In most other studies using ANNs to model the visual system, the CNN is used to model the complete hierarchy. Given this it is a bit disappointing that the authors did not evaluate a pure CNN-based architecture, and that they did not try to evaluate precisely how the CNN and LSTM components of their network contribute to the claimed results. This really needs to be done, and given the ease of fitting ANNs these days there is really no reason not to give this a more systematic treatment.

This is not relevant to the paper's stated aims. We are not evaluating CNN's as models of a visual hierarchy, which itself is up for debate (e.g., <https://www.science.org/doi/10.1126/sciadv.abm2219>). The study design and stimuli are

also poorly matched to that research question, but perfectly suited to our research question.

The CNN is simply there to provide visual representations for the LSTM to learn from. Furthermore, the CNN-LSTM, as the paper states, is decidedly not intended as a model of the brain. Instead, it is an error-minimizing system that displays stretching. It makes sense to analyze the LSTM, but not the CNN, for the questions we ask.

11. The authors use RSA as a core method for their analysis. Justification for this choice of analysis should be provided. Why was this method chosen? What are the advantages and disadvantages relative to methods used in other papers (most commonly GLM-based methods)? Furthermore, some information should be provided about the statistical assumptions underlying RSA, and whether they are met here. No such information is provided in the current manuscript.

Our analyses concern changes in similarity with task demand, which fits RSA. Thankfully, every paper does not have to be a methods paper that pits commonly used methods against one another. Occasionally, papers can use established methods to make theoretical contributions. We hope the reviewer is satisfied with our revision making it clear why we use RSA for model comparisons (along with the revised statistics). We now adopt a consistent approach to the statistical analyses which improves readability.

The modifications are as follows:

Page 7, Line 144, “To assess how the six brain regions and LSTM model layers are modulated across dimensions in a quantitative manner, we conducted a cognitive model-based analysis (adapted from the generalized context model \cite{nosofsky1986attention}) using 5-fold cross-validation. Unlike the previous analysis, this model-based analysis uses all the data as opposed to select pairs of items and assumes a dimensional structure (color and motion) when estimating stretching along the relevant stimulus dimension. These aspects may make this analysis more sensitive and reveal stretching across all six brain regions, whereas the previous analysis only found significant effects in PFC, FEF, LIP.” has been added.

Page 7, Line 152, “Specifically, we conducted a cognitive model that assigns w_m (ranging from 0 to 1) to the motion dimension and $1-w_m$ to the color dimension. By utilizing representational similarity analysis (RSA), we were able to optimize the cognitive model parameters to best match observed patterns of neural activity (see

Methods), thereby inferring the relative attention of motion and color information in the task context.” has been added.

Page 9, Line 188, “To quantify this relationship, we considered how the RSA-based correlation between brain region (V4 or MT) and model-layer activity varied with task context. We performed a three-way repeated ANCOVA with fixed factors of region (MT or V4), task (motion or color relevant) and layer (1-6, a continuous variable) and a random effect (4 simulations). The interaction between region (MT or V4) and task (color or motion) was $F_{(1,85)}=670.5$, $p<0.0001$ (Supplementary Table S17), indicating that MT aligned more with the LSTM when motion was relevant whereas V4 did when color was relevant.” has been added.

Page 13, Line 314, we detailed “In order to relate our artificial deep neural network to the brain, we utilize representational similarity analysis (RSA), an experimental and analytical framework, relating representations in the brain and our model by computing and comparing representational dissimilarity matrices (RDMs) that characterize the information encoded in a given brain or model. Each 16×16 RDM in this study contains distances/dissimilarities for pairwise stimuli. The distance indicates how dissimilar those two stimuli are in the monkey brain or how dissimilar those two model representations are in the model.”

Page 13, Line 331, we detailed “To identify the best neural dissimilarity measure to analyze neural data, we opt for a RSA-based approach where we correlate RDMs derived from candidate neural dissimilarity measures to a static reference matrix of stimuli. This reference matrix reflects experimenter’s intended relationship between stimuli and we assume that the best measure should be the one that yields the highest Spearman’s rank correlation to the reference matrix.”

12. The procedures used to derive RDMs for the LSTM model are quite complicated, and it appears that many specific decisions were made that are not clearly justified based on first principles or prior art. How can we trust these results or their interpretation?

We wish the Reviewer was more specific in their criticisms, as we fail to see any issues. All our choices were straightforward given the goal with the first thing we tried working. The main goal for the LSTM was to get it to learn to perform the task from the inputs analogous to what the monkeys received. In general, the presentation should be clearer. We now include more rigorous statistics and comparisons to baseline models that will hopefully satisfy the reviewer.

The modifications are as follows:

Page 14, Line 346, “The LSTM-based representational dissimilarity matrix (RDM) is constructed purely from the output of the LSTM cell. After the CNN-LSTM model processes a trial (i.e., a stimulus), it generates a stimulus representation as a vector of shape (1000,) for each layer, as the LSTM cell size was set to $h=1000$. During training trials, the layer parameters are updated, but they are fixed in test trials. Since each stimulus can be represented as a 1000-dimensional vector consisting of model representations, we are able to compute the dissimilarity between these stimulus representations using Euclidean distance (Fig. 3) and Cosine distance (Supplementary Fig. S17). Specifically, we computed and averaged the Euclidean distance and Cosine distance for every pair of stimuli, forming a 16x16 dissimilarity matrix. In total, we derived 6 RDMs, one for each of the 6 individual LSTM layers. The entries in each RDM reflect the dissimilarity between the LSTM representations.” has been added.

Page 8, Line 169, “We conducted a cognitive model-based analysis on the LSTM that paralleled the previous analysis on the brain data. The LSTM afforded the opportunity to evaluate stretching across layers and training. The results were analyzed by a three-way repeated ANCOVA on the Spearman correlation scores, incorporating fixed factors of model (cognitive or baseline, a categorical variable), training step (1-10, a continuous variable) and LSTM layer number (1-6, a continuous variable), and with simulations (4 simulations) as a random effect. The key finding was that the cognitive model was superior to the baseline model ($F(1,4789)=6.2e4$, $p<.0001$, see Supplementary Table S16 for other statistics). The fitted attention parameter of the cognitive model was increasingly task driven as training progressed and at more advanced LSTM layers (Supplementary Fig. S12), whereas the baseline model's fitted parameter was flat across these two factors (Supplementary Fig. S13).” has been added.

13. The model-free analysis of dimensional stretching is not, in fact, model free. Any statistical test has an underlying statistical model, and an RDM-based analysis certainly does. Rather than stating that this is a model-free approach, the authors should either state the underlying model and show that the data fit the assumptions of that model, or they should use a simpler approach (based on a GLM or some other straightforward method), as others have done.

From the context and standard usage of terms, we hope it is now clear that we are not referring to statistical models in the passage. The analyses have been refreshed so hopefully the Reviewer is now satisfied.

14. The authors use ANOVA, t-tests, and permutation tests. Permutation tests should be used in all cases, just to provide some consistency in the statistical methods, and to maximize robustness in these classical significance-based statistical tests.

Statisticians have developed different tests because different situations and assumptions require different tests. In response to other concerns, we have revised the statistical approach which now largely consists of ANOVAs and ANCOVAs, providing a consistent approach and presentation. For readability, most of the details of these factorial analyses are in the Supplement with key results reported in the main text.

Reviewer #2 (Remarks to the Author):

In the manuscript, the authors tested a hypothesis regarding the reconfiguration of visual representations along task-relevant dimensions using a previously published monkey dataset collected during color- and motion-related decision-making tasks. They also investigated whether the same reconfiguration was observed in an artificial neural network model (CNN_LSTM model) trained with the same inputs and rewards as the monkeys. The authors initially conducted a representational similarity analysis (RSA) to identify the best neural similarity measure from various versions of the rate coding measure and spike timing measures using the experimenter-intended coordinate, which assumes a fixed reference in both color and motion conditions. Subsequently, the authors employed inter-spike intervals to test their dimensional stretching hypothesis by examining whether neural distances between experimental conditions were modulated in task-dependent manners. The authors demonstrated modulations on neural distances between conditions relevant to the tasks in both the monkey brain and the deep learning model. They also applied a cognitive model that can take attentional effects on similarity matrices into consideration to illustrate task-dependent weight modulations for both the monkey brain and the model. Based on these results, the authors claimed that both the monkey brain and deep learning model exhibited response modulations depending on task relevance, indicating that stretching is an adaptive strategy for both biological and artificial agents.

The correspondence between the brain and artificial neural networks (ANN) beyond sensory cortical areas is a hot topic in the fields of neuroscience and computer science, attracting the interest of researchers with a broad background. While task-dependent attentional modulations on neural representations in broad brain areas were not entirely new (e.g., Çukur et al., Nat. Neurosci., 2013, for human data), demonstrations with color- and motion-relevant tasks with the monkey brain, along with the attempt to explore a similar mechanism in ANN, would provide additional insights into this topic. However, despite potential interest in the question and framework, the evidence supporting their claims is not compelling. Various concerns, particularly related to the transparency of results and methods, as well as the validity of statistical tests, need to be addressed. In particular, it was quite difficult to fully understand the details of the methods only from the manuscript to replicate the present study. Please see below for details on these issues.

Thank you for the careful reading of our manuscript and taking the time to make helpful suggestions. As you will see, we have completely revamped our approach to the statistical analyses. We have also added the Çukur et al cite to the Introduction, making

clear how what we did differs (i.e., task-relevant stretching along a clear stimulus dimension or feature).

Specifically, in the revision, Page 1, Line 38, “While numerous attentional influences on neural activity have been observed \cite{Connor1996, Gallant2008, Treue2010, Çukur2013}, we focus on the specific case of adaptive stretching along a goal-relevant stimulus dimension.” has been added.

1. The most crucial issue in this manuscript is the absence of statistical tests conducted to validate the reproducibility of the present findings with independent (or new) data samples. For instance, one of the main results illustrating dimensional stretching in the monkey brain and CNN_LSTM (Figure 3, lines 118-123) was performed to assess significance across variances of elements in RDMs (24 item pairs; combinations of stimulus conditions). With this setup, the authors can confirm that significant differences were observed in the current dataset and can claim that similar results might be observed in additional combinations of stimulus conditions. However, because variances across data samples were not tested, they cannot claim that this finding is generalizable to new data samples (e.g., trials and monkeys), which is crucial for supporting the reproducibility of the findings. Particularly, the observed effect size was quite small ($M = 0.0007$) compared to the baseline dissimilarity (ranging from 0.359 to 0.363) and the 95% confidence intervals (Fig. 2a and b, ~ 0.5). Therefore, it would be critical to examine if similar differences are observed robustly under variances across data samples. Likewise, the authors tested the significance of attentional modulations by comparing the results with null distributions produced by label shuffling (Figure 4), while such usage of statistical tests cannot support the reproducibility of the results. Overall, given the small effect size reported, the statistical tests performed in the present manuscript seem not sufficient to support the reproducibility of the findings. I recommend the authors conduct statistical testing to support the reproducibility of the findings, as the dataset contains rich data samples (e.g., 40 trials with different random dots' initial positions). The same analysis can be performed to test the CNN_LSTM model analysis.

Thank you for these comments. We took them to heart and completely redid the statistical analyses, relying on ANOVAs and ANCOVAs with random effects. We removed the permutation test.

We went further than what reviewers requested and in our main RSA analysis compared a model of stretching to a baseline model that itself can stretch representations, albeit not in a task-driven manner (i.e., the model can be biased toward

color or motion). This is a stricter and more appropriate test that found all considered regions displayed significant stretching (with a random effect of monkey).

The modifications are as follows:

We have included multiple statistical testing towards small observed effect size using independent data among subjects.

(a) For brain analyses, a two-way repeated measures ANOVA with fixed factors (motion/color mismatch and motion/color task) and a random effect (subject) was conducted on 24 stimulus pairs (figures and statistics in Supplement Sec. III.3 and III.4, see Table S12-13 and Figures S8-10).

Page 6, Line 132, “We assessed this stretching effect at the whole brain an regional level by two-way repeated measures ANOVA with fixed factors of task (color or motion relevant) and mismatching dimension (color or motion mismatch) with a random effect for the monkey subjects. Stretching is consistent with an interaction such that dissimilarity is greatest for pairs mismatching on the dimension (color or motion) that is task relevant. This interaction was significant in the whole brain analysis, as well for analyses of PFC, FEF, LIP (see Supplementary Fig. S8 and Table S12). For comparison, rate coding results are reported in Supplementary Figs. S9-10 and Table S13.” has been added.

(b) For LSTM analyses, a three-way ANOVA with fixed factors (motion/color mismatch and motion/color task as categorical variables, while layer number as a continuous variable) and a random effect (simulations) was conducted to analyze 24 stimulus pairs, see Table S14 and Figure S11.

Page 6, Line 139, “We analyzed stretching in the LSTM using the same analysis strategy used for the brain, but simulation served as the random effect instead of monkey and a third continuous factor was included for LSTM layer (1-6). Our main prediction was that the degree of stretching would increase across layers (i.e., stretching would be most pronounced at advanced layers), which would lead a three-way interaction, which we observed ($F(1, 52981) = 1.59e4$, $p < .0001$). See Supplementary Fig. S11 and Table S14 for full details.” has been added.

We have also included statistical testing towards RSA analysis compared a model of stretching to a baseline model using independent data.

(c) For brain analyses, we adopted 5-fold cross-validation per task and per monkey. A one-way repeated ANOVA was conducted to evaluate Spearman correlation score with a fixed factor of model (cognitive or baseline model) and a

random effect for monkey subjects. The correlation scores are obtained from 5-fold cross-validation, while the attention weights are trained using 4 folds. For the cognitive model, these 4 folds are from the same task context, whereas for the baseline model, 2 folds are from the color task and 2 folds are from the motion task. A stronger correlation score was observed and found significant across all brain regions (Table S15).

Page 8, Line 156, “We employed a 5-fold cross-validation procedure to train the attention weights w_m using four folds within the same task context. The fitted attention parameters (i.e., w_m on the motion dimension) of the cognitive model under two tasks (motion and color relevant) are illustrated in Fig. 4a. To assess the effectiveness of our cognitive model-based analysis, we compared it to a baseline model structurally identical to the cognitive model but fitted to two folds from the color task and two folds from the motion task. We evaluated the RSA-based correlation score using a one-way repeated ANOVA with a fixed factor of model (cognitive or baseline model) and a random effect for the monkey subjects. A stronger correlation was observed for the cognitive model and found to be statistically significant across all brain regions examined (whole brain: $F(1,37)=5.96$, $p=0.0195$; PFC: $F(1,37)=4.84$, $p=0.0342$; FEF: $F(1,37)=6.31$, $p=0.0165$; LIP: $F(1,37)=4.73$, $p=0.0362$; IT: $F(1,37)=11.8$, $p=0.0015$; MT: $F(1,37)=4.31$, $p=0.0449$; V4: $F(1,37)=10.2$, $p=0.0028$; see Fig. 4b and Supplementary Table S15). This finding suggests that the allocation of attention to motion or color features is adaptively modulated based on task contexts, and that this modulation is particularly pronounced in all brain areas.” has been added.

(d) For LSTM analyses, a three-way repeated ANOVA was conducted to evaluate the Spearman correlation score, with fixed factors of model (cognitive or baseline), layer number (1 to 6 LSTM layers), and training step (500-5000; in 500-step intervals, i.e., 1-10) and a random effect for simulations (4 simulations). The model type is treated as categorical variables, while the training step and the layer number are treated as continuous variables. The results showed a significantly stronger correlation score, influenced by the model, training step, layer number, and their interactions (e.g., model and training step, model and layer number).

Page 8, Line 169, “We conducted a cognitive model-based analysis on the LSTM that paralleled the previous analysis on the brain data. The LSTM afforded the opportunity to evaluate stretching across layers and training. The results were analyzed by a three-way repeated ANCOVA on the Spearman correlation scores, incorporating fixed factors of model (cognitive or baseline, a categorical variable),

training step (1-10, a continuous variable) and LSTM layer number (1-6, a continuous variable), and with simulations (4 simulations) as a random effect. The key finding was that the cognitive model was superior to the baseline model ($F(1,4789)=6.2e4$, $p<.0001$, see Supplementary Table S16 for other statistics). The fitted attention parameter of the cognitive model was increasingly task driven as training progressed and at more advanced LSTM layers (Supplementary Fig. S12), whereas the baseline model's fitted parameter was flat across these two factors (Supplementary Fig. S13)." has been added.

Additionally, we have shown the spearman correlation in the whole trial, with the standard variance across sessions (Figure S4). The confidence interval of Fig. 2 in the main text is computed by $\text{CI} = \text{Mean} \pm \left(1.96 \times \frac{\text{Standard Deviation}}{\sqrt{\text{Num. of Session}}}\right)$, hence it would be a tight error-bound than standard deviation due to the number of sessions.

2. Regarding the validity of the neural measure selection, the authors claimed that "Spike timing was crucial to this code" (Abstract) based on the results showing relatively high Spearman correlations by ISI and SPIKE in both the motion and color tasks than those based on rate coding (Figure 2). However, I was confused about the results in Fig. 2 at several points. For example, I wonder why the Spearman correlation was moderately high for ISI and SPIKE from the early timing after stimulus onset, while relatively late (at least ~50-100 ms) rises of stimulus-related responses were reported in the original study of the dataset (Siegel et al., 2015). The 50 ms time window may explain this early rise of correlations, but the same trend was observed even with a shorter time window (20 ms, Extended Data Fig. 2). Is there any possibility that these high correlations do not actually reflect differences in neural responses to different experimental stimuli (modeled by experimenter-intended coordinates)? To make this point clearer, I recommend that the authors show the time course of the Spearman correlation for the whole trial period (from fixation to response) to confirm that the observed correlations truly reflected stimulus-related representations, which may provide positive control of the validity of the analysis. If high correlations were observed even before the stimulus onset, the rationale for selecting the spike timing measure for the main analysis might be violated, leading to the failure to support the authors' claim about the importance of spike timing as stated in the abstract.

Thank you for this sensible suggestion; it's always good to do these sanity checks. Fortunately, these effects are driven by the stimulus. The correlations were high at time 0 because that time bin starts at 0 s and continues to .05 s. We now plot Figure 2

starting before 0 s so the ramp up is clear. Figure S4 takes this further to show more of the ramp up and ramp down.

3. Related to the results in Figure 2 and comment 2, I wonder why the results by the rate coding did not exhibit positive correlations at all. As the authors mentioned in the Discussion (lines 168-170), the activity of the LSTM is akin to rate-based coding. If the LSTM model showed positive correlations to the experimenter intended coordinates (actually it was not shown in the manuscript), the rate coding results should also exhibit positive correlations in this analysis. Therefore, I recommend the authors elaborate on this apparent contradiction and show the Spearman correlation between the experimenter intended coordinates and representational dissimilarity matrices (RDMs) derived from the LSTM model. Furthermore, because the manuscript lacks the details of how RDMs were computed from the LSTM model in the main analysis (e.g., Fig 3; although the authors explicitly described that they used cosine similarity in Extended Data Fig. 5), I suggest the authors provide additional details to allow readers understand and replicate the methods in the main analysis.

This is another good check. Fortunately, the results hold. Upon further checks, we did notice that some recording sites were “bad”, reporting not a number (NaN). Removing those problematic sites from the analyses smoothed the results. We also provide some analyses of rate coding measures, which, as one would likely predict, show some effects but are not as robust as the ISI timing measure. From line 135, “For comparison, rate coding results are reported in Supplementary Figs. S9-10 and Table S13.” When looking at Table S13, the interaction term is key to assessing stretching effects in our new analyses.

The details are follows:

Supplement, Page 5, “Both ISI and SPIKE distances are calculated for each recording site and then averaged across all sites. Unlike rate coding, ISI and SPIKE distances are not affected by empty recording sites. Since empty sites return NaN values when no data is recorded, they are excluded from the calculations.” has been added.

Page 14, Line 346, we have detailed how RDMs were computed from the LSTM model in the main analysis, “The LSTM-based representational dissimilarity matrix (RDM) is constructed purely from the output of the LSTM cell. After the CNN-LSTM model processes a trial (i.e., a stimulus), it generates a stimulus representation as a vector of shape (1000,) for each layer, as the LSTM cell size was set to $h=1000$. During training trials, the layer parameters are updated, but they are fixed in test trials. Since each stimulus can be represented as a 1000-dimensional vector consisting of model

representations, we are able to compute the dissimilarity between these stimulus representations using Euclidean distance (Fig. 3) and Cosine distance (Supplementary Fig. S17). Specifically, we computed and averaged the Euclidean distance and Cosine distance for every pair of stimuli, forming a 16x16 dissimilarity matrix. In total, we derived 6 RDMs, one for each of the 6 individual LSTM layers. The entries in each RDM reflect the dissimilarity between the LSTM representations.” has been added.

4. Related to comment 3, I suspect that the difference between the two spike measures and the rate coding measure might be due to the differences in the computation procedures for these measures. In the Supplemental Information, the authors described that “To construct rate-coding-based RDM, we first represent each stimulus as a 1D vector whose entries are the number of spikes across neurons/electrodes.” This procedure was totally different from the procedure for constructing RDMs for spiking measures described in lines 277-281: “When processing neural data, we extract the spike trains of isolated neurons detected under electrodes. Regardless of which candidate measure we use, each element in task-relevant RDMs is the averaged distance (dissimilarity) from spike trains (in the time bins) across task-relevant trials and monkeys...”. To conduct a fair comparison between the rate-coding measure and spike timing measures, it would be better to use a consistent method for constructing RDMs. Thus, I suggest the authors perform the same analysis in Fig. 2 using RDMs constructed by a similar procedure with the spike timing measure (e.g., first computing RDMs for individual trials/monkeys and then averaging them).

We apologize that we described the construction of these vectors and recording sites incorrectly in the original submission. The actual analyses were done correctly, but the descriptions of what we did were poor. What we did was in fact comparable across approaches.

Here is the revised text in the Result’s section that correctly explains what we did:

Page 4, Line 106, “For some time interval, rate coding measures sum the spikes at each recording site to calculate distances. Spike timing within the interval is inconsequential. For example, the Euclidean distance between two items involving five recording sites is simply the Euclidean distance between the two five-dimensional vectors where each vector entry is the sum of the spikes at a site. In contrast, rather than simply sum across the temporal interval, spike timing measures take into account temporal information within the interval, such as the timing between spikes for two items at a site, pooling these distances across recording sites for an overall measure of distance.” has been added.

5. Related to the analysis in Fig. 2 and the validity of the experimenter intended coordinates, I actually was unable to understand the intention the authors have for the reference matrix. As mentioned by the authors, the matrix assumes a fixed reference in both color and motion conditions and then cannot be suited for aligning neural/model RDMs, which can have representations biased to specific tasks (e.g., V4 or MT). An assumption of the matrix is that a single step difference along the color axis is equivalent to any difference along the color and motion axes (e.g., the difference between orange to pink is equivalent to -90 to -30). As this assumption is not intuitive, it would be helpful to add more explanations about this point for readers to better understand the authors' intentions.

Here, we really relied on the authors of the original study to create a decently balanced stimulus set. If one dimension dominated the similarity space, then the original authors could not have observed task effects. We presume in the process of constructing their stimulus set they did some pilot work to get a decent approximation of balance. That said, we agree getting this perfectly right is impossible and different animals may bring different biases to the task. Fortunately, ISI was so far beyond the rate-based measures that this concern is somewhat minimized.

Fortunately, our new statistical analyses of stretching admit this possibility, focusing on interactions in similarity driven by task and stimulus dimension match. Moreover, our new model-based analyses include a baseline model (which the stretching model beats in every region) that allows individual animals to have biases toward either dimension.

6. On lines 118-121 and in Fig. 3a and b, the manuscript lacks information on which brain area was used for the analysis. Please provide the details of this point. Furthermore, to support the claim that “the brain radically reconfigured itself to stretch representations along task-relevant dimensions in lateral PFC, frontal eye fields (FEF), lateral intraparietal cortex (LIP), and inferotemporal cortex (IT)” in the abstract, provide the results for individual brain areas with appropriate statistical tests as I suggest in comment 1.

This is a good suggestion that we took even further. While Figure 3 does include all the brain measures to show the overall effect, we now provide statistical analyses and figures for each individual region in the Supplement (see Figures S8-10 and Tables S12-13). The data are now much more fully analyzed than in the original submission.

7. In Fig. 3d, I was unable to understand why all dissimilarity values start from 0 at the 0.00 LSTM time. In my understanding, the dissimilarity does not necessarily go through

0 at this point. I recommend the authors show the time course during the entire trial for the LSTM model to enhance the clarity of the entire data. If the authors performed any normalization to set the dissimilarity at the onset to be 0, please also provide detailed information about the operation and elaborate on the rationale.

A related issue came up above with regards to time bins. Notice that Figure 3D is concerned with stretching by considering mismatches on color or motion. Although the values are small, they are above 0 in the first time bin. Figures S15 and S16 might make more sense when we look at the LSTM's ability to capture the structure of the entire stimulus space over a wider time course.

8. Regarding the statement in lines 64-65: "one possibility is that these control systems themselves can be learned by nonhuman animals and ANNs, enabling them to reconfigure themselves in response to task cues", it would be helpful if the authors could show the increasing effect of dimensional stretching through the training/learning of the LSTM model. This is a demonstration often reported in studies (Yamins et al., 2014; Kell et al., 2018).

Thank you for these excellent suggestions, which we adopted. Please see Figure S14 and consider,

Page 8, Line 169, "We conducted a cognitive model-based analysis on the LSTM that paralleled the previous analysis on the brain data. The LSTM afforded the opportunity to evaluate stretching across layers and training. The results were analyzed by a three-way repeated ANCOVA on the Spearman correlation scores, incorporating fixed factors of model (cognitive or baseline, a categorical variable), training step (1-10, a continuous variable) and LSTM layer number (1-6, a continuous variable), and with simulations (4 simulations) as a random effect. The key finding was that the cognitive model was superior to the baseline model ($F(1,4789)=6.2e4$, $p<.0001$, see Supplementary Table S16 for other statistics). The fitted attention parameter of the cognitive model was increasingly task driven as training progressed and at more advanced LSTM layers (Supplementary Fig. S12), whereas the baseline model's fitted parameter was flat across these two factors (Supplementary Fig. S13)."

9. In Fig. 4, the authors showed weights fitted to the data from individual brain areas and I agree that comparing weights estimated from the data under different tasks is a good demonstration of the task-specific reconfigurations. However, to make a stronger demonstration of the claim, I suggest the authors evaluate how this fitting enhances the Spearman correlations between RDMs derived from brain data and RDMs of the

baseline reference (i.e., experimenter-intended coordinates). Furthermore, as I suggest in comment 1, the statistical test performed with null distributions generated by label shuffling does not support the reproducibility of the observed effect on new data samples. To support the reproducibility, the authors may want to fit weights using a subset of data and test using independent test data samples to examine if the weighted model show higher correlations than the original reference or the opposite model (e.g., using the color-task-fitted model for evaluation of motion task data fit) as the authors performed similarly in the CNN_LSTM model analysis.

We took this comment and ran with it, as touched on above. We now analyze individual brain areas as well, all using random effects. Are new model-based analysis compares to a baseline model that is a tougher bar to pass as it also has an attention parameter that can tune itself in hopes of increasing performance on the held-out test data. In this new analysis, which leverages all the data, we find stretching in all brain regions. We thank the reviewer for pushing us to do these analyses.

The modifications as follow:

Page 8, Line 156, “We employed a 5-fold cross-validation procedure to train the attention weights w_m using four folds within the same task context. The fitted attention parameters (i.e., w_m on the motion dimension) of the cognitive model under two tasks (motion and color relevant) are illustrated in Fig. 4a. To assess the effectiveness of our cognitive model-based analysis, we compared it to a baseline model structurally identical to the cognitive model but fitted to two folds from the color task and two folds from the motion task. We evaluated the RSA-based correlation score using a one-way repeated ANOVA with a fixed factor of model (cognitive or baseline model) and a random effect for the monkey subjects. A stronger correlation was observed for the cognitive model and found to be statistically significant across all brain regions examined (whole brain: $F(1,37)=5.96$, $p=0.0195$; PFC: $F(1,37)=4.84$, $p=0.0342$; FEF: $F(1,37)=6.31$, $p=0.0165$; LIP: $F(1,37)=4.73$, $p=0.0362$; IT: $F(1,37)=11.8$, $p=0.0015$; MT: $F(1,37)=4.31$, $p=0.0449$; V4: $F(1,37)=10.2$, $p=0.0028$; see Fig. 4b and Supplementary Table S15). This finding suggests that the allocation of attention to motion or color features is adaptively modulated based on task contexts, and that this modulation is particularly pronounced in all brain areas.” has been added.

Page 8, Line 169, “We conducted a cognitive model-based analysis on the LSTM that paralleled the previous analysis on the brain data. The LSTM afforded the opportunity to evaluate stretching across layers and training. The results were

analyzed by a three-way repeated ANCOVA on the Spearman correlation scores, incorporating fixed factors of model (cognitive or baseline, a categorical variable), training step (1-10, a continuous variable) and LSTM layer number (1-6, a continuous variable), and with simulations (4 simulations) as a random effect. The key finding was that the cognitive model was superior to the baseline model ($F(1,4789)=6.2e4$, $p<.0001$, see Supplementary Table S16 for other statistics). The fitted attention parameter of the cognitive model was increasingly task driven as training progressed and at more advanced LSTM layers (Supplementary Fig. S12), whereas the baseline model's fitted parameter was flat across these two factors (Supplementary Fig. S13).” has been added.

10. In Fig. 5, the authors compared RDMs of V4/MT and the LSTM model for each of the color and motion tasks to claim that “LSTM layers freely reconfigure as a function of the task context”. However, the observation that “For example, V4 shows a decent correspondence with LSTM layers when color is relevant, whereas MT does when motion is relevant.” (lines 142-143) can also occur when the LSTM-RDM was fixed, and V4-/MT-RDMs changed depending on the tasks. For rigorously testing the claim, it would be better to additionally show higher Spearman correlations between motion-task-fitted brain-derived-RDMs (including MT and other areas) and the motion-data-fitted LSTM-RDM than between motion-task-fitted brain-derived-RDMs and the color-data-fitted LSTM-RDM. Similarly, Spearman correlations between color-task-fitted brain-derived-RDMs and the color-data-fitted LSTM-RDM can be compared with those between color-task-fitted brain-derived-RDMs and the motion-data-fitted LSTM-RDM. If higher correlations were observed from this analysis, it would strengthen the support for the authors’ claim.

Thank you, we now provide inferential statistics to back this point.

The modifications are as follows:

Page 9, Line 179, “Despite the LSTM and brain both adaptively stretching representations depending on the task, there is no straightforward layer-to-brain correspondence between LSTM layers and brain regions (Fig. 5). Previous analyses (e.g., Fig. 3) indicated that the LSTM reconfigures more in response to the task than the brain. Whereas LSTM layers freely reconfigure as a function of the task context, areas like MT and V4 appear dedicated to processing motion and color, respectively. While MT and V4 do exhibit stretching (e.g., Fig. 4), they appear less flexible and do not reconfigure to the same extent as prefrontal cortex. Accordingly, V4 shows a decent correspondence with LSTM layers when color is relevant, whereas MT does when motion is relevant. ” has been added.

Page 9, Line 188, “To quantify this relationship, we considered how the RSA-based correlation between brain region (V4 or MT) and model-layer activity varied with task context. We performed a three-way repeated ANCOVA with fixed factors of region (MT or V4), task (motion or color relevant) and layer (1-6, a continuous variable) and a random effect (4 simulations). The interaction between region (MT or V4) and task (color or motion) was $F_{1,85}=670.5$, $p<0.0001$ (Supplementary Table S17), indicating that MT aligned more with the LSTM when motion was relevant whereas V4 did when color was relevant.” has been added.

11. In Fig. 2c and d, asterisks are shown on ISI and SPIKE, but the type of statistical tests conducted was not clear. Please add details about this point.

Thank you, addressed. We have added “A two-tailed test was conducted to determine whether the correlation is different from 0, with statistical significance noted.” in Fig.2’s caption.

12. On lines 118-123, in Fig. 3a and c, and Extended Data Tables 1 and 2, it was not clear which timing of data points was used for the analysis and for generating figures and tables (entire 250 ms stimulus period?). Please provide sufficient information to understand the procedures for the data analyses correctly.

Thank you, addressed. We have detailed the time bins used in respective captions.

13. The authors did not clearly describe how data samples from multiple neurons/electrodes are summarized for producing results of individual brain areas (e.g., Fig. 4a and Extended Data Fig. 3) and lobes (Extended Data Fig. 4). Please provide details on this point.

Yes, as discussed above, our initial draft was confusing on this point, and we apologize. The details are as follows:

Page 13, Line 317, “Rate coding considers spike count within a time bin and represents each stimulus as a n_r -dimensional neural firing rate vector where n_r is the number of recording sites” has been added.

Supplement, Page 3, “To construct rate-coding-based RDM, we first represent each stimulus as a n_r -dimensional vector, where n_r is the number of recording sites. Each element of the vector corresponds to the firing rate of a specific neuron/electrode in a time bin. To maintain uniform dimensionality for calculating dissimilarity between

stimulus vectors, we used 30 recording sites per brain region, resulting in a total of $n_r=180$ recording sites (sites with no recordings during a trial were excluded from the analysis” has been added.

14. On lines 118-121 and in Fig. 3a and b, the information about which LSTM layer was used for the analysis is missing. Please provide this information.

Thank you, corrected. Additional plots are now also available in the Supplement of by model layer and individual brain regions (see Figures S8-11). Figure 3 is intended to provide an overview.

15. In Fig. 4b, the authors only showed the combined modulation of attention, but to make the results more transparent, I recommend that the authors additionally show the attention weights for both motion and color tasks as in Fig. 4a.

We have completely revamped Figure 4 as we have improved the model-based analysis. We now break down attention weights in detail and it is clearer.

16. Regarding the readability of the entire manuscript, I occasionally encountered difficulty in finding the correspondence between the descriptions in the main text and the figures (including main figures, Extended Data Figs, and Supplemental Figures). For instance, the authors did not refer to Figure 4b and Extended Data Fig. 2 elsewhere. Therefore, I recommend that the authors thoroughly review the manuscript and include appropriate references to all figures. This will help readers easily identify the connections between statements and results throughout the text.

We have worked on the readability of the manuscript and think it should now be clearer. There is also a consistent approach to our statistics that should make analyses easier to follow. However, there are now so many supplemental figures and tables that I am not sure it would be ideal to refer to all from the main text. Again, there is a consistent logic where results are broken down in finer detail in the Supplement, so we hope it will be straightforward for readers who really want to dig into the details.

1.Çukur, T., Nishimoto, S., Huth, A. G. & Gallant, J. L. Attention during natural vision warps semantic representation across the human brain. *Nat Neurosci* 16, 763–770 (2013).

2.Yamins, D. L. K. et al. Performance-optimized hierarchical models predict neural responses in higher visual cortex. *Proceedings of the National Academy of Sciences* 111, 8619–8624 (2014).

3.Kell, A. J. E., Yamins, D. L. K., Shook, E. N., Norman-Haignere, S. V. & McDermott, J. H. A Task-Optimized Neural Network Replicates Human Auditory Behavior, Predicts Brain Responses, and Reveals a Cortical Processing Hierarchy. *Neuron* 98, 630-644.e16 (2018).

Reviewer #2 (Remarks on code availability):

The authors provided code for replicating their findings, primarily focusing on the visualization of the main results (demo code) with accompanying instructions on how to run the program. Additionally, they included code for the neural data preprocessing and LSTM model training but with fewer comments. The code for visualizing the main results only covers half of the results and does not include the analysis for presenting results from the CNN_LSTM model. In my view, considering the comparable importance of the CNN_LSTM section with the brain data section, it would be beneficial to include code for the CNN_LSTM section as well. While the requirements for running the code are provided, there are no instructions on how to install and set up the environment. Consequently, it may be challenging for novices to execute the entire code, except for the visualization part.

Thank you for noting this. These materials and code are now provided.

Finally, thank you for your careful reading which led to numerous improvements in the manuscript.

Reviewer #3 (Remarks to the Author):

Review for Adaptive stretching of representations across brain regions and deep learning model layers

I have now had time for an in-depth review of paper “Adaptive stretching of representations across brain regions and deep learning model layers”. The paper presents a reanalysis of an existing dataset to answer a timely and interesting question: how does the brain dynamically adapt to the ever-changing demands as it goes from one task to another? To answer this question, the authors contrast brain responses in two different task settings based on spike timing information, as well as by training an LSTM model. They find that many brain areas can adapt their coding to the task - with the exception of V4 and MT, which both show attentional affects limited to their original

coding preference (here colour and motion). The paper is well written and provides an important first datapoint that will spark further work in a direction that I am excited about. Yet, I must admit that the rather short submission made the impression as if the few analyses provided represent a starting point rather than a full-fledged paper. Below, I suggest a few plots/analyses that I think would complement the work and thereby allow the reader to better understand the underlying mechanisms/coding changes. I hope that the authors will find my comments useful and do take my suggestions as coming from a place of interest rather than unwarranted criticism.

Signed
Tim Kietzmann

Thank you for taking the time to supply a thoughtful review, which we acted on to improve the paper.

Major:

1. Self-limitation/reliance on ISI:

- I find it surprising that, if representational stretching happens throughout the brain, it would not be evident in the rate codes. Would the claim be that the rate code is virtually untouched while all attentional differences are to be found predominantly in timing? This feels unaligned with common understanding of how attention affects rates and hence requires further analyses (see below).

Yes, as referenced above, we now include an analysis of rate information and stretching. As you would expect, there is some stretching, but, likely also as you would expect given our preliminary analyses that selected ISI as our focal measure, the effects are not as robust.

“For comparison, rate coding results are reported in Supplementary Figs. S9-10 and Table S13.”

To guide you, in our new ANOVA analyses, the interaction term is key to assessing stretching.

The modifications are as follows:

Page 6, Line 132, “We assessed this stretching effect at the whole brain an regional level by two-way repeated measures ANOVA with fixed factors of task (color or motion

relevant) and mismatching dimension (color or motion mismatch) with a random effect for the monkey subjects. Stretching is consistent with an interaction such that dissimilarity is greatest for pairs mismatching on the dimension (color or motion) that is task relevant. This interaction was significant in the whole brain analysis, as well for analyses of PFC, FEF, LIP (see Supplementary Fig. S8 and Table S12). For comparison, rate coding results are reported in Supplementary Figs. S9-10 and Table S13.” has been added.

- Second to the previous point, ISI was selected as it best mapped (via RSA) to the experimental design. On these grounds, only ISI is used to test for effects of dimensional stretching from thereon. Yet, changes across tasks could also have been tested for with the other measures to drive home the point that in fact, nothing else changes. In fact, I am unsure whether the experimenter-intended-based RDM (distances in design space) reliably map onto actual coding dimensions in the various brain regions. If this were not the case, then relying on a measurement that best aligns the data and model RDM will be misguided. The model-free approach can easily be applied to all distance measures, including spike timing and rate code based measures. For us to understand how the brain reconfigures itself, we cannot afford to not look at other measurements, including rate-based codes.

Prompted by you and other reviewers, we have totally revamped our statistical analyses, including the model-free one and a new more powerful model-based one that leverages all the data and compares to a reasonable baseline model that can reflect the biases of each monkey. We do still focus on ISI because it is clearly the best similarity measure given our data. Rather than a weakness, we believe this extra step of selecting a similarity measure is a strength rather than arbitrarily selecting one as in most work or featuring many making the results difficult to follow. Importantly, the similarity measure was selected in a manner orthogonal to our hypothesis.

- Expanding on the above, ISI task stretching is estimated via univariate differences (a t-statistic). Why not look at patterns, too? Pattern differences are very powerful and are widely investigated in visual coding. Reversely, why not report univariate and multivariate differences for rate codes?

Now, the model-free analysis is done by a more encompassing ANOVA that I hope satisfies the reviewer. We do look at patterns of similarity in this analysis, though not using RSA. We see whether pairs of items matching on color or motion become more or less dissimilar depending on the task (color or motion based). We find this analysis intuitive, but not ideal as it does not leverage all the data, nor provide model-based

estimates with interpretable parameters. The revamped RSA analysis does just this and finds effects in all brain regions.

The details are as follows:

(a) We have included statistical testing towards dimensional stretching on 24 pairs using independent data among subjects. A two-way repeated measures ANOVA with fixed factors (motion/color mismatch and motion/color task) and a random effect (subject) was conducted on 24 stimulus pairs (see Table S12-13 and Figures S8-10). For comparison, rate coding results are reported in Figures. S9-10 and Table S13. The revised text in the Result's section on Page 6, Line 132, as reported in Comment 1.

(b) We have also included statistical testing towards RSA analysis compared a model of stretching to a baseline model using independent data. We adopted 5-fold cross-validation per task and per monkey. A one-way repeated ANOVA was conducted to evaluate Spearman correlation score with a fixed factor of model (cognitive or baseline model) and a random effect for monkey subjects. The correlation scores are obtained from 5-fold cross-validation, while the attention weights are trained using 4 folds. For the cognitive model, these 4 folds are from the same task context, whereas for the baseline model, 2 folds are from the color task and 2 folds are from the motion task. A stronger correlation score was observed and found significant across all brain regions (Table S15).

In summary, I think that all distance measures need to be tested univariately and multivariately to get a full picture as to what is happening. A reliance on univariate ISI, only based on RDM alignment is, in my eyes, not sufficient to describe the whole picture this exciting data has to offer.

We think the Reviewer will appreciate how the two stretching analyses, applied to every brain region and model layer, complement each other. Personally, we put greater weight in the model-based RSA analysis because it seems more sensitive (using more of the data) and provides interpretable fitted parameters (e.g., the attention parameter).

2. Presentation of (ISI) results and elevated baselines:

- Figure 2 does not mention which brain area is shown (same for figure 3). I expect an average across all areas? The supplemental materials show effects for all areas and here, the effects development over time are often smaller than in the whole-brain average show in the main figure. When taking into account that areas MT and V4 are (i) uniquely positioned to code for the experimental dimensions and (ii) show quite different

attentional effects to all other areas, then plotting an average across everything becomes questionable.

Yes, we took the strategy of making the figures understandable in the main text, but in conducting full analyses and plotting results for every brain region in the supplement (see Figures S6, and S8-9). We do now report the pattern of results for the supplemental analyses in the main text so that the reader is less likely to be misled.

- More importantly, the results of ISI are, more or less across the board, significantly higher than chance already during stimulus onset. Why is this? How long into the past and future does the effect go? I think the authors need to show a much longer timeframe (from say -200ms to 400ms) and understand what is going on. Why would the experimenter-intended-based RDM already be significantly correlated before stimulus information arrives? If the stimulus sequence is random, this should not be happening and I am worried that this points to an uncontrolled confound.

This comment was echoed by a previous reviewer and is addressed above. To quote from above, "Thank you for this sensible suggestion; it's always good to do these sanity checks. Fortunately, these effects are driven by the stimulus. The correlations were high at time 0 because that time bin starts at 0 s and continues to .05 s. We now plot Figure 2 starting before 0 s so the ramp up is clear. Figure S4 takes this further to show more of the ramp up and ramp down."

3. Figure 3 - main result

- The main figure of the paper is Figure 3 (the data used for the graphic remains unknown again, but I am hoping that I did not miss it somehow). This plot focuses on the effect of stretching. In this case, an average across brain regions really misleading, because regions V4 and MT behave so very differently from the others. Think of it as an interaction effect that prohibits an easy analysis of the main effect. At the very least, the plots should be made for all regions separately. Same goes for the timecourse in Figure 3B.

We agree in principle. Figure 3 is still all recording sites, but we now conduct analyses separately for each brain region and report which ones show effects in the main text and provide plots for all regions in the Supplement (see Figures S8-9). Interestingly, the model-based analysis that follows finds stretching in all regions (Figure 4). As mentioned, this analysis of stimulus pairs, while intuitive, does not leverage all the data so only the "strongest" stretchers come through.

- Please note furthermore that the effect size (e.g. condition differences) are more or less constant across the whole time period, and differences already appear at 0ms after stimulus onset (note that dissimilarity goes up over time, but the relevant difference between the curves are not changing). Is this due to a confound, or an aspect of the experimental design (as the cue was shown earlier)? If this is due to the cue, then the whole time course from the cue onwards should be presented.

I hope the previous discussion and changes made to address the time bin issue handle this comment. There is no confound with the cue or issue with timing being misaligned, etc. Yes, it does look like the absolute difference between curves is somewhat maintained.

4. Cognitive Modelling

- I do not understand the rationale behind using the cognitive model. What does it add other than another layer of complexity into the analysis? Why not plot the actual data across areas and conditions (as in, differences between activations, pattern or univariate, in the contrasted task conditions across time in the various regions)? This should reveal the same patterns and would much more informative.

We took this comment to heart and have introduced a new model-based RSA analysis paired with ANOVA that compares the cross-validated performance of the cognitive model to a very reasonable baseline model that can reflect the biases of each monkey. We applied this analysis to every brain region. This analysis seems ideal as it is interpretable (providing an attention stretching estimate under different conditions), gives an effect in terms of Spearman correlation, and provides a statistical test between the cognitive model and baseline model. We include all these measures and results in plots in the Supplement, which tell a coherent story. Every region shows significant stretching.

Here is the revised text in the Result's section:

Page 7, Line 144, "To assess how the six brain regions and LSTM model layers are modulated across dimensions in a quantitative manner, we conducted a cognitive model-based analysis (adapted from the generalized context model \cite{nosofsky1986attention}) using 5-fold cross-validation. Unlike the previous analysis, this model-based analysis uses all the data as opposed to select pairs of items and assumes a dimensional structure (color and motion) when estimating stretching along the relevant stimulus dimension. These aspects may make this analysis more sensitive and reveal stretching across all six brain regions, whereas the previous analysis only found significant effects in PFC, FEF, LIP." has been added.

Page 7, Line 152, “Specifically, we conducted a cognitive model that assigns w_m (ranging from 0 to 1) to the motion dimension and $1-w_m$ to the color dimension. By utilizing representational similarity analysis (RSA), we were able to optimize the cognitive model parameters to best match observed patterns of neural activity (see Methods), thereby inferring the relative attention of motion and color information in the task context.” has been added.

Page 8, Line 152, “We employed a 5-fold cross-validation procedure to train the attention weights w_m using four folds within the same task context. The fitted attention parameters (i.e., w_m on the motion dimension) of the cognitive model under two tasks (motion and color relevant) are illustrated in Fig. 4a. To assess the effectiveness of our cognitive model-based analysis, we compared it to a baseline model structurally identical to the cognitive model but fitted to two folds from the color task and two folds from the motion task. We evaluated the RSA-based correlation score using a one-way repeated ANOVA with a fixed factor of model (cognitive or baseline model) and a random effect for the monkey subjects. A stronger correlation was observed for the cognitive model and found to be statistically significant across all brain regions examined (whole brain: $F(1,37)=5.96$, $p=0.0195$; PFC: $F(1,37)=4.84$, $p=0.0342$; FEF: $F(1,37)=6.31$, $p=0.0165$; LIP: $F(1,37)=4.73$, $p=0.0362$; IT: $F(1,37)=11.8$, $p=0.0015$; MT: $F(1,37)=4.31$, $p=0.0449$; V4: $F(1,37)=10.2$, $p=0.0028$; see Fig. 4b and Supplementary Table S15). This finding suggests that the allocation of attention to motion or color features is adaptively modulated based on task contexts, and that this modulation is particularly pronounced in all brain areas.” has been added.

Minor:

- The LSTM results are shown in “seconds” whereas model time does not adhere to real-world time.

We now explain our conversion between frame rate and seconds. There are time steps in the model and we tried to “bend” them into something reasonable for comparison purposes.

The details are as follows:

Page 14, Line 356, “LSTM time estimated. Given that the LSTM processes stimulus images sequentially, it encompasses hidden states H that dynamically evolve within a trial (i.e., over the sequence of stimulus images in the trial). By capturing the hidden states from processing the first image (fixation) to the last image (moving colored

stimulus), we can derive n_m task-relevant RDMs within a trial, where n_m is the number of stimulus images within a trial. To align with sliding windows, we estimate LSTM time based on the frames of images. For instance, if selecting 60 frames per second, each image is displayed for $1/60$ of a second, allowing the input of each image to be considered as a timestamp. LSTM time at 0 is similar to the 0 to 0.05 s time window, encompassing the information of $0.05 \text{ s} / (1 \text{ s} / 60 \text{ images}) = 3$ input images. Note that with each time window (wide of 0.05 s) sliding by 0.01 s, we assume each sliding step of the window corresponds to one stimulus image. For the long time window (0 s to 0.25 s), dissimilarity for LSTM (Fig. \ref{stretching}c) is estimated as the averaged information of $0.25 \text{ s} / (1 \text{ s} / 60 \text{ images}) = 15$ input images.” has been added.

Reviewer #4 (Remarks to the Author):

In this paper the authors performed a novel analysis on a previously published data set (Siegel et al., 2015). They found evidence for “stretching” of task-relevant representations. They also simulated learning in the task using an ANN, which they then compared with the behavioral data results. The ANN also exhibited “stretching” as an emergent property.

I thought the reanalysis of the data set was interesting and well done. The authors tested the hypothesis that representations of task relevant features would be “stretched”, which they operationalized as greater dissimilarity in activity for mismatched pairs when the mismatched dimension was task relevant. They found that the representations were stretched for both tasks in all the areas they examined except V4 and MT which were specific to color and motion, respectively. I had some concerns detailed below about the theoretical framing of the results and integration with other research in the field, but those should be easily addressable.

Thank you for your thoughtful review, which helped improve our manuscript, and for the kind words.

Major concerns:

I was not persuaded by the relevance of the model results. The model gives some predicted results that are qualitatively roughly similar to the empirical results, for example in Figure 3. However, the similarity was not very close, and I was left unpersuaded that the model provided significant insights into the mechanisms involved. It is interesting that a model will develop stretching of representations even without that being programmed into the model, and that does seem to parallel the neural results, but

that nevertheless seems like a very general similarity to me. Is there any reason to think that such stretching wouldn't be a general feature of many computational models? In that case, what is the importance of the specific modeling approach taken?

I think this is a fair question. We tried to make the rationale clearer in the revision. The chief purpose of the LSTM model was to show that a fairly generic model that minimizes error will also stretch its representations in response to task cues. This is a kind of distortion that one might expect in a discriminative model, but not in all classes of models. Here's a related paper with some early thinking on this point, <https://doi.org/10.1177/0956797609357712>. Notice that most Bayesian models do not predict stretching. It might be that the Reviewer just has good intuitions about how these models tend to work that might not be shared with all readers.

One aspect we find exciting is that the model seems to build control structures to implement a kind of top-down attention based on task cue. Of course, those control structures are not built into the LSTM, but are learned from doing the task and minimizing error. It's interesting to ponder whether there is some parallel there with the human brain. Chiefly, the LSTM supports our argument that stretching is optimal for reducing error. Our new statistical analyses of the layers of these models and the time course of learning may add something as well. Finally, we believe it's good to offer a model that can do the task considered. In this case, the model does it from sequences of images (i.e., movies) and corrective feedback, which closely parallels what the monkeys experienced. One possibility is that others in future work elaborate our approach to make a model that, rather than functioning as a "blank slate" as in our contribution, serves as a model of different brain regions and their interactions.

We hope passages like this help clarify matters, Page 4, Line 91, "Rather than serving as a model of the brain, the ANN is intended to illustrate that a discriminative model will adaptively stretch its representations to minimize error \cite{davis_memory_2010}, which invites general comparisons to the brain's operation." has been added.

I had difficulty understanding the rationale and importance of the results depicted in Figure 5 relating layers of the ANN to the neural regions. The text (lines 139-144) argues that there is no straightforward correspondence between the layers and brain regions, and that the model is more flexible than brain regions and can fully reconfigure. Both of these statements sound to me like reasons that this analysis should be uninterpretable – if the ANN is highly flexible and can reconfigure to adapt to many things, isn't that an argument that it is NOT a good model for what is happening in the neural regions? I don't understand why one would hypothesize alignment (that is

correlation) between the layers specifically and the neural regions, and what that shows beyond the previous analyses in Figure 3.

Exactly! We tried to make this clearer. The brain faces constraints that the LSTM does not and may have areas that are modality bound. The LSTM is not intended as a model of the brain, but a demonstration that a reasonable machine learning model (a stand-in for a quasi-optimal model) also shows stretching, suggesting stretching may be a general property of adaptive systems. We bolstered this analysis with statistical tests. It's worth mentioning that our new model-based RSA analysis finds statistically significant stretching in V4 and MT, though not to the extent as is found in prefrontal regions.

This added passage might help,

This added passage might help, Page 4, Line 91, "Rather than serving as a model of the brain, the ANN is intended to illustrate that a discriminative model will adaptively stretch its representations to minimize error \cite{davis_memory_2010}, which invites general comparisons to the brain's operation."

The authors did not discuss the different roles of the areas they studied, with the exception of V4 and MT. I would like to see a greater integration of these results with the results from the Siegel et al paper and other relevant research. What are the implications of "stretching" of representations? How does the presence of stretching relate to the roles of FEF, LIP, etc in decision making? In particular, area IT has a very different role in decision making than FEF, LIP, and PFC, and the results in the Siegel et al paper show different patterns of activity in IT. The authors note in Figure 5 that IT differed from the other regions in alignment between the model and the neural activity, but they did not further discuss this result or the role of IT.

We tried to make our thinking clearer here. Yes, all these regions play different roles, but they also all seem to stretch along goal-relevant dimensions.

We now note this, Page 10, Line 200, "Despite differing functions ascribed to the brain regions considered, including cognitive control, object recognition, motion perception, and evidence accumulation, all regions showed adaptive stretching, to varying degrees."

Here is one way to understand this effect,

Page 4, Line 85, “While each brain region considered is associated with a different cognitive function \cite{siegel2015cortical}, we predict that stretching will broadly be observed. Akin to how object category signatures are observed in early visual areas as a result of activity in higher-level areas \cite{sexton2022reassessing}, task-relevant signals should propagate and representations should stretch in a system optimizing performance.”

Minor concerns:

The authors should define “stretching” of representations in a rigorous manner in the Introduction. What exactly is stretching? What does it indicate about representations?

We tried to better convey the definition throughout. Here are some examples:

Page 1, Line 36, “Attention can be viewed as {em stretching} representations along relevant dimensions \cite{kruschke1990alcove, nosofsky1986attention, love2004sustain}, which is reflected both in behavior and brain response \cite{mack2016dynamic, braunlich2019occipitotemporal, folstein2013category}.”

Page 1, Line 38, “While numerous attentional influences on neural activity have been observed\cite{connor1996responses,david2008attention,fix2010influence,ccukur2013attention}, we focus on the specific case of adaptive stretching along a goal-relevant stimulus dimension.”

Page 14, Line 367, “In two categorization tasks, dimensional stretching refers to the expansion of the perceived psychological difference beyond the physical difference between stimuli.”

We also tried to make the adaptive significance clearer. One example,

Page 4, Line 91, “the ANN is intended to illustrate that a discriminative model will adaptively stretch its representations to minimize error \cite{davis_memory_2010}, which invites general comparisons to the brain's operation.”

The authors often refer to their results as indicating “radical” reconfiguration of representations. How exactly do they define radical? Why do they think that their results show radical reconfiguration, rather than a more limited degree of reconfiguration – what makes it radical?

This is a good question. In addition to models that don't distort representations, one could imagine an alternative in which only one brain region stretches, and other regions

operate on that representation. Indeed, cognitive models are like this in which only their internal representation is affected by attention but encoding, decision procedures, etc. are unaffected. Instead, we find something more radical in that there is evidence (to varying degrees) that every brain region stretches along the goal-relevant dimension.

The authors should add a precise but brief definition of ISI and SPIKE and how they differ to the main text so that a reader can understand the importance of the result without having to go to the methods or supplementary materials.

Thank you for this suggestion. In the revision, we have added a brief definition in the main text, “For some time interval, rate coding measures sum the spikes at each recording site to calculate distances. Spike timing within the interval is inconsequential. For example, the Euclidean distance between two items involving five recording sites is simply the Euclidean distance between the two five-dimensional vectors where each vector entry is the sum of the spikes at a site. In contrast, rather than simply sum across the temporal interval, spike timing measures take into account temporal information within the interval, such as the timing between spikes for two items at a site, pooling these distances across recording sites for an overall measure of distance.” on Page 4, Line 106.

In the Supplement, Page 3, Sec. II, we have detailed the mathematical definition of neural coding measures and how we applied them in our cases.

Figure 3: It is difficult to remember what the condition names in the legend for parts b and d mean – could more meaningful phrases or abbreviations be used?

We got rid of our previous notation that was probably overly complicated and have relabeled the legend in a way that we think is more intuitive. Thanks for the prompt to do this. In the revised Figure 3, specifically, we used the triangle and square to denote the motion and color task, respectively, and then used the blue and orange line to indicate item pairs mismatching in motion and color, for simplicity.

Figure 5: The authors use a human anatomical model on which they superimposed the areas from the macaque study. This is problematic for two reasons. First, there were no human subjects in this study, so why use a human model? It is misleading. Second, and more important, areas MT, IT, and V4 are in different locations in the human brain than in the macaque so the figure is inaccurate. V4 for example in humans is much more ventral, as is IT.

Thank you for catching this. We have corrected the figure.

REVIEWER COMMENTS

Reviewer #1 (Remarks to the Author):

The current manuscript is much improved over the original submission. A lot of missing details have been filled in and additional statistical support provided.

I was very happy to see that the second reviewer provided many very specific and detailed suggestions for how the paper could be improved. Many of the concerns of the second reviewer were consistent with mine.

The paper still has some issues and there are some missing pieces and ad hoc methods, but the general conclusions are consistent with what is already known about attention in the brain from prior studies (including especially studies from the primate lab that supplied these data), so I expect that the main results will hold even after further revision.

We are delighted to see that the Reviewer appreciated the solidness and revisions of our work. To clarify and improve the significance of our results, we revised the manuscript extensively according to the Reviewer's insightful comments, as detailed next.

My suggestions to improve the paper before publication are as follows:

The use of RSA to analyze spike timing is an interesting idea, and it might be a nice addition to the toolbox. However, the use of this novel method imposes a responsibility on the authors to verify the veracity of the method, and to provide some information about the underlying assumptions and the boundary conditions. Given the current statistical crisis in psychology and biology its not a good idea to use analyses that aren't fully vetted. The most appropriate thing to do here would be to prove the analysis, but that might be complicated and slow, and it might require working with a statistician. So in lieu of that a bare minimum would be to show that the method is valid in simulation. I suggest that the authors do these simulations and report them in the supplementary materials, as such an effort would increase confidence in the results.

We agree with the Reviewer's point that applying RSA to analyze spiking timing is an interesting idea. To put our results in context, note that Ref. [37] (Kriegeskorte et al., 2008) studied RSA as a method to bridge the gap between various branches of neuroscience, focusing on fMRI data to demonstrate RSA's utility. Subsequent developments have extended RSA applications to EEG and MEG data, please see Refs. [38-40] (Kriegeskorte et al., 2009, Su et al., 2012, Su et al., 2014). Fortunately,

RSA toolbox has been developed and now supports data analysis across multiple modalities, including human fMRI, EEG, MEG, as well as monkey fMRI and electrophysiology (available at <https://www.mrc-cbu.cam.ac.uk/methods-and-resources/toolboxes/>). RSA is well established in neuroscience research. Yet, we realized that we lacked answers to several fundamental questions: Which features of brain spikes data (e.g., timing or firing rate) are most aligned with behavioral data? How does the underlying strategy in the brain compare to the performance of artificial neural networks (ANNs) trained on the same tasks, particularly in the absence of explicit attentional mechanisms?

One contribution of our paper, by virtue of being open minded about what measure of neural similarity is operable in the dataset, is that we offer some answers to these basic questions. Indeed, we show that spike timing measures exhibit a stronger correlation with the experimenter-defined stimulus setting. Moreover, even in modality-constrained regions such as MT and V4, we observe stretching effects, yet there appear to be inherent limits to this flexibility. As discussed in the manuscript, our comparative analysis between brain and ANN model have direct implications into how biological systems either benefit from or are constrained in their ability to localize perceptual processing across different stimulus dimensions.

To address the question raised by the Reviewer, we conducted the permutation test by shuffling spiking timing per trial by 1000 dependent runs. We employed Kolmogorov-Smirnov test (KS test) to compare the distributions of RSA score between empirical and permutation data and controlled for multiple comparisons using false discovery rate (FDR) correction. As shown in Table S17 (Page 22 in Supplementary Information), all comparisons confirm that the false discovery rate remained below 5%, indicating significant differences between empirical spike timing and chance-level distributions across both timing measures (ISI and SPIKE) and task contexts (motion and color).

In the revised manuscript, we also added supporting references accordingly:
Line 102: “One initial question is which measure of neural similarity is most aligned with the neural recordings \cite{bobadilla2020measures}. To address this question, we applied representational similarity analysis (RSA) \cite{kriegeskorte2008representational}, a method that computes and compares representational dissimilarity matrices (RDMs) to characterize the information encoded in brain data across various modalities including electrophysiology recording \cite{su2012estimating, kriegeskorte2009relating, su2014mapping}, as well as in computational models. Here we used RSA to determine how spiking neural activity relates to behavioral data (250 ms period data was used as our default setting, as indicated in Fig. \ref{model}b). Specifically, by assessing various neural similarity measures, we determined which one maximized representational similarity with the experimenter-defined stimulus coordinates of color and motion.”

It would be extremely helpful if the "cognitive model" were explained better in the main text. The term "cognitive model" is extremely general and its therefore a bit unclear on its face. I had to dig through the methods to figure out what was going on. The paper would be easier to read if some of the information given in the methods was moved to (or repeated in) the results.

We wish to thank the Reviewer for this prompt. In the revised manuscript, we have described the cognitive model in the Results:

Line 154: "Here we hypothesize that attention weights are allocated to task-relevant and task-irrelevant dimensions and determine these weights by maximizing the correlation between RDMs from each fold of brain data and the experimenter-defined stimulus coordinates for color and motion (see Methods for formula details). "

Line 163: "By optimizing the cognitive model parameters to best match observed stimulus coordinates, we inferred the relative attention of motion and color information in the task context. Here we employed a 5-fold cross-validation procedure to train the attention weights w_m where 4 folds were used for training, and the remaining fold was used for RSA-based correlation score within the same task context per subject. We observed that more attention parameters (i.e., w_m on the motion dimension) are placed on the motion dimension when motion is relevant (Fig. 4a)."

I suggest that the authors try to improve the figure captions further. Some of them (e.g., Figure 4a) are difficult to understand as written.

We thank the Reviewer for this thoughtful suggestion. We have revised the figure captions as follows accordingly:

Figure 4a: "The allocation of attention (estimated by fitting a cognitive model) in the whole brain and for the six brain regions is shown. The bars represent the averaged attention weights obtained by fitting the cognitive model using 5-fold cross-validation with weights trained on four folds within the same task context per subject. The error bars indicate the standard deviation, capturing the variability across 20 iterations, which were conducted across two task contexts and two subjects. "

Figure 4b: "Representational similarity analyses were conducted that Spearman correlated a model's estimated similarity structure with that of the brain data for the 20 held-out validation folds. The baseline model was identical to the cognitive model, but was fitted to two folds from the color task and two folds from the motion task rather than four folds from the relevant task. The cognitive model is better aligned with the brain data in all cases, with higher correlation suggesting the presence of stretching in the neural representations."

Other more specific issues:

Abstract: "Despite lacking an explicit selective attention or other control mechanism, the model displayed task-relevant stretching as a consequence of error minimization, indicating that stretching is an adaptive strategy." Error minimization is always associated with some cost function. I think that in the context of ANN training, the cost function is analogous to some specific state of a selective attention or control mechanism. Given that, this statement should be revised for clarity.

We appreciate the Reviewer's astute observation regarding the implicit relationship between error minimization and selective attention mechanisms. In response, we have revised the sentence for clarity as follows:

Abstract: "Despite lacking an **explicit** selective attention or other control mechanism, by minimizing error during learning, the model's representations stretched along task-relevant dimensions, indicating that stretching is an adaptive strategy."

Lines 48-49: "The ideal study to evaluate whether radical reconfiguration occurs would record from multiple brain sites while cueing the relevant dimension for a categorization decision on a trial-by-trial basis." Many people (including this reviewer) might argue that the ideal study would parametrically vary both the stimulus and attention in order to examine tuning curves of neurons under different states of attention. This study doesn't do that, so in that sense it is not ideal. However, this is just a quibble: I understand that all studies are limited in some way, there is no perfect study.

We appreciate the Reviewer's thoughtful comment. Perhaps the Reviewer's comment highlights that subtle differences in research goals can lead to differences in what is the "ideal" study. For our theoretical and modeling approaches, the present study is ideal or near ideal. We agree with the Reviewer that parametrizing both the stimulus and attention to examine tuning curves under different states of attention would also be valuable. Instead, we emphasize that attention can be viewed as a consequence of the task, which the Reviewer picks up in their next comment.

Lines 65-67: "While models of goal-directed attention contain dedicated control systems that selectively weight relevant dimensions [7, 9, 8, 23, 24], one possibility is that these control systems themselves can be learned by non human animals and ANNs, enabling them to reconfigure themselves in response to task cues." It is unclear precisely what this means in terms of the conventional language of control systems theory. One possibility is that this is just a (mis)wording issue, but it should be revised for clarity.

We thank the Reviewer for raising this important point. To clarify, we are not referring to control systems in the traditional engineering sense. Instead, we are discussing **attention control systems** within the context of cognitive neuroscience, which are mechanisms that modulate the allocation of attention in response to task demands. This is distinct from conventional control systems theory. We have revised the sentence for clarity: "While models of goal-directed attention contain dedicated **attention** control systems that selectively weight relevant dimensions [7, 9, 8, 23, 24], one possibility is that these systems themselves can be learned by non-human animals and ANNs, enabling them to reconfigure themselves in response to task cues.

Line 89-91: "Observing stretching in the ANN would provide a formal account of how control and top-down attentional mechanisms can arise from simply maximizing task performance, as opposed to relying on preordained mechanisms." This would not be a "formal account" in any way that an engineer or a statistician would use that word, especially if the LSTM is not interpreted but is taken at face value. This language should be revised or toned down.

We agree different communities may use terms in different ways. Our intent was to convey that observing stretching in the ANN offers insights into how control and top-down attentional mechanisms may emerge naturally from task performance optimization, rather than relying on predefined mechanisms.

We have revised the sentence as follows: "Observing stretching in the ANN would *demonstrate* how control and top-down attentional mechanisms can arise from simply maximizing task performance, as opposed to relying on preordained mechanisms.

Lines 127-129: "Stretching was assessed by considering item pairs that mismatched on one dimension and matched on the other. For example, we predict items mismatching on color and matching on motion should be more dissimilar when color is relevant than when motion is relevant." I thank the authors for clarifying the analyses, this brief summary lays out the main strategy of the analysis clearly.

We appreciate the Reviewer for the positive feedback on the clarity of our analysis strategy.

Lines 153-155: "By utilizing representational similarity analysis (RSA), we were able to optimize the cognitive model parameters to best match observed patterns of neural activity (see Methods), thereby inferring the relative attention of motion and

color information in the task context." Please clarify the steps taken to ensure that this analysis was not overfit. It seems that the description of the cognitive model is given in the methods, while the description of the cross-validation procedure is in the main results? It would be helpful to make sure that the cross-validation methods for this analysis (and all other analyses) are really clear in the methods.

We thank the Reviewer for pointing out the need for clarification regarding the cross-validation procedure. To address this, we have added two subsections “ k -fold cross-validation procedure” and “Baseline model” in the Methods section to explicitly describe the cross-validation procedure we used. The cross-validation procedure ensures that our analysis was not overfit by training the model on subsets of the data and validating it on held-out data, minimizing the risk of overfitting.

To be specific:

Line 423: “ k -fold cross-validation procedure.” To infer the relative attention of motion and color information in the task context, we optimize the cognitive model parameters to best match the observed stimulus coordinates. In this work, we employed the standard k -fold cross-validation approach. In this approach, the dataset is randomly divided into k equally sized partitions. The model is trained on $k-1$ folds and evaluated on the remaining fold. This process is repeated with k iterations, each iteration with a different fold used for evaluation. For our study, we use a 5-fold cross-validation procedure. To ensure that each subject's data is appropriately accounted for, this procedure is performed separately within each task context (motion and color) for each subject. Thus, this analysis results in a total of 20 iterations across the two task contexts and two subjects.”

Line 432: “Baseline model.” The baseline model is identical to the cognitive model but is trained using an equal number of folds from each task context. Specifically, the baseline model is fitted using two folds from the color task and two folds from the motion task, as opposed to the four folds from the relevant task context in the cognitive model. This baseline model is also trained following the 5-fold cross-validation procedure, providing a direct comparison to determine how well the cognitive model captured neural activity patterns using the 20 held-out validation folds.”

Nature Portfolio Reporting Summary: The data analysis portion refers to EEG, which is not included in this paper. I suggest that the authors look over the reporting summary closely to see if there are other errors or omissions.

Thank you for this valuable feedback. We have reviewed and corrected it.

Reviewer #1 (Remarks on code availability):

I am not as concerned about the code as I am about the ad hoc RSA analysis of spike timing. RSA is a common procedure, but the underlying assumptions and boundary conditions of the method have been poorly validated. In the original submission I asked for references to support the method, but I know of none so I'm not surprised that those references weren't provided. In lieu of that, I suggest that the authors undertake some simulations just so that we can be assured that their new methods are likely to deliver the claimed results without undue bias. I don't think that this is a big deal, though it will require a bit of work.

We appreciate the Reviewer's concern regarding the application of RSA to spike timing analysis. As the Reviewer noted, RSA is a widely used method. Previous studies (e.g., Kriegeskorte et al., 2008, 2009; Su et al., 2012, 2014) have extended RSA beyond fMRI to EEG, MEG, and electrophysiology, demonstrating its applicability across multiple data modalities. Furthermore, dedicated toolboxes have been developed to facilitate RSA analyses across these modalities.

To further ensure the robustness of our approach and address the Reviewer's potential concerns, we conducted permutation tests to assess the statistical significance of our results and control for false discovery rates. As mentioned above, our results help verify that our findings are not driven by spurious correlations and provide additional confidence in the validity of our RSA-based spike timing analysis.

In summary, we wish to thank the Reviewer for these thoughtful comments that have enhanced the clarity and robustness of our manuscript. We hope that the Reviewer will find the revised version suitable for the journal.

Reviewer #2 (Remarks to the Author):

The authors have made efforts to address the issues raised in my previous review, and some of these revisions have successfully resolved the concerns. However, there remain several aspects of the revised manuscript that are still unclear, and new questions have emerged based on the changes in the results. These points require further clarification.

Additionally, while I agree that some of Reviewer 1's criticisms were unnecessarily harsh and ungrounded, I share the overall concerns regarding the quality and readability of the manuscript. Specifically, the data used for the analysis (e.g., figures and statistical tests) is often unclear, with insufficient information provided for understanding the analytical conditions (e.g., time windows, brain regions, task contexts). Furthermore, although many results have been added to the supplementary materials in the revised manuscript, they appear to be presented in a

rather disjointed manner, without sufficient explanation of how each result supports the main claims of the study.

These issues seriously hinder readers from fully understanding the content and implications of the study, as well as from replicating it. Thus, the current version still presents significant issues in terms of clarity, which detracts from the overall quality of the work. As it stands, the current manuscript does not meet the standard expected for publication in Nature Communications. I acknowledge the potential value of the scientific contributions in this work; however, further revisions are necessary to improve both the readability and presentation, ensuring the content is accessible to a broader readership.

We are encouraged by the Reviewer's acknowledgement that the revisions we have made successfully addressed previous concerns. We fully agree with the Reviewer on the importance of ensuring clarity and reproducibility while maintaining conciseness in the main text and providing sufficient detail in Supplementary Information. In the revised manuscript, we have implemented additional analyses and refined our presentation to enhance the clarity and accessibility of both the data and code. We hope that these revisions effectively address the Reviewer's concerns.

1. The clarification of the results in Fig. 2 through the correction of the error in the rate coding analysis is commendable. Additionally, showing results over a longer time period in Fig. S4 is very helpful, as it provides a clearer understanding of the overall pattern. Given that this approach makes the data more comprehensible, it may be beneficial to present similar extended time period results for Fig. 2, Fig. 3, and Fig. S6-S11 as well. However, these results raise a new question. Specifically, in Fig. S4, the peak performance for all neural measures occurs around the 250-500 ms period. Despite this, the authors have conducted many analyses (e.g., region-wise analysis in Tables 3-11) using data from only the first 50 ms post-stimulus onset or have shown results only up to around 250 ms (e.g., Figs. 2 and 3), without providing a clear explanation for this choice. Given that relatively high correlations are also observed in the rate coding results around the peak period, it is possible that stretching effects might be detected if analyses are conducted over this time frame. This could call into question the validity of the claim made in the abstract that "spike timing was crucial to this code" (indeed, Fig. S9 seems to show). Therefore, it is recommended that the authors provide a rationale for the choice of time windows used in their analyses and discuss whether the claims made in the manuscript, including the assertion that spike timing is crucial, hold independently of the potentially arbitrary selection of time periods for analysis.

We appreciate the Reviewer's comments. Please note that the choice of the 250 ms period for the majority of our analyses was made a priori, based on prior knowledge

of the temporal dynamics of neural responses in our specific task context. This decision was not influenced by the data or any exploratory analyses, thereby avoiding potential biases (e.g., p-hacking). Nevertheless, prompted by the Reviewer's comments, we did find that stretching effects occurred in regions such as FEF, and LIP when analyzing with rate coding measures using 250 ms period data (as mentioned in the main text and Table S11 for details). We acknowledge that the Reviewer's perspective has merit: given that peak performance is observed in Fig. S4 around the 250-500 ms period, extending the time period for certain analyses may provide additional insights.

In response to the Reviewer's comment, we extended the time period for certain analyses to assess the potential influence of longer time periods on our findings. Indeed, for rate coding, we observed stretching effects in the whole brain and regions FEF and LIP using 500 ms period data (see Table S12). Importantly, for comparison between 250ms and 500ms period (Tables S9, 11 and Tables S10, 12), we found this extension did not qualitatively alter the key results or the main conclusions of our study.

To maintain the coherence of 250ms period (as indicated in Fig.1) as a priori in the manuscript and avoid redundancy in Supplementary Information, we have included the extended period stretching statistical tests, but attached the corresponding dissimilarity curves (Figure R1) for the extended 500 ms period for your reference.

Figure R1. We showed extended 500 ms period data for mismatching pairs measured by rate coding (Euclidean distance) across brain areas (50 ms sliding window).

We have revised our manuscript in the main text, Line 106: “Here we used RSA to determine how spiking neural activity relates to behavioral data (250 ms period data was used as our default setting, as indicated in Fig. 1b).”

2. In Fig. 2, the rate coding results have been improved and appear reasonably smoothed compared to the spike timing measures. However, the region-wise and lobe-wise rate coding results in Figs. S6 and S7 still show considerable variability, which appears unreasonable compared to the results in Fig. 2. As these figures only display results for the 0-0.25s time window, similar to the earlier version of Fig. 2, it may be beneficial to extend the range, including time before 0s and a longer time period, as in the current Fig. 2. This could aid in debugging and help verify the validity of the results.

We thank the Reviewer for this thoughtful comment. We fully agree that extending the range is beneficial for a more comprehensive presentation, as a narrower range may exaggerate variations. In the revised manuscript, we have updated the results to include a longer time period for correlation across regions, now presented in Figure S6 (also attached below as Figure R2).

Figure R2. Spike timing matters in each individual brain region with a 50 ms sliding window. Spike timing measures best capture the experimenter-designed coordinates in six brain regions (two-tailed t -test with Bonferroni correction; see Tables S3–8).

3. In the analysis using the cognitive model, while the authors modified the analytical procedure from the original manuscript, these changes were not reflected in the corresponding methods section. Specifically, there is no explanation of the baseline model, which was newly introduced in this revised version. To ensure that readers fully understand the rationale behind the methodology, it is recommended that the methods section be appropriately updated to include detailed explanations of the changes, including the newly added baseline model.

We thank the Reviewer for suggesting to strengthen our presentation of the baseline model. We have now updated to the Methods section, as well as the main text, to provide a detailed information of the introduced baseline model and the modifications made to the analytical procedure accordingly:

Line 163: “By optimizing the cognitive model parameters to best match observed stimulus coordinates, we inferred the relative attention of motion and color information in the task context. Here we employed a k -fold cross-validation procedure to train the attention weights w_m where 4 folds were used for training, and the remaining fold was used for RSA-based correlation score within the same task context per subject.”

Line 423: “ k -fold cross-validation procedure.”

To infer the relative attention of motion and color information in the task context, we optimize the cognitive model parameters to best match the observed stimulus coordinates. In this work, we employed the standard k -fold cross-validation approach. In this approach, the dataset is randomly divided into k equally sized partitions. The model is trained on $k-1$ folds and evaluated on the remaining fold. This process is repeated with k iterations, each iteration with a different fold used for evaluation. For our study, we use a 5 -fold cross-validation procedure. To ensure that each subject's data is appropriately accounted for, this procedure is performed separately within each task context (motion and color) for each subject. Thus, this analysis results in a total of 20 iterations across the two task contexts and two subjects.”

Line 432: “Baseline model.”

The baseline model is identical to the cognitive model but is trained using an equal number of folds from each task context. Specifically, the baseline model is fitted using two folds from the color task and two folds from the motion task, as opposed to the four folds from the relevant task context in the cognitive model. This baseline model is also trained following the 5 -fold cross-validation procedure, providing a direct comparison to determine how well the cognitive model captured neural activity patterns using the 20 held-out validation folds.”

4. Regarding the results in Fig. 4, the authors claimed that adaptive task-relevant stretching was observed at all recording sites, including MT and V4, in the revised manuscript. However, in Fig. 4a, it appears that MT and V4, as well as IT, did not exhibit weight changes according to the task context. While the authors argued that the higher correlation of the cognitive model compared to the baseline model in Fig. 4b indicates adaptive modulation depending on the task context, this alone does not fully support their claim. In particular, the baseline model for V4 showed correlations close to chance level (near zero), raising questions about whether the fitting is working properly and suggesting potential issues with the validity of the method. It is recommended that the authors provide an appropriate discussion on these subtle differences in the results and revise their claims to better reflect the observed data.

We thank the Reviewer for this insightful comment. Indeed, regions MT and V4 exhibit weaker weight changes across task contexts, which aligns with our understanding of their functional roles. We acknowledge that if the correlation versus weight relationship forms a flatter landscape, the model fitting might face limitations, such as difficulty in selecting the optimal weights. However, compared with the baseline model, we did not aim to focus on selecting the optimal weights but rather to select the optimal correlation. Using a 5-fold cross-validation procedure, we did find the selective cognitive model outperforms the baseline for both MT and V4.

In the revised manuscript, we clarify that the baseline model is fitted using a combined dataset from both motion and color tasks. The lower correlation observed in V4 for the baseline model may indicate that task modulation in this region is relatively weak, as compared to higher-order cognitive areas such as PFC. This is consistent with the understanding that V4, while essential for processing visual information, may not be as sensitive to task demands as regions like PFC, which are more directly engaged in modulating task-relevant information. Despite the weaker modulation in V4, the cognitive model still leads to higher scores in both MT and V4 compared to the baseline, indicating that stretching occurs even in regions with relatively weaker task-dependent adjustments.

To address these concerns, we have adjusted our claims accordingly: “Line 177: While significant, MT and V4 exhibit weaker weight changes across tasks (Fig. 4a), which is consistent with our understanding of the function of these areas.”

5. On Page 5, lines 112-114, as well as in Fig. 2, Fig. S5, and Tables S3-S11, the authors presented the results of statistical tests. However, it is unclear how data from the two different task contexts (motion and color) were utilized in the analysis. Additionally, the exact stimulus timing used in the analysis is not clearly specified in the main text (lines 112-114; is it a 50 ms time bin beginning at 0 seconds?). Please

provide sufficient information to clarify these points and ensure the procedures are fully understood.

We thank the Reviewer for this suggestion. To address these concerns, we would like to clarify: (I), To be simplicity, we averaged the correlation between motion and color tasks to evaluate overall performance, and this approach was applied to all pairwise *t*-test comparisons in our study (e.g., Tables S1-S8). (II), Data from the 250 ms period was used as our default setting (as indicated in Fig. 1b).

We have revised the manuscript to clearly specify these details:

Line 106: “Here we used RSA to determine how spiking neural activity relates to behavioral data (250 ms period data was used as our default setting, as indicated in Fig. 1b).”

In Supplementary information:

Table S1: “Notice we averaged the correlation between motion and color tasks to evaluate overall performance, and this approach was applied in subsequent pairwise *t*-test comparisons.”

6. In Fig. S4, the authors used the term “session,” but its definition is not provided, making the meaning unclear. For example, does “session” refer to the same unit as a “trial,” or is it a separate unit used in the monkey experiments that encompasses multiple trials? Please clarify the definition of “session.” If “session” differs from “trial,” it may be more appropriate to redraw Fig. S4 based on variances across trials. This would better support the significance and reproducibility of the results. Additionally, in the legend of Fig. 2, it is not clear how the 95% confidence interval was calculated. Please provide details regarding the basis for the confidence intervals and specify what was used to compute them (e.g., across trials).

We wish to thank the Reviewer for pointing this out. Before addressing the question regarding the session, we would like to recap some information: In this study, we employ RSA-based analysis, where RSA-based correlation score is evaluated by comparing the correlation of representational dissimilarity matrices (RDMs). In our case, constructing an RDM from one trial is not feasible, instead, a minimum of 16 item trials is required to form a 16×16 RDM.

As the monkey data was recorded over a long duration, electrode replacements occurred, leading to variations in the number of electrodes across trials. In the original experimental setting (Siegel et al., 2015), a session consists of multiple trials with fixed electrode positions, with replacements occurring between sessions. To minimize potential variability introduced by electrode replacements, we constructed RDMs within each session using data from the same electrode placements and then evaluated correlations across sessions.

To address the Reviewer's concern, we have now included this original experimental information in Methods:

Line 363 “\textbf{RDM based on neural data.} In the original experimental setup \cite{siegel2015cortical}, as the monkey data was recorded over a long duration, electrode replacements occurred, leading to variations in the number of electrodes across trials. Each session consisted of multiple trials (~1000–1900 trials per session, 53 sessions in total), with electrodes repositioned and microdrive assemblies removed after each session. We apply RSA by constructing RDMs from random pairwise trials within a session, as only one stimulus was presented per trial. We compute pairwise trial correlations and average them within sessions, and report confidence intervals across sessions (e.g., Fig. 2). For rate coding, each stimulus was represented as a high-dimensional vector derived from neural activity recorded during that trial. To ensure consistency across trials and brain regions, we randomly selected 30 recording sites per region, maintaining uniformity across all stimuli.”

Also, we have deposited source data with this paper (please see Data Availability).

7. In Fig. S15 and S16, the authors mentioned that the earlier and later model layers reach their highest correlation earlier in the motion and color contexts, respectively. However, they did not provide any discussion regarding the implications of this finding. As the significance of this distinctive trend remains unclear, it is recommended that the authors include a discussion on what these results suggest and how they contribute to the overall understanding of the analysis.

We thank the Reviewer for this sensible observation. We do agree that the observed trend in Fig. S14 and S15, where earlier model layers reach their highest correlation earlier in the motion context, while later layers do so earlier in the color context. Indeed, this trend aligns with the principles of hierarchical feature representation over time. This suggests a temporal aspect to feature specialization, complementing previous findings. For instance, Allen-Zhu et al. (2023) proposed that later layers in neural networks refine earlier-learned features, while Fel et al. (2024) found that simpler features are learned earlier in the network and training process, whereas more complex features, such as color, emerge in later layers.

In response to the Reviewer's suggestion, we have now added a discussion on the implications of this phenomenon as follows:

Line 228: “However, feature specialization can be seen in the LSTM's dynamics. Specifically, we observed that earlier model layers reach their peak correlation more rapidly in the motion context, whereas later layers achieve their highest correlation earlier in the color context (Supplementary Figs. S14-15). In tasks with richer stimuli,

perhaps related models would display feature hierarchies not just across layers \cite{fel2024understanding, allen2023backward} but also in time.”

8. Regarding the readability, titles of some figures and tables lack sufficient information, making it difficult to understand what they represent and how they differ from other results. For instance, although Fig. S5 has the same format as Fig. 2a and b, with only different time bins, the title of Fig. S5 is identical to that of the entire Fig. 2. Furthermore, the titles and legends of the figures and tables often lack sufficient detail to help readers quickly grasp their content (e.g., analyzed time bins and brain regions). Additionally, there are some incorrect titles in the supplementary tables (e.g., Tables S1 and S2 are not ANOVA results, despite being listed under Section III.2 “ANOVA analysis to determine the best neural measure”). It is recommended that the authors carefully review all figure and table titles and revise them to be more specific and reflective of the content. This will help readers easily understand the data being presented and clearly distinguish between the figures and tables.

We thank the Reviewer for drawing our attention to the lack of clarity in the captions.

To remedy it, we have revised our manuscript accordingly:

(I), We revised the caption of Fig. S5 as “Spike timing measures best capture the experimenter intended coordinates when using a 20 ms sliding window. We found ISI measure surpasses other measures in a 20 ms sliding window (A one-way ANOVA analysis is conducted to show the significance of different types of measures from stimulus onset, $F(4, 105) = 180.79$, $p < .0001$). Two-tailed t -test with Bonferroni correction was shown in Table S2.) A 50 ms time window is shown in the main text, indicating the length of time window chosen does not affect the advantage of timing measures in assessing spike trains in such categorization tasks.”

(II), We changed Section III.2 “ANOVA analysis to determine the best neural measure” to Section III.2 “Identify the best corresponding neural measure”.

(III), We have reorganized the layout of Supplementary Information to group the same analyses together, reducing redundancy while retaining necessary details. Additionally, we added data range information in the top-left corner of the tables for clarity.

9. On page 3 of the Supplementary Information, in the section “II.1 Rate Coding,” the authors mention that they used 30 recording sites per brain region but do not explain the criteria for selecting these sites. To ensure that the analysis is replicable, please provide a detailed explanation of the selection criteria used for choosing the recording sites.

We thank the Reviewer for this suggestion. As mentioned in Point 6, the selection of recording sites per brain region was intended to ensure a balanced sample of neuronal activity across each region. As the monkey data was recorded over a long duration, electrode replacements occurred, leading to variation in the number of electrodes across trials. To ensure a consistent and fair evaluation, we selected 30 sites per region as a representative number. To facilitate the replication of our results, we have deposited source data with this paper (please see Data Availability). We hope this will address the Reviewer's concern regarding the reproducibility of our findings.

10. Regarding the comment 10 of my review, it seems that the authors' response does not fully address the point I raised. While the authors mention that they provided inferential statistics, my comment was specifically about the interpretation of the results supporting reconfiguration. I kindly ask the authors to revisit my original comment and respond appropriately to this concern.

We wish to thank the Reviewer for drawing our attention back to previous comment 10, attached here *“Comment 10. In Fig. 5, the authors compared RDMs of V4/MT and the LSTM model for each of the color and motion tasks to claim that “LSTM layers freely reconfigure as a function of the task context”. However, the observation that “For example, V4 shows a decent correspondence with LSTM layers when color is relevant, whereas MT does when motion is relevant.” (lines 142-143) can also occur when the LSTM-RDM was fixed, and V4-/MT-RDMs changed depending on the tasks. For rigorously testing the claim, it would be better to additionally show higher Spearman correlations between motion-task-fitted brain-derived-RDMs (including MT and other areas) and the motion-data-fitted LSTM-RDM than between motion-task-fitted brain-derived-RDMs and the color-data-fitted LSTM-RDM. Similarly, Spearman correlations between color-task-fitted brain-derived-RDMs and the color-data-fitted LSTM-RDM can be compared with those between color-task-fitted brain-derived-RDMs and the motion-data-fitted LSTM-RDM. If higher correlations were observed from this analysis, it would strengthen the support for the authors' claim.”*

We agree with the Reviewer that *“it would be better to show higher Spearman correlations between motion-task-fitted brain-derived-RDMs (including MT and other areas) and the motion-data-fitted LSTM-RDM than between motion-task-fitted brain-derived-RDMs and the color-data-fitted LSTM-RDM”*.

This is exactly what we have done and even further validated through rigorous ANOVA analysis, as detailed in the main text and Table S16. We added in our manuscript Line 200: “To quantify this relationship, we considered how the RSA-based correlation between brain region (V4 or MT) and model-layer activity varied with task context. We performed a three-way repeated ANCOVA with fixed

factors of region (MT or V4), task (motion or color relevant) and layer (1-6, a continuous variable) and a random effect (4 simulations). The interaction between region (MT or V4) and task (color or motion) was $F_{(1,85)}=670.5$, $p<0.0001$ (Supplementary Table S16), indicating that MT aligned more with the LSTM when motion was relevant whereas V4 did when color was relevant.”

We hope that we have sufficiently clarified our treatments and demonstrated the significant results for the Reviewer to reassess our work.

Reviewer #2 (Remarks on code availability):

The authors have improved the code provided for replicating their findings, addressing many of the concerns raised in my previous review. The updated code now includes more comprehensive coverage, particularly with additional sections for the CNN_LSTM model, which were previously missing. This enhancement ensures that both the brain data analysis and the CNN_LSTM section are adequately represented, reflecting the comparable importance of these parts of the study.

Moreover, the code is now better documented, with clearer instructions on how to run the programs and set up the necessary environment. This makes it much more accessible, even for users who may not have extensive technical expertise. Overall, the revisions have made the code richer and more user-friendly successfully addressing prior feedback.

We wish to thank the Reviewer for his/her very positive assessment of the code we provided. We fully agree with the Reviewer that detailed code is crucial for ensuring reproducibility.

In summary, we wish to thank the Reviewer for the interesting discussions summarized in the manuscript that their comments have led to. Following the Reviewer's patient recommendations, we have improved the clarity, reach, and accuracy of the manuscript. We hope the Reviewer will find the revised version appropriate for publication in *Nature Communications*.

Reviewer #3 (Remarks to the Author):

Thank you for engaging with my comments and for addressing many of them with this revision.

Thank you for recognizing our efforts and for your helpful suggestions. We have carefully revised the manuscript based on your additional comments and hope the changes enhance its presentation.

One comment that I 100% stand by, and that I think needs to still be addressed, is that the graphical display of the data of the individual ROIs should not be buried in the supplement (Figures 2 and 3). They may not be as pretty as the grand average, but especially in light of the authors' discussion on how the ROIs exhibit different effect sizes (see for example MT and V4), this becomes essential. In fact, the observation of different effect sizes alone could imply that a grand average does not precisely represent the data.

We greatly appreciate the Reviewer's suggestion to present the individual ROI results in the main text. We recognize that the variation in effect sizes across regions like MT and V4 is an important aspect of the analysis, and displaying these figures in the main text rather than SI could provide convenience for readers.

To address this, we have ensured that the statistical analysis and key information are now included in the main text, with references to the detailed data in SI. In line with the journal's format and length constraints, we have opted to place these detailed ROI figures in SI, while presenting the whole brain figure in the main text to enhance readability and clarity for the readers.

We understand the Reviewer's concern and are open to the possibility of including ROI figures in the main text if the Editor deems it appropriate given the formatting constraints of *Nature Communications*, perhaps by allowing us extra space for an additional figure in the main text.

Furthermore, I have read the description of the "cognitive model" multiple times and still struggle understanding what exactly was computed/fitted. Could the authors significantly expand this section, and ideally provide a graphical depiction of the model and what was scaled where/how? The main claim of the paper rests on this analysis, as some ROIs only show the stretching effects in this, perhaps more uncommon, analysis. Hence, a clear and detailed description will be of importance. Could the authors also comment on how this analysis is different from deriving attentional model predictions using RSA and fitting the RDMS to jointly model the neural data? This would perhaps be a more common approach (or, say, one that I and perhaps others would be more familiar with).

We thank the Reviewer for raising an important point regarding the clarity of our cognitive model. To better illustrate the cognitive model, we attached a simplified graphical depiction (Figure R3) for intuitive understanding.

Figure R3. Visual representations of attentional effect. The dissimilarities between stimuli pairs vary depending on the task context. For instance, stimulus $(1,1)$ and stimulus $(4,4)$ have equal feature distance between motion and color dimension, however, in color-relevant context, see the left and orange shape, there would be dimensional stretching on color dimension between stimulus $(1,1)$ and stimulus $(4,4)$, that is, in the psychology space, the attention towards color-relevant information is greater than that towards motion-relevant information, and vice versa in motion-relevant context.

We hypothesize that attention weights are allocated to task-relevant and task-irrelevant dimensions and determine the weights by maximizing the correlation between RDMs from neural data and the experimenter-defined stimulus coordinates for color and motion, which is formulated in Methods Eqs. 5 and 6. We fitted the task-relevant weight by maximizing the attentional RDMs (AE) with RDMs from neural data (Fig. 4a; neural data from the same task context) and evaluated its RSA-based correlation score (Fig. 4b). Importantly, we employed the cross-validation procedure here to prevent overfitting and introduced a baseline model (fitted weight from balanced/joint color and motion tasks) that can reflect the biases of each monkey. By providing a statistical test between the cognitive model and baseline model, each region shows significant stretching.

In the revised manuscript, we added detailed information for explanation in the main text accordingly:

Line 154: “Here we hypothesize that attention weights are allocated to task-relevant and task-irrelevant dimensions and determine these weights by maximizing the correlation between RDMs from each fold of brain data and the experimenter-defined stimulus coordinates for color and motion (see Methods for formula details).”

Line 163: “By optimizing the cognitive model parameters to best match observed stimulus coordinates, we inferred the relative attention of motion and color information in the task context. Here we employed a 5-fold cross-validation procedure to train the attention weights w_m where 4 folds were used for training, and the remaining fold was used for RSA-based correlation score within the same task context per subject.”

In the Methods section:

Line 423: “**5-fold cross-validation procedure.**”

To infer the relative attention of motion and color information in the task context, we optimize the cognitive model parameters to best match the observed stimulus coordinates. In this work, we employed the standard k -fold cross-validation approach. In this approach, the dataset is randomly divided into k equally sized partitions. The model is trained on $k-1$ folds and evaluated on the remaining fold. This process is repeated with k iterations, each iteration with a different fold used for evaluation. For our study, we use a 5-fold cross-validation procedure. To ensure that each subject's data is appropriately accounted for, this procedure is performed separately within each task context (motion and color) for each subject. Thus, this analysis results in a total of 20 iterations across the two task contexts and two subjects.”

Line 432: “**Baseline model.**”

The baseline model is identical to the cognitive model but is trained using an equal number of folds from each task context. Specifically, the baseline model is fitted using two folds from the color task and two folds from the motion task, as opposed to the four folds from the relevant task context in the cognitive model. This baseline model is also trained following the 5-fold cross-validation procedure, providing a direct comparison to determine how well the cognitive model captured neural activity patterns using the 20 held-out validation folds.”

We hope that the additional details and explanations have addressed the Reviewer's concern regarding the definition of the cognitive model and clarified the validity of our results related to the radically stretching observed in the brain.

Reviewer #4 (Remarks to the Author):

The authors have thoroughly revised their paper, including taking a new analysis approach. They also added additional background information and definitions that make the manuscript much easier to read and better convey the important take-home message of the work.

We thank the Reviewer for the very positive assessment of the manuscript. We are extremely encouraged by the Reviewer's appreciation of the clarity and significance of our results.

The concerns that I raised in the first round of review were responded to appropriately overall. I have just one more remaining minor concern. In the first round of review I questioned the use of the word "radically" as a modifier for how brain areas were reconfigured. In the response, the authors made it clear that what they mean by radical is that stretching occurs across multiple brain areas, not just within limited perceptual regions, and is spread across regions to a larger degree than would be predicted by cognitive models. I found the way they expressed this in the response letter to be persuasive. However, it is still not clear in the manuscript what is meant. I recommend that the authors add a brief definition of what they mean by radical to the introduction and discussion. I think that pointing out how stretching reconfigures activity across many brain regions and going into a bit more depth about the implications of this finding would actually enhance the power of their message to the reader.

We fully agree with the Reviewer that a brief review of radical stretching is necessary. In the revised manuscript, we have added the following information in the Discussion section:

Line 210: "We found support for the hypothesis that the brain radically re-configures itself to optimize performance by accentuating differences along task-relevant dimensions. Despite differing functions ascribed to the brain regions considered, including cognitive control, object recognition, motion perception, and evidence accumulation, all regions showed adaptive stretching, **to varying degrees.**"

We would like to thank the Reviewer for appreciating our important and clearly presented results. The Reviewer's comments helped us improve our manuscript, and along with the other Reviewers, we appreciate their engagement with our work.

REVIEWER COMMENTS

Reviewer #1 (Remarks to the Author):

This paper has improved substantially through the last two rounds of review.

We are pleased to see the Reviewer considered our paper acceptable for publication. We sincerely thank the Reviewer for their thoughtful comments and for recognizing the improvements made throughout the revision process.

I am still not fully satisfied with the RSA analysis, simply because as far as I know there are no foundational papers that formally prove the underlying statistical assumptions of this method. Without such proof it is impossible to know when a statistical test should be used or what biases it might produce in the results.

But as the authors note this has become a widely used tool nevertheless, and after all, we use many tools in neuroscience without fully understanding them. In any case the results reported here are broadly consistent with other reports of dynamic changes in representation during different tasks.

So in my opinion this paper is now acceptable for publication.

We appreciate the Reviewer's comment on the limitations of RSA. As noted, while a formal proof of its assumptions remains an open topic in the field, RSA has nevertheless become a widely accepted and practical tool for analyzing neural representations. To solidify our methods, we have made sure to interpret our RSA results with appropriate caution: SI, Sec. III 9 Additional results in our defined cases for "Permutation test comparing empirical and shuffled spike time distributions to validate the RSA analysis of spike timing." The results demonstrated that the false discovery rate (FDR) remained below 5%, thereby validating the robustness of our findings using RSA.

Once again, we are pleased to see the Reviewer considered our paper acceptable for publication. We thank the Reviewer for the constructive feedback and engaging discussion throughout the review process.

Reviewer #2 (Remarks to the Author):

The revised manuscript represents a substantial improvement over the original version. I appreciate the authors' efforts in addressing many of the concerns raised during the initial review. Below, I highlight a few points that, if further clarified or revised, could enhance the manuscript's informativeness and transparency for readers.

We are pleased that the revised manuscript have addressed many concerns of the Reviewer #2 and have made substantial improvements. Below, we respond to each point and outline

the corresponding revisions we have made in the revised manuscript to further refine clarity and transparency.

1. In response to my previous comment #1 regarding the analysis time window, the authors stated: “In our study, data from the 250 ms period was used as the primary analysis based on prior knowledge.” However, the nature of this “prior knowledge” remains unclear. I recommend that the authors either cite appropriate references or elaborate on the specific empirical or theoretical basis for this choice. If the decision was informed by preliminary data or prior empirical findings (e.g., in this or related paradigms), this should be explicitly stated.

Clarifying this rationale is particularly important given that stretching effects were observed in later time periods (as reported in Tables S9–S12), suggesting that alternative temporal windows may yield different conclusions. Such information would be valuable for guiding future studies in selecting appropriate time frames for RSA-based analyses.

Furthermore, while the authors included new supplementary analyses that explore extended time windows (Tables S9–S12), these results are currently presented without sufficient context or interpretation in the main text. It would be helpful if the authors could clearly explain the motivation for conducting these additional analyses, describe how they relate to the central claims of the study, and clarify whether—and in what ways—these extended analyses qualitatively alter or reinforce the main conclusions. While some of this reasoning is outlined in the rebuttal, it should be included in the main text to ensure completeness and ease of understanding for readers.

Thank you for this comment. To reiterate from the previous letter, we made our choice of the analysis window prior to conducting the data analyses. This decision was informed in part by previous work conducted by the senior author in collaboration with Nick Sexton analyzing monkey data (Sexton et al., *Sci. Adv.*, 2022), as well as by similar choices made in prior literature that employed a 250 ms window for analyzing monkey neural activity. In the revised manuscript, we have added the corresponding references: “Following previous studies (Sexton et al., *Sci. Adv.*, 2022; Takeda et al., *Cereb. Cortex*, 2004; Ifft et al., *Front. Neuroeng.*, 2012; Logiaco et al., *BMC Neurosci.*, 2013), our primary analyses were over a 250 ms period.”

Moreover, we further explain the motivation for conducting these additional analyses for the extended window: “To evaluate the potential impact of analysis duration, we extended the time window to 500 ms and found that extending this period did not qualitatively alter our key results (extended 500 ms period data in Table S10).”

2. Regarding the response to my comment #2, I appreciate the updated results shown in Figure S6, which appear more consistent with the trend observed in the main Figure 2. However, I noticed that the results of the rate coding analysis have changed substantially compared to the original manuscript, and this change is not explicitly acknowledged or discussed in the rebuttal.

I strongly encourage the authors to transparently report and briefly explain the reasons for these changes (e.g., bug fixes, analysis corrections, or different time windows). Additionally, they should comment on whether and how these changes affect the main conclusions of the manuscript.

Thank you for bringing this to our attention. The changes in the rate coding results are due to corrections made during our data preprocessing stage. When processing raw neural data, especially focusing on spikes frequency, it is important to assess the utility of each unit/electrode. To be specific, we applied an exclusion criterion, involving

- Missing raw data: Units/Electrodes with missing spike information were excluded. These entries were typically represented as empty arrays ([]).
- Abnormal raw data: Units/Electrodes with extreme or implausible values, such as negative values.

In the earlier version, the rate coding (spike frequency) analysis was performed without applying proper exclusion of low-quality or non-responsive units, which may have obscured the detection of the stretching effect observed in our study. After incorporating preprocessing procedures, we are able to detect the stretching effect using rate coding measures. In contrast, temporal spiking measures (e.g., ISI and SPIKE-distance) are more robust to such low-quality raw data issues and consistently reveal the stretching effect across analyses.

In the revised manuscript, we have added on Line 368: “We excluded electrodes with missing recordings (e.g., empty units) and abnormal values (e.g., negative values) in the raw data.”

3. In response to my previous concern (comment #3), the authors revised the manuscript to state that the results were “consistent with our understanding of the function of these areas” (line 178). However, the phrase “our understanding” remains vague and may be difficult for readers unfamiliar with the domain to interpret.

I recommend replacing this subjective phrasing with more objective language. For instance, referencing existing literature or known functional characteristics of the regions (e.g., MT, V4) would clarify the basis of the interpretation. The rebuttal provides a satisfactory explanation, and incorporating that into the manuscript would resolve the issue.

We fully agreed with this recommendation. In the revised manuscript, we have added: “While significant, MT and V4 exhibit weaker weight changes across tasks (Fig. 4a), which is consistent with known functional specializations of these regions -- for example, the involvement of MT in motion processing and V4 in encoding shape and color information (Roe et al., Neuron, 2012; Born et al., Annu. Rev. Neurosci., 2005).”

*Reviewer #2 (Remarks on code availability):
I have no further concerns regarding the code.*

Thank you.

Reviewer #3 (Remarks to the Author):

I appreciate the edits and clarifications provided in this new revision.

We sincerely thank the Reviewer for their very positive feedback and appreciation of the revisions.

If in any way possible, I would like to ask the editor to allow showing the ROI-specific data in the main paper, instead of only the grand average data.

The grand average is misleading given the different effect sizes, as discussed with and appreciated by the authors earlier. Thus, if space is needed, an option to consider would be to move the grand average plots to the supplement, or dropping them altogether, and to be replaced them with the ROI-specific plots.

Thank you for highlighting the importance of ROI-specific results. As we stated in the previous revision letter, we are open to moving this material to the main text. We now flagged this issue in the cover letter to the editor for their consideration. Within the formatting limitations of this journal, we did our best to convey the basic results that we think will interest the average reader. In service of completeness, we have included comprehensive ROI-specific plots and results in the Supplementary Information and clearly referenced them in the main text. Should the Editor deem it appropriate and be willing to allocate additional space in the main text, we would be happy to move figures that are currently in the Supplementary Information to the main text. We leave the issue to the Editor to decide. If expansion is not possible, at least the ROI-specific information is available for interested readers and referenced in the main text.

REVIEWERS' COMMENTS

We thank all Reviewers for their very positive assessment and constructive feedback. We have carefully addressed Reviewer 1's suggestion, and the Supplemental Information file has been updated accordingly.

Reviewer #1 (Remarks to the Author):

This most recent revision has further improved the manuscript. I think, for the most part, that we have reached the point of diminishing returns, and that the paper can be published with only a few minor tweaks that do not require further review. Specifically, I think that the authors should put ALL of the figures that they provided to the reviewers into the supplementary figures of the paper. There is no real limit on supplementary figures, so there is plenty of room to put them in there. If the information that they provide was valuable for the reviewers, it will be valuable for the readers.

We thank the Reviewer for this suggestion. All figures that were provided during the review process have now been included in the Supplementary Figures (Fig. S10 & S12), as recommended.

Reviewer #2 (Remarks to the Author):

The authors have made reasonable efforts to revise the manuscript, and I appreciate their responsiveness to the earlier comments. Key issues raised in previous rounds—such as the rationale for the analysis window and the preprocessing steps—have been addressed appropriately.

I consider the current version acceptable for publication. I thank the authors for their engagement throughout the review process.

We sincerely thank the Reviewer for their positive assessment and appreciate their acknowledgment of our revisions. We are glad that the rationale for the analysis window and preprocessing steps are now clear.

Reviewer #2 (Remarks on code availability):

I have no further concerns regarding the code.

We thank the Reviewer for confirming that the code availability is satisfactory.